# TBC1D23 mediates Golgi-specific LKB1 signaling

Yingfeng Tu[1,9], Qin Yang[1,9], Min Tang[1,9], Li Gao[2], Yuanhao Wang[3], Jiuqiang Wang[4,5,6,7], Zhe Liu[1], Xiaoyu Li[1], Lejiao Mao[1], Rui zhen Jia[1], Yuan Wang [2], Tie-shan Tang [4,5,6], Pinglong Xu [8], Yan Liu [3], Lunzhi Dai [2] & Da Jia [1]✉

Liver kinase B1 (LKB1), an evolutionarily conserved serine/threonine kinase, is a master regulator of the AMPK subfamily and controls cellular events such as polarity, proliferation, and energy homeostasis. Functions and mechanisms of the LKB1-AMPK axis at specific subcellular compartments, such as lysosome and mitochondria, have been established. AMPK is known to be activated at the Golgi; however, functions and regulatory mechanisms of the LKB1-AMPK axis at the Golgi apparatus remain elusive. Here, we show that TBC1D23, a Golgi-localized protein that is frequently mutated in the neurodevelopment disorder pontocerebellar hypoplasia (PCH), is specifically required for the LKB1 signaling at the Golgi. TBC1D23 directly interacts with LKB1 and recruits LKB1 to Golgi, promoting Golgi-specific activation of AMPK upon energy stress. Notably, Golgi-targeted expression of LKB1 rescues TBC1D23 deficiency in zebrafish models. Furthermore, the loss of LKB1 causes neurodevelopmental abnormalities in zebrafish, which partially recapitulates defects in TBC1D23-deficient zebrafish, and LKB1 sustains normal neuronal development via TBC1D23 interaction. Our study uncovers a regulatory mechanism of the LKB1 signaling, and reveals that a disrupted Golgi-LKB1 signaling underlies the pathogenesis of PCH.

Eukaryotic cells utilize sophisticated mechanisms to sense cellular nutrient supply and to regulate cell growth and proliferation. Liver kinase B1 (LKB1, also known as STK11), a highly conserved and ubiquitously expressed master serine/threonine kinase, is one of the major players in coordinating cellular energy status and cell growth[1,2]. LKB1 was initially identified to be a major upstream kinase of the nutrient sensor AMPK[3,4]. Upon energy stress, LKB1 phosphorylates conserved threonine (Thr172) in AMPKα and activates AMPK[5]. Once activated, AMPK phosphorylates many downstream substrates critical for regulating metabolic processes to maintain metabolic homeostasis[6]. Based on its essential role in maintaining cellular and organismal energy homeostasis, AMPK has attracted intense interest as a therapeutic avenue for various diseases, including diabetes, obesity and cancer. In addition to its canonical role in energy metabolism, AMPK also regulates other cellular events, such as T cell activation, cell adhesion and migration[7,8]. In contrast to AMPK-dependent roles of LKB1 in rewiring metabolism, LKB1 functions in the nervous system

mainly by activating other downstream effectors[1,9]. In addition to AMPKα1/α2, LKB1 also phosphorylates twelve other members of the AMPK subfamily, including BRSK1/2, NUAK1/2, SIK1/2/3, MARK1/2/3/4 and SNRK[10,11]. Accordingly, recent studies have uncovered critical roles of LKB1 in processes beyond controlling cellular metabolism, such as regulating neuronal development and homeostasis. LKB1 is necessary to promote neuronal polarity in cultured neurons and developing cortical neurons, and loss of LKB1 almost abolishes axonal formation[12,13]. LKB1-mediated activation of BRSK1/2 is critical for its role in neuronal polarity[12–14]. LKB1 regulates axon terminal branching and dendrite spacing through NUAK1 and SIK, respectively[15,16]. In addition, LKB1 is also essential for neuronal migration, myelination and maintenance of neural integrity[9]. Therefore, LKB1 can function in an AMPK-dependent or -independent manner.

LKB1 has multiple subcellular localizations. One emerging concept is that each compartmentalized pool of LKB1 is differentially regulated and has distinct functions[17–19]. LKB1 is mainly localized in the

nucleus under resting conditions, and the formation of the LKB1/STRADα/MO25 ternary complex promotes the nuclear export and activation of LKB1[20–22]. In lysosomes, a crucial hub for glucose starvation-induced AMPK activation, the scaffold protein AXIN is responsible for tethering LKB1 to the lysosome and subsequent lysosomal AMPK (Lyso-AMPK) activation[23]. Cytosolic LKB1 mediates activation of cytosolic AMPK in an AXIN-dependent or -independent manner, determined by the severity of energy stress[19]. In mitochondria, LKB1-mediated AMPK activation leads to the phosphorylation of downstream proteins such as mitochondrial fission factor (MFF) and Armadillo repeat containing protein 10 (ARMC10) and regulates mitochondrial biogenesis, fission/fusion, and mitophagy[19,24–27]. Therefore, compartmentalized LKB1 is a critical determinant of downstream signaling. In addition to lysosome and mitochondria, recent studies using genetically encoded fluorescent AMPK activity sensors have detected AMPK activity across other subcellular compartments, including Golgi[28,29]. Of note, a previous study reported that energy stress induces robust activation of AMPK at the Golgi (Golgi-AMPK)[29]. However, the regulatory components of Golgi-AMPK remain entirely unknown, and it is yet to be determined whether LKB1 is involved. The illustration of the LKB1-AMPK signaling at Golgi will provide more comprehensive understanding of the LKB1 signaling at subcellular level.

TBC1D23, a Golgi-localized Tre2-Bub2-Cdc16 (TBC) family member, is highly conserved in many eukaryotic organisms and ubiquitously expressed in various tissues and cells, indicating its fundamental roles[30]. TBC1D23 contains an N-terminal catalytically inactive TBC domain, followed by a Rhodanese-like domain and a C-terminal pleckstrin homology (PH) domain[30,31]. The TBC and Rhodanese domains of TBC1D23 bind to the Golgi proteins golgin-97/golgin-245, which determines its Golgi localization[31,32]. Intriguingly, mutations in TBC1D23 lead to pontocerebellar hypoplasia (PCH), a group of rare neurological disorders characterized by impaired development of the brain, especially the pons and cerebellum[33,34]. On the other hand, TBC1D23, via its C-terminal PH domain, interacts with the WASH complex component, a major regulator of actin cytoskeleton on endosomes[31,35]. Thus, TBC1D23 serves as an adapter to bridge endosomal vesicles with the Trans-Golgi Network (TGN), and mediates endosome-to-Golgi trafficking of cargo molecules, including cation-independent mannose-6-phosphate receptor (CI-MPR) and TGN46[31,32,35]. Despite these advances, it remains to be established how mutations of TBC1D23 lead to the pathogenesis of PCH.

Here, we identify TBC1D23 as a regulator of the LKB1 signaling. TBC1D23 specifically controls AMPK activation at the Golgi, but does not affect the lysosomal and mitochondrial AMPK activation. Mechanistically, we show that TBC1D23 directly interacts with LKB1 and promotes the recruitment of LKB1 to the Golgi, thus contributing to Golgi-AMPK activation. Furthermore, TBC1D23 and LKB1 cooperate to control neuronal development in zebrafish models. Collectively, our study establishes TBC1D23 as a Golgi-specific scaffolding protein for Golgi-LKB1 signaling, and reveals distinct functions and regulatory mechanisms of the LKB1 signaling at different cellular compartments.

## Results
### TBC1D23 interacts with LKB1 to promote AMPK activation in response to energy stress
TBC1D23 was previously identified as a bridging factor for endosome-derived vesicles and golgins at Trans-Golgi Network (TGN)[31,32,35]. To further characterize the functions of TBC1D23, we analyzed proteins associated explicitly with GST-tagged TBC1D23 versus GST from HEK293T cells, using mass spectrometric analysis (Fig. 1a, b). Besides several reported TBC1D23-interacting proteins, such as subunits of the WASH complex and WDR11 complex, and golgin-97[31,32,35,36], we identified the liver kinase B1 (LKB1) complex and AMPKα1 as potential TBC1D23-interactors (Fig. 1c, and Supplementary Data 1). Reactome

gene sets and canonical pathway analysis also indicated that the LKB1 pathway, in addition to vesicle-mediated transport, is among the top TBC1D23-associated pathways (Supplementary Fig. 1a, b). Indeed, ectopically expressed GST-TBC1D23 could readily precipitate LKB1, similar to the WASH complex subunit FAM21 (Fig. 1d). In contrast, the association between TBC1D23 and AMPKα1 or MO25, a subunit of the LKB1 complex, seemed much weaker than those of TBC1D23-LKB1 and TBC1D23-FAM21 (Fig. 1e and Supplementary Fig. 1c). These data suggest that LKB1 could be an interaction partner of TBC1D23.

Next, we sought to determine the functional significance of the TBC1D23-LKB1 interaction. Since LKB1 is the primary upstream kinase responsible for AMPK activation upon energy stress[3,4], we examined whether TBC1D23 deficiency would affect energy stress-induced AMPK activation, by assessing the phosphorylation levels of AMPKα at T172. LKB1 was reported to be required for activation of AMPK in HUVEC cells[37], so we first examined effects of TBC1D23 knockdown on AMPK activation in HUVEC cells. Using siRNA specially targeting TBC1D23, we were able to efficiently abrogate expression of TBC1D23. Activation of AMPK, induced by glucose starvation (GS) and CCCP treatment, was significantly compromised in siTBC1D23 cells, relative to control cells (siNC) (Fig. 1f). Importantly, depletion of TBC1D23 did not alter the expression levels of LKB1 or AMPKα (Fig. 1f). To test whether TBC1D23 regulates the LKB1-AMPK pathway in other cell lines, we chose hepatocyte cell lines HepG2 for subsequent experiments and generated TBC1D23 knockout (TBC1D23 KO) HepG2 cells using the CRISPR–Cas9 technology. Consistent with our results from HUVEC cells, loss of TBC1D23 in HepG2 cells resulted in remarkably reduced AMPK activation upon energy stress (Fig. 1g, h and Supplementary Fig. 1d–g), suggesting that TBC1D23 and LKB1 may function together to promote AMPK activation. Furthermore, we found that TBC1D23 KO cells exhibited higher sensitivity to glucose starvation, as revealed by significantly decreased cell proliferation and colony formation (Fig. 1i, j). In contrast, abrogation of TBC1D23 did not alter the cell cycle progression and cell proliferation under normal conditions (Supplementary Fig. 1h, i).

To exclude the possibility that TBC1D23 regulates AMPK activation via CaMKK2[38–40], we chose HeLa and A549 cells which express CaMKK2, but lack the expression of LKB1. Calcium ionophore A23187, used to activate CaMKK2, activated AMPK in the TBC1D23-deficient HeLa and A549 cells to a comparable extent as the WT cells, indicating that deletion of TBC1D23 did not affect the CaMKK2-AMPK pathway (Fig. 1k, and Supplementary Fig. 1j). Altogether, these results suggest that TBC1D23 functions to promote AMPK activation, likely via the interaction with LKB1.

### TBC1D23 specifically regulates Golgi-AMPK activation
As TBC1D23 mainly localizes at the Golgi[31,32,35], we next probed the Golgi localization of LKB1 and AMPKα. We found that the co-localization between LKB1 and Golgi marker golgin-97 were negligible under basal conditions. However, LKB1 was translocated to Golgi upon glucose starvation (Fig. 2a). We also observed LKB1 puncta after glucose starvation, which might be endosomal and/or lysosomal LKB1[18,23,41]. In contrast with LKB1, AMPKα partially co-localized with Golgi marker golgin-97 under basal conditions and upon glucose starvation (Fig. 2b), indicating the presence of a residential Golgi pool of AMPK.

We next investigated whether TBC1D23 specifically affected AMPK activation at the Golgi. We utilized the recently-developed organelle-specific AMPK activity probes[29], which fuse genetically encodable Forster resonance energy transfer (FRET)-based reporters with organelle-targeting sequences (Fig. 2c). In TBC1D23 KO cells, activation of Golgi-AMPK was significantly compromised relative to WT cells upon glucose starvation (1.27 in WT cells vs 0.91 in TBC1D23 KO cells) (Fig. 2d and Supplementary Fig. 2a). Interestingly, glucose starvation still induced the activation of Golgi-AMPK in TBC1D23 KO

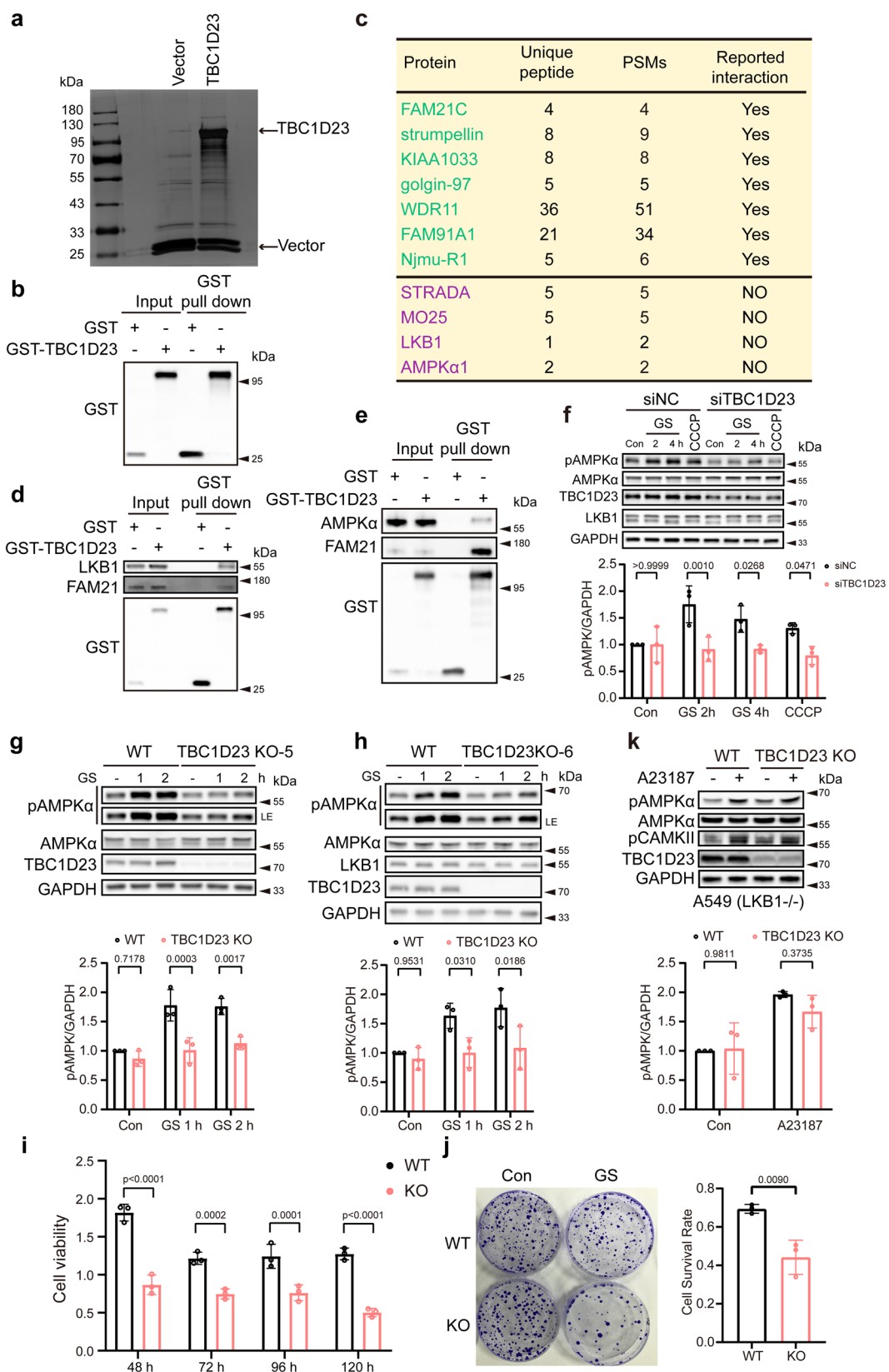

**c**

| Protein | Unique peptide | PSMs | Reported interaction |
|---------|----------------|------|----------------------|
| FAM21C | 4 | 4 | Yes |
| strumpellin | 8 | 9 | Yes |
| KIAA1033 | 8 | 8 | Yes |
| golgin-97 | 5 | 5 | Yes |
| WDR11 | 36 | 51 | Yes |
| FAM91A1 | 21 | 34 | Yes |
| Njmu-R1 | 5 | 6 | Yes |
| STRADA | 5 | 5 | NO |
| MO25 | 5 | 5 | NO |
| LKB1 | 1 | 2 | NO |
| AMPKα1 | 2 | 2 | NO |

cells, indicating that some unidentified factor(s) might promote Golgi-AMPK activation. We also observed a decrease of Golgi-AMPK activity under basal conditions (1.00 in WT cells vs 0.79 in TBC1D23 KO cells) (Fig. 2d). In contrast, depletion of TBC1D23 did not alter activity of mitochondrial AMPK (Mito-AMPK) under basal conditions or upon glucose starvation (Fig. 2e and Supplementary Fig. 2b). To further consolidate our conclusions, we assessed AMPK activation in Golgi or mitochondria in HeLa cells, in which LKB1 is naturally missing, using the identical probes. Consistent with the previous report that LKB1 mediates AMPK activation at mitochondria[19], glucose starvation did

**Fig. 1 | TBC1D23 is an interactor of LKB1 and is required for stress-induced AMPK activation. a** Silver staining gel of proteins precipitated from HEK293T cells transfected with GST-TBC1D23 or GST-empty vector. Specific TBC1D23-interacting proteins were resolved by SDS-PAGE and silver staining, followed by mass spectrometry. **b** The enrichment of TBC1D23 is confirmed by western blotting and IP-MS data (Supplementary Data 1). **c** Partial list of proteins identified by mass spectrometry analysis. PSMs: peptide-to-spectrum matches. **d, e** HEK293T cells were transfected with GST-vector or GST-TBC1D23, followed by pull down assay with GST beads and immunoblotting with indicated antibodies. **f** HUVEC cells transduced with control siRNA or siTBC1D23 were subjected to glucose starvation (GS) for the indicated time, or treated with CCCP (20 μM, 4 h). **g, h** WT, TBC1D23 KO single clone KO-5 (**g**) and KO-6 (**h**) HepG2 cells glucose starved for the indicated time were lysed and subjected to immunoblotting. LE: long exposure. **i, j** WT or TBC1D23 KO HepG2 cells were glucose starved for 12 h, and cultured for the indicated time. The cell viability was normalized to that of 0 h (**i**).WT vs KO: 48 h, *p* value: 1.95e−008; 72 h, *p* value: 1.65e−004; 96 h, *p* value: 1.27e−004; 120 h, *p* value: 3.62e−007; WT and TBC1D23 KO HepG2 cells were subjected to glucose starvation for 12 h and cultured for 14 days, then stained by crystal violet. The number of colonies and cell survival rate were normalized to control cells without glucose starvation. Data are the mean of biological replicates from a representative experiment (**j**). **k** WT and TBC1D23 KO A549 cells were incubated with or without 2 μM A23187 for 30 min; cells were then lysed and analyzed by indicated antibodies. The graph shows the levels of pAMPK quantified by densitometry and normalized to GAPDH (**f, g, h, k**). Experiments were performed in triplicate (**d–k**). Quantification data are the mean ± SD (**f–k**), with *p* values calculated by two-way ANOVA, followed by Sidak's test (**f, g, h, i, k**), or by unpaired two-tailed t test (**j**). Source data are provided as a Source data file.

not alter the Mito-AMPK activity in HeLa cells (Supplementary Fig. 3a). Similarly, we failed to detect the change of Golgi-AMPK activity in HeLa cells, indicating that LKB1 might be responsible for activation of Golgi-AMPK as well (Supplementary Fig. 3b). AICAR (5-Aminoimidazole-4-carboxamide ribonucleotide), a widely used AMPK activator, was reported to cause the phosphorylation of GBF1 (Golgi-specific Brefeldin A Resistance Factor 1) at T1337 and subsequent Golgi disassembly[42]. To determine the role of LKB1 in Golgi-AMPK activation, we treated cells with AICAR. Notably, the introduction of LKB1 in HeLa Cells significantly activated total AMPK and Golgi-AMPK in the presence or absence of AICAR, as revealed by phosphorylation of AMPK and Golgi-localized GBF1, a reported substrate of AMPK[42,43] (Supplementary Fig. 3c). To further demonstrate the role of LKB1 in Golgi-AMPK activation in response to energy stress, we performed Golgi extraction in control HeLa cells (plvx neo) and cells stably expressing LKB1. Consistent with the essential role of LKB1 in AMPK activation in response to energy stress, introduction of LKB1 remarkably promoted AMPK activation of the Golgi fraction upon AICAR treatment (Supplementary Fig. 3d). Therefore, TBC1D23 promotes the activation of Golgi-AMPK, but not Mito-AMPK.

To investigate whether TBC1D23 regulates the activation of Lyso-AMPK, we treated WT and TBC1D23 KO cells with low-dose metformin, which specifically activates Lyso-AMPK[44,45]. Similar levels of AMPK phosphorylation were detected in WT and TBC1D23 KO cells, indicating that TBC1D23 deletion did not impair the activation of Lyso-AMPK (Fig. 2f and Supplementary Fig. 3e). To further confirm our results, we measured the activation of Lyso-AMPK using Lyso- ExRai-AMPKAR in WT and TBC1D23 KO cells[46]. Consistent with previous studies, glucose starvation caused a significant increase of Lyso-AMPK activity, as revealed by upregulated 480 nm excitation/400 nm excitation of Lyso-ExRai-AMPKAR[46]. TBC1D23 deletion did not impair the activation of Lyso-AMPK (Fig. 2g, h). Thus, we concluded that TBC1D23 specifically promotes Golgi-AMPK activation, likely via LKB1.

## TBC1D23 preferentially promotes the phosphorylation of Golgi-localized proteins

We hypothesized that TBC1D23 recruits LKB1 to Golgi to activate AMPK and other downstream effectors. To determine how TBC1D23 contributed to phosphorylation events, we performed global quantitative phosphoproteomic analysis by mass spectrometry (MS) using wild-type and TBC1D23 knockout HEK293T cells upon glucose starvation with 3 repeats. As a result, 103 phosphosites were found to have significantly lower levels in TBC1D23 KO cells (fold-change(TBC1D23 KO/WT) < 0.67, and Student's t-test, *p* < 0.01) (Fig. 3a), including ULK1-pS555, a known AMPK-dependent phosphorylation site[47–49] (Supplementary Data 2). In addition, we also identified 40 phosphosites with higher phosphorylation levels in TBC1D23 KO cells (fold-change(TBC1D23 KO/WT) > 1.5 and Student's t-test, *p* < 0.01) (Fig. 3a and Supplementary Data 2).

We speculated that the TBC1D23-dependent phosphorylation sites should be enriched in Golgi-localized proteins relative to AMPK-dependent sites. Indeed, comparison of our phosphoproteomic data and previously published phosphoproteomic data affected by AMPKα1/α2 double knockout (DKO) revealed that Golgi-localized proteins are highly enriched in TBC1D23 samples, including both downregulated and upregulated phosphosites[26]. Among the downregulated phosphosites, 1.8% of AMPKα-dependent sites are from proteins localized at the Golgi; in contrast, 10.6% of TBC1D23-dependent sites are at the Golgi. Similarly, among the upregulated phosphosites, 5.9% and 10% of AMPKα- and TBC1D23-dependent sites are from proteins at the Golgi, respectively (Fig. 3b). Furthermore, four of top 10 downregulated phosphosites due to the depletion of TBC1D23 are from proteins localized at Golgi (Fig. 3c). We chose ARF1 (S147), top of the list when ranked by p value, for further analysis.

To analyze how TBC1D23 contributes to ARF1 phosphorylation at S147, we transfected Flag-tagged ARF1 in WT or TBC1D23 KO cells and harvested the cells after glucose deprivation for 2 h. Consistent with our phosphoproteomic data, the phosphorylation level of ARF1 in TBC1D23 KO cells, detected with an anti-phospho-(Ser/Thr) antibody, was significantly lower than that in WT cells (Fig. 3d). Furthermore, S147 was primary phosphorylation site affected by depletion of TBC1D23, as the ARF1 S147A mutant displayed similar phosphorylation levels in WT cells and TBC1D23 KO cells (Fig. 3e). In addition to ARF1, we also assessed how depletion of TBC1D23 contributed to phosphorylation of GBF1 at T1337. Golgi-localized GBF1 functions as a guanine nucleotide exchange factor for ARF1 and ARF4, and regulates anterograde trafficking and dynamics of Golgi apparatus[42,43,50]. AMPK is known to mediate GBF1 phosphorylation at T1337 in response to low-glucose or AMPK activator treatment, such as AICAR and 2-DG[42,50]. Depletion of TBC1D23 led to a significantly downregulated GBF1 p-T1337 level upon glucose deprivation (Fig. 3f), consistent with defective Golgi-AMPK activation (Fig. 2d). Since AMPK-mediated GBF1 phosphorylation at T1337 promotes Golgi disassembly[42], we next determined whether compromised Golgi-AMPK activation due to TBC1D23 depletion also impacted Golgi disassembly. Under basal conditions, Golgi apparatus displayed a similar stacked perinuclear structures in both WT and TBC1D23 KO cells, as revealed by immunostaining of Golgi marker protein GM130. Treatment with high-dose metformin caused fragmentation of Golgi in WT cells, which appeared as multiple punctuate structures surrounding the nucleus (Fig. 3g, h). By contrast, deletion of TBC1D23 prevents Golgi disassembly upon metformin treatment, as determined by the number of GM130-positive structures (Fig. 3g, h). To minimize the impact of manual counting, we also determined the ratio between the total area occupied by GM130 and the nucleus (DAPI), GM130/DAPI area. Consistently, TBC1D23 KO did not affect the GM130/DAPI area under basal conditions. However, TBC1D23 KO cells displayed a significantly decreased GM130/DAPI area value, indicating defective Golgi disassembly (Fig. 3h). Altogether,

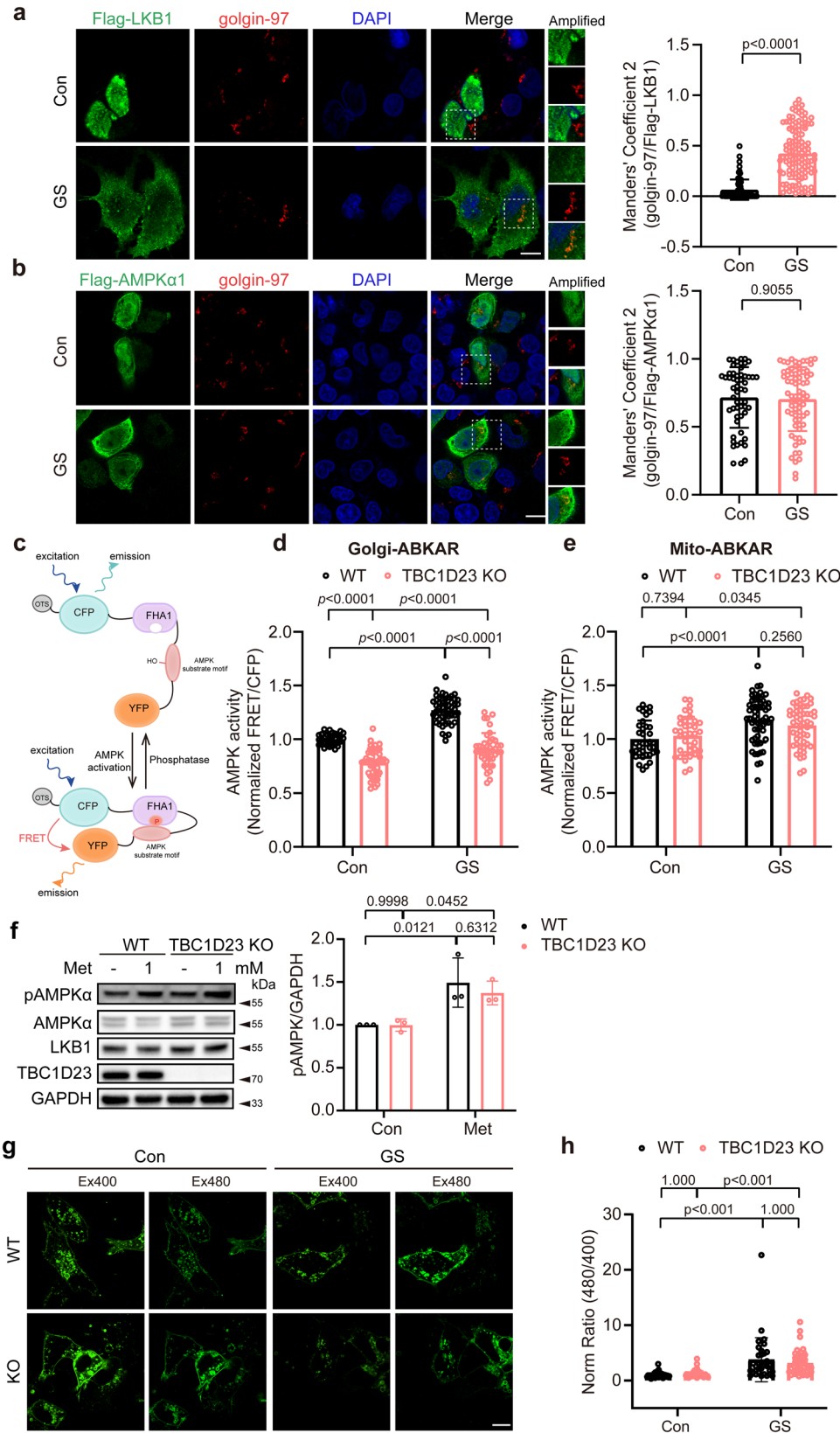

our data indicate that TBC1D23 preferentially promotes the phosphorylation of Golgi-localized proteins.

## TBC1D23 directly interacts with LKB1

So far, our data indicate that TBC1D23 and LKB1 might cooperate to promote AMPK activation at Golgi. Next, we sought to further characterize their interactions. To map the domain of TBC1D23 that is responsible for its binding to LKB1, we generated a series of truncations of TBC1D23, and found that Pleckstrin homology (PH) domain of TBC1D23 (comprising residues 545–684) was necessary and sufficient for the interaction with LKB1 (Fig. 4a–c). It should be noted that it is the same domain that interacts with WASH complex subunit FAM21[31,35].

**Fig. 2 | Deletion of TBC1D23 specifically impairs Golgi-AMPK activation.**
**a** HepG2 cells were transiently transfected with Flag-LKB1 for 24 h, and incubated with or without glucose-free medium for 2 h. Cells were stained with antibodies against Flag and golgin-97, a Golgi marker. Scale bar, 10 μm. Mander's coefficient 2 (golgin-97 overlapping with Flag-LKB1) were graphed as mean ± SD. *p* value: 2.41e−030. **b** HepG2 cells transiently transfected with Flag-AMPKα1 were subjected to glucose starvation for 2 h. Scale bar, 10 μm. Mander's coefficient 2 (golgin-97 overlapping with Flag-AMPKα1) were graphed as mean ± SD. Each dot represents Manders' coefficients from one cell (**a**, **b**). **c** Schematic diagram of subcellular compartment-specific biosensor osABKAR. **d**, **e** WT and TBC1D23 KO HepG2 cells transiently transfected with Golgi-ABKAR (**d**) or Mito-ABKAR (**e**) were incubated with glucose-free medium for 2 h and 4 h, respectively. The FRET/CFP ratio of WT cells incubated with DMEM was set as 1, and FRET/CFP ratio of other groups was normalized to WT cells incubated with DMEM. **d** WT Con vs KO Con: *p* value: 1.47e

−011; WT GS vs KO GS: *p* value: 0.00e + 000; WT Con vs WT GS: *p* value: 0.00e + 000; KO Con vs KO GS, *p* value: 4.07e−005. **e** WT Con vs WT GS: *p* value: 1.91e−005. **f** WT and TBC1D23 KO HepG2 cells were treated with or without 1 mM metformin for 2 h. Cells lysates were analyzed by immunoblotting. The graph shows the levels of pAMPK quantified by densitometry using Image J software and normalized to GAPDH. **g**, **h** WT and TBC1D23 KO HepG2 cells transiently transfected with Lyso-ExRai-AMPKAR for 24 h were incubated with or without glucose-free medium for 2 h. Representative images (**g**) and statistical analysis (**h**) of Lyso-AMPK activity measured by Lyso-ExRai-AMPKAR. Scale bar, 10 μm. Similar results were obtained in three independent experiments (**a–h**). **a–h** Statistical data are presented as mean ± SD. *P* values were determined by unpaired two-tailed Mann–Whitney test (**a**, **b**), or by two-way ANOVA, followed by Sidak's test (**d–f**), or by Scheirer-Ray-Hare Test (**h**). Source data are provided as a Source data file.

Our previous studies indicated that three consecutive positively-charged residues (K632K633 K634) in the PH domain of TBC1D23 are critical for its binding to FAM21[35]. To determine whether the same residues are involved in LKB1 interaction, we generated a triple mutant by converting all three residues to the oppositive charge (3 K: K632E/K633E/K634E). TBC1D23 3K mutant almost abolished the binding to LKB1 (Fig. 4d). These results suggested that LKB1 and FAM21 might compete with each other for binding to TBC1D23.

We then defined which domain of LKB1 interacted with TBC1D23. Full-length LKB1 (LKB1 FL) robustly immunoprecipitated TBC1D23. Further truncation experiments demonstrated that the C-terminal regulatory domain (CRD) of LKB1 was necessary and sufficient to mediate the interaction (Fig. 4e, f). We discovered a motif (aa 349–364) in the CRD of LKB1 that is highly analogous to the LFa motif of FAM21 (L-F-[D/E]3-10-L-F), which mediates the interaction with TBC1D23[35]. Indeed, GST-pulldown experiments using purified proteins demonstrated that the LFa motif of LKB1 directly interacted with the PH domain of TBC1D23 (Fig. 4g). The interaction was impaired when mutagenesis was introduced in the residues preceding the LFa motif (4 K), the first 2 residues (2 A), or the 6th and 7th residues of LFa motif (2 K) (Fig. 4g, h). These mutagenesis data further indicated that LKB1 binds to TBC1D23 in a manner analogous to FAM21. Finally, Isothermal Titration Calorimetry (ITC) experiments demonstrated that the LFa motif of LKB1 bound to TBC1D23 PH domain with a stoichiometry of 1:1, and with a high affinity (Kd = 800 nM, Ka = $1.25 \times 10^6 \, M^{-1}$) even in the presence of 500 mM NaCl (Fig. 4i). Taken together, our data demonstrate that TBC1D23 associates with the LKB1 complex via a direct interaction with LKB1.

## Glucose starvation and consequent AMPK activation promote the interaction between TBC1D23 and LKB1

LKB1 predominantly locates in the nucleus under basal conditions. The formation of a heterotrimeric complex comprised of LKB1, STRAD and MO25 is required for the activation and cytosolic localization of LKB1[20–22]. Axis inhibition protein (AXIN) tethers LKB1 to phosphorylate AMPK upon glucose starvation and thus plays a crucial role in lysosomal AMPK activation[23]. As glucose starvation dramatically enhances the interaction between AXIN and LKB1[23], we then sought to explore whether and how glucose starvation affected the TBC1D23-LKB1 association. We found that the association between LKB1 and TBC1D23 was temporally regulated upon glucose deprivation. The interaction was dramatically increased 0.5 h after glucose starvation, peaked around 2 h, and decreased at 4 h (Fig. 5a). In addition to full-length LKB1, the interaction between LKB1 CRD and TBC1D23 is also dynamically regulated (Fig. 5b and Supplementary Fig. 4a).

As the dynamic interaction pattern between TBC1D23 and LKB1 was highly similar to that of AMPKα activation upon glucose starvation, we suspected that AMPK may play a role in modulating the TBC1D23-LKB1 interaction. Indeed, co-transfection of AMPKα could stimulate the interaction between TBC1D23 and LKB1 FL under basal

conditions and upon glucose starvation (Fig. 5c,d). Intriguingly, AMPKα promoted the interaction between TBC1D23 and LKB1 CRD only under the condition of glucose starvation, but not basal conditions (Fig. 5e,f). LKB1 FL, but not LKB1 CRD, is capable to promote AMPK activation, suggesting that activated AMPK is required to promote the interaction between TBC1D23 and LKB1. When LKB1 CRD was used in the experiment, endogenous LKB1-mediated AMPK activation upon glucose starvation, but not in basal conditions, could be responsible to enhance the interaction between TBC1D23 and LKB1 CRD. Altogether, our data suggested a model for the Golgi-AMPK activation upon glucose starvation: (1) glucose starvation promotes the export of LKB1 from the nucleus, which is then recruited to the Golgi by TBC1D23; (2) LKB1 promotes the activation of Golgi-AMPK; (3) activated AMPK further enhances the interaction between TBC1D23 and LKB1, and likely additional AMPK activation, resulting in a positive feedback loop (Fig. 5g and Supplementary Fig. 4b).

## TBC1D23 and LKB1 cooperate to promote Golgi-AMPK activation

So far, our data indicate that TBC1D23 might function as a Golgi-specific scaffolding factor of LKB1 to promote Golgi-AMPK activation. Indeed, deletion of TBC1D23 led to a dramatically decreased recruitment of LKB1 to Golgi, as determined by quantitative immunofluorescence, upon glucose starvation (Fig. 6a). Similarly, TBC1D23 WT, but not the 3 K mutant that is defective for LKB1 binding, could rescue the impaired AMPK activation (Fig. 6b), further confirming that TBC1D23 recruits LKB1 to Golgi to activate AMPK.

To further test our hypothesis, we generated a LKB1-GIANTIN (LKB1-G) chimaera construct by fusing GIANTIN (aa 3131–3259), a Golgi-resident protein, to the C-terminus of LKB1 (Supplementary Fig. 5a). Immunofluorescence analysis indicated that LKB1-G was efficiently confined to the Golgi apparatus (Supplementary Fig. 5b,c). We further compared the ability of LKB1 and LKB1-G in rescuing defective Golgi-AMPK activation in TBC1D23-deficient cells. As shown in Fig. 6c, LKB1-G was much more efficient than LKB1 to promote AMPK activation and subsequent GBF1 phosphorylation. The Golgi-targeting LKB1 kinase-dead mutant (LKB1-G D194A) was not as effective as LKB1-G WT, indicating that both the Golgi localization and the kinase activity were critical.

As deletion of TBC1D23 led to impaired Golgi disassembly upon high-dose metformin treatment, we assessed the ability of LKB1, LKB1-G WT, and LKB1-G D194A in rescuing Golgi disassembly in TBC1D23 KO cells. Consistent with our biochemical assays, LKB1-G WT was more effective than LKB1 and LKB1-G D194A to rescue compromised Golgi disassembly caused by TBC1D23 knockout (Fig. 6d, e). Similar results were obtained when we analyzed Golgi disassembly by determining the ratio between the total area occupied by GM130 and DAPI (Fig. 6e). Altogether, our data indicate that TBC1D23 is a Golgi-specific scaffolding protein for LKB1, and the functions of TBC1D23 could be partially replaced by a Golgi-targeting LKB1.

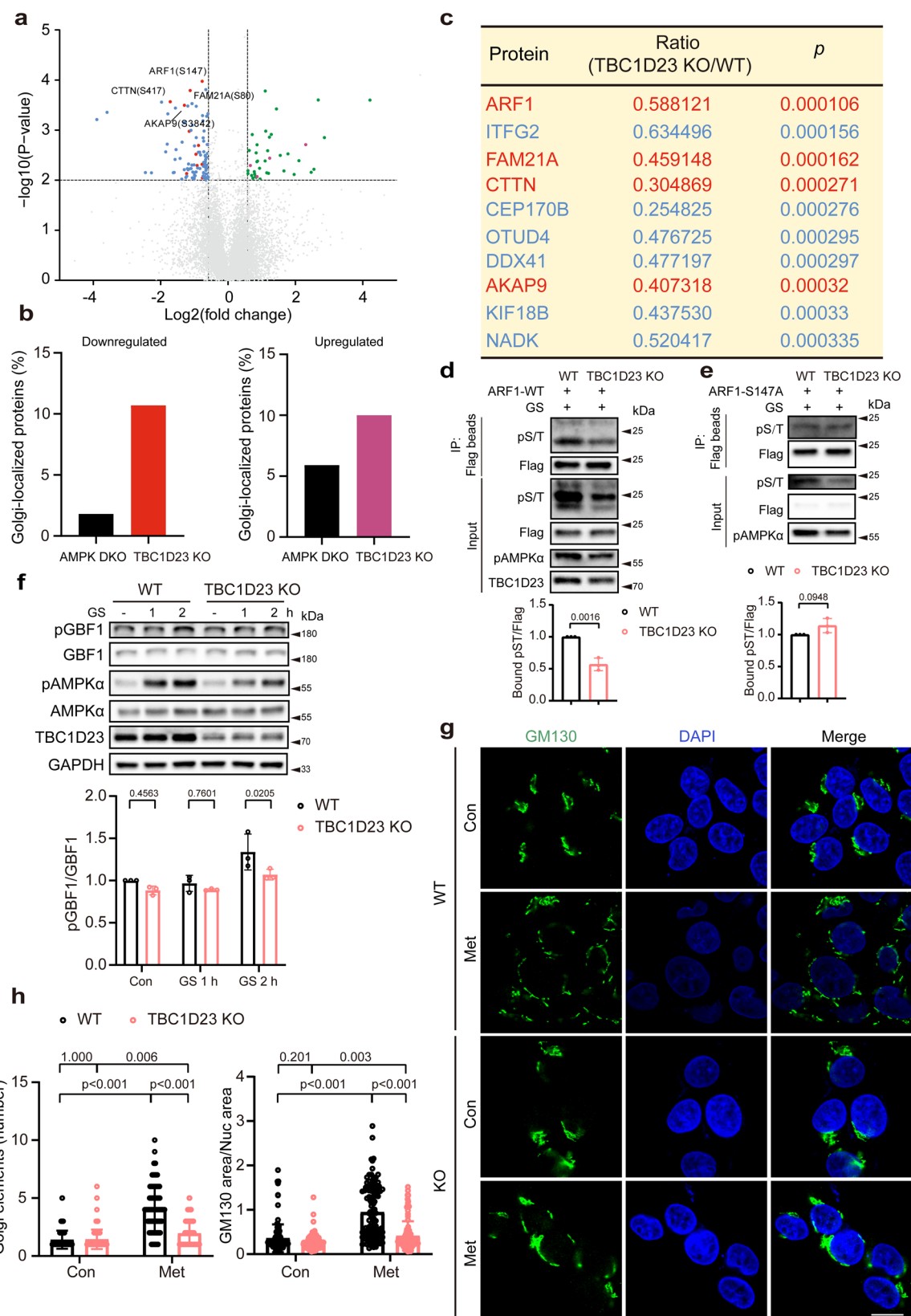

## TBC1D23-mediated AMPK activation likely inhibits endosome-to-TGN trafficking

Previous studies established that TBC1D23 mediates endosome-to-TGN trafficking of specific transmembrane proteins by engaging with a large number of proteins, such as the WASH complex subunit FAM21[31,35]. Since LKB1 and FAM21 bind to the same domain (PH domain) and sites (3 K) of TBC1D23, TBC1D23-mediated AMPK activation could compete with its function in mediating endosome-to-TGN trafficking. Indeed, we found that glucose starvation for 1 h or 2 h led to a decreased association between TBC1D23 and FAM21 (Supplementary Fig. 6a), whereas the TBC1D23-LKB1 interaction was significantly enhanced under this condition.

**Fig. 3 | TBC1D23 preferentially regulates the phosphorylation of Golgi-localized proteins, and TBC1D23 loss prevents Golgi disassembly. a** Volcano plot of quantitative analysis of phosphopeptides identified by MS. Blue or red: peptides with fold-change (TBC1D23 KO:WT) < 0.67 ($p < 0.01$, t-test); green or purple: peptides with fold-change (TBC1D23 KO:WT) > 1.5 ($p < 0.01$, t-test); red or purple: golgi-localized proteins; gray: no significant difference. **b** Combined analysis of phosphosites identified from AMPK DKO (AMPKα1/α2 double knockout) cells and TBC1D23 KO cells. The proportion of Golgi-localized proteins was determined. **c** Top 10 phosphosites with higher levels of phosphorylation in WT cells than TBC1D23 KO cells, ranked by $p$ value. Golgi-localized proteins were shown in red color. **d, e** Constructs encoding Flag-tagged ARF1 WT (**d**) or S147A mutant (**e**) were transfected into WT or TBC1D23 KO HEK293T cells. 24 h later, cells glucose starved for 2 h were collected and subjected to immunoprecipitation using Flag beads and immunoblotting. The graph showed ratios of bound pST to Flag

quantified by densitometry using Image J software. **f** WT and TBC1D23 KO HEK293T were glucose starved for the indicated time, and cell lysates were subjected to immunoblotting. The graph shows the levels of pGBF1 quantified by densitometry using Image J software and normalized to GBF1. **g, h** WT and TBC1D23 KO HepG2 cells were treated with or without 10 mM metformin for 4 h. The cells were stained with an anti-GM130 (Golgi marker) antibody. Representative confocal images (**g**) and quantitation of Golgi elements in cells as treated in (**g**) (**h**, left panel). Scale bar, 10 μm. The ratio between the total area occupied by GM130 and DAPI was also used to indicate Golgi disassembly (**h**, right pannel). Each dot in the figure represents the ratio GM130 area/nucleus area from one cell. Similar results were obtained in three independent experiments (**d–h**). Results are presented as mean ± SD (**d–h**). *P* values were determined by unpaired two-tailed t test (**a, c, d, e**), or by two-way ANOVA, followed by Sidak's test (**f**), or by Scheirer-Ray-Hare Test (**h**). Source data are provided as a Source data file.

To further assess how LKB1 regulated endosome-to-TGN trafficking, we monitored the localization of TGN46, a known cargo of TBC1D23-mediated endosome-to-TGN trafficking[31,35], in the presence or absence of LKB1 CRD. The LKB1 CRD is sufficient to interact with TBC1D23 but lacks the kinase activity, thus precluding the effect of LKB1 kinase activity. LKB1 CRD significantly inhibited the endosome-to-Golgi delivery of TGN46, as determined by quantitative image analysis (Supplementary Fig. 6b, c). Additionally, we tested how glucose starvation and metformin treatment affected the localization of TBC1D23-dependent cargoes. AMPK activation, due to either glucose starvation or metformin treatment (Supplementary Fig. 6d), led to significant inhibition of localization of TGN46 and CI-MPR at Golgi (Supplementary Fig. 6e–h). These data indicate that TBC1D23 has distinct functions in different cellular status: under basal conditions, TBC1D23 functions as an adapter to mediate endosome-to-Golgi trafficking; upon energy stress, TBC1D23 shifts from a mediator of cellular trafficking to a scaffolding protein for LKB1 to regulate Golgi-AMPK activation.

## Golgi-targeted expression of LKB1 partially rescues TBC1D23 deficiency

We and others have previously shown that the knockdown of TBC1D23 impairs neuronal development in zebrafish and mice[32,33,35], and LKB1 is known to regulate various steps of neuronal development through its downstream effectors of AMPK-related kinases, such as BRSK 1/2, NUAK1, and SIK[9,12–16]. We next sought to investigate whether TBC1D23 and LKB1 cooperate in regulating neuronal development in zebrafish. We and others have previously shown that the knockdown of TBC1D23 in zebrafish can mimic many features of PCH patients, including reduced brain size, altered brain structure, impaired mobility, and abnormal neuronal development[32,35]. As TBC1D23 was critical for the development of CaP motor neurons, we tested whether LKB1 played a role in the process. Knockdown of TBC1D23 by injection of a translation blocking morpholino (MO) targeting TBC1D23 led to dramatically-shortened CaP motor neurons in Tg [Hb9: GFP][ml2] transgenic zebrafish. Co-injection of TBC1D23 or LKB1-G WT mRNA could partially restore the length of CaP motor neurons. In contrast, co-injection of LKB1 or LKB1-G D194A failed to increase the neuronal length (Fig. 7a, b). These data suggest that TBC1D23 and LKB1 cooperate in regulating neuronal development.

Injection of TBC1D23 MO significantly reduced the amount of HuC (elavl3), an early marker of panneuronal cells. Neuron loss was particularly evident in the midbrain, cerebellum and hindbrain of morphants, which manifested altered brain morphology relative to control MO embryos. Altogether, TBC1D23 disruption in zebrafish mimics the human phenotype by impairing brain growth and development. It should be noted that zebrafish TBC1D23 defects cause more severe brain development damage compared to human patients characterized by hypoplasia of pons and cerebellum[33,34]. It will be interesting to understand these differences in future studies. Since the

midbrain size can be readily measured in lateral views of zebrafish, we chose the midbrain size to characterize this brain development deficiency, similar to our previous studies. The midbrain size of the MO group is only 40% of that of the control group (Fig. 7c, d). Similar to our previous studies, co-injection of TBC1D23 mRNA could effectively restore the midbrain size (Fig. 7c, d), indicating the direct role of TBC1D23 in brain development[32,35]. Remarkably, co-injection of the Golgi-targeting LKB1 (LKB1-G WT) could partially rescue the midbrain defects, although not as effective as TBC1D23. Importantly, neither wild-type LKB1 nor the Golgi-targeting LKB1 kinase-dead mutant (LKB1-G D194A) could rescue the defects (Fig. 7c, d). Taken together, our data indicate that TBC1D23 regulates neuronal growth and brain development, at least partially, through recruiting LKB1 to Golgi.

## LKB1 promotes neuronal growth and brain development via interacting with TBC1D23

Next, we tested whether LKB1 regulates neuronal growth through its interaction with TBC1D23. In this regard, we noticed that an F354L mutation occurs in the LFa motif of LKB1 and is associated with impaired AMPK activation through an unknown mechanism[51]. The F354L mutation was first reported in patients with the Peutz–Jeghers syndrome (PJS), a rare disease often associated with loss-of-function mutations in LKB1, and is present in at least 17 homozygotes in the gnomAD database[52,53]. Since F354 resides in the LFa motif that mediates the interaction with TBC1D23, we hypothesized that the F354L mutation could impair AMPK activation by decreasing the interaction with TBC1D23. Indeed, both F354L and F354A mutations immunoprecipitated less TBC1D23, relative to LKB1 WT (Fig. 8a). Furthermore, GST pulldown assays indicated that both LKB1 F354L and F354A retained less purified TBC1D23 PH, in comparison with LKB1 WT (Fig. 8b). In agreement with the pull-down experiments, quantitative measurement using ITC indicated that the binding affinity between LKB1 WT and TBC1D23 PH was about three times higher than that of LKB1 F354L and F354A (Fig. 8c).

To define the functional significance of LKB1-TBC1D23 interaction in neuronal development, we used zebrafish as a model. A splice-blocking MO could effectively deplete LKB1 in zebrafish embryos[54] (Supplementary Fig. 7). Consistent with the established role of LKB1 in mammalian neuronal development[9], depletion of LKB1 decreased the length of the CaP motor neurons in zebrafish (Fig. 8d, e). Furthermore, LKB1 depletion also led to reduced midbrain size (Fig. 8f, g), similar to what we observed in the TBC1D23 deficiency zebrafish. Both the motor neuron and midbrain phenotypes were specific, as the mRNA encoding human LKB1 wild-type could rescue these defects (Fig. 8d–g). To test whether LKB1 regulates neuronal development through TBC1D23, we co-injected mRNA encoding human LKB1 F354L or F354A together with LKB1 MO. In contrast with LKB1 WT, neither F354L nor F354A could restore the length of the CaP motor neurons (Fig. 8d, e). Similarly, both LKB1 F354L and F354A mutants could not restore the size of the midbrains (Fig. 8f, g). Remarkably, whereas the Golgi-localized LKB1 could

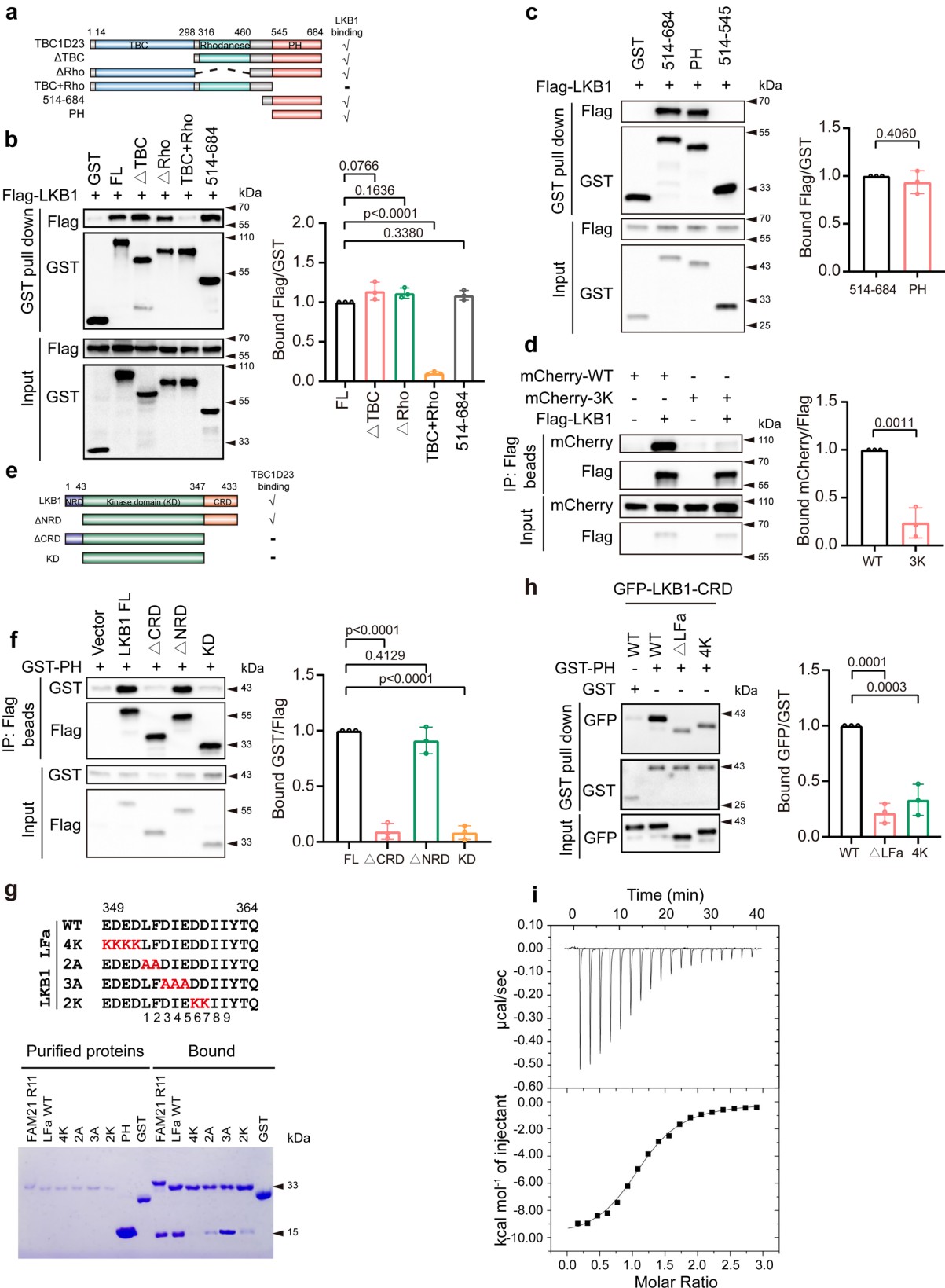

partially rescue TBC1D23 deficiency, TBC1D23 failed to rescue the reduction of axonal length and midbrain size caused by the depletion of LKB1 (Fig. 8d–g). These data suggest that the TBC1D23-LKB1 association is critical for their functions in neuronal growth and brain development. Genetically, TBC1D23 functions upstream of LKB1 to ensure its proper functions in Golgi.

## Discussion

LKB1 is a master upstream kinase that regulates various cellular and physiological processes, including maintaining metabolic homeostasis and controlling neuronal development[1,6,9]. Here, we demonstrate that the direct interaction with TBC1D23 recruits LKB1 to Golgi upon energy stress, leading to Golgi-AMPK activation and subsequent Golgi

**Fig. 4 | The PH domain of TBC1D23 directly interacts with LKB1. a** Schematic diagram of full-length (FL) and various truncations of TBC1D23, and a summary of LKB1 binding determined by GST pull-down. **b, c** HEK293T cells were transfected with indicated constructs for 24 h. Cell lysates were subjected to precipitation with GST beads, and the bound proteins were immunoblotted with indicated antibodies. The graph showed ratios of bound Flag to GST quantified by densitometry using Image J software. **d** HEK293T cells were co-transfected with mCherry-TBC1D23 WT or 3 K and Flag-LKB1 for 24 h, and control cells were co-transfected with mCherry-TBC1D23 WT or 3 K and an empty vector. Cells were harvested and subjected to immunoprecipitation with anti-Flag beads followed by immunoblotting with indicated antibodies. The ratios of bound mCherry to Flag derived from a densitometric analysis of the blot were shown. **e** A schematic diagram of full-length (FL) and truncated mutants of LKB1, and a summary of TBC1D23 binding determined by immunoprecipitation. **f** HEK293T cells co-transfected with LKB1 FL or its truncations (vector as control) and GST-TBC1D23 PH were lysed and incubated with anti-Flag beads, the immunoprecipitates were then examined by immunoblotting. The graph showed ratios of bound GST to Flag quantified by densitometry using Image J software. **g** GST-FAM21-R11, GST-LKB1-LFa WT, mutants, or GST pull-down of purified TBC1D23 PH domain. **h** HEK293T cells were transfected with indicated constructs. 24 h later, cell lysates were precipitated with GST beads and immunoblotted with indicated antibodies. The graph showed ratios of bound GFP to GST quantified by densitometry using Image J software. **i** Isothermal titration calorimetry of an LKB1-LFa motif peptide (GADEDEDLFDIEDDIIYTQ) titrated into TBC1D23 PH domain. Top and bottom panels show raw and integrated heat from injections, respectively. Experiments in (**a**–**i**) were performed in triplicate. Results are presented as mean ± SD. *P* values were determined by one-way ANOVA, followed by followed by Dunnett's test (**b, f, h**), or by unpaired two-tailed t test (**c, d**). Source data are provided as a Source data file.

re-organization. Furthermore, we show that TBC1D23 and LKB1 cooperate to regulate neuronal development, likely through other members of the AMPK subfamily (Fig. 8h). Together, our study establishes that TBC1D23 as a regulator of the LKB1 signaling, emphasizing that the LKB1 signaling has distinct functions and is differentially regulated at different intracellular compartments.

As the master regulator of energy homeostasis, AMPK activity is tightly regulated by many different mechanisms, including allosteric activation, different upstream kinases, the severity of energy stress and post-translational modifications[6,19,55–57]. Subcellular compartments, such as lysosome, mitochondria and ER, have emerged as another critical mechanism to fine-tune AMPK signaling upon specific stimuli[6,55]. Among them, the function and regulation of Lyso-AMPK are best characterized[19,23,41,44,45,58,59]. We note at least three significant distinctions between Lyso-AMPK and Golgi-AMPK. First, different scaffolding proteins. The scaffolding protein AXIN is responsible for the activation of Lyso-AMPK and is recruited to the lysosome by the v-ATPase-Ragulator complex in response to glucose starvation[23,41]. In contrast, we show that TBC1D23 functions as a Golgi-specific scaffold protein for the LKB1-AMPK signaling. Second, distinct substrates and functions. Lysosomal AMPK activation results in the phosphorylation of multiple downstream targets, such as ACC1, Raptor, SREBP1, and TSC2, critical to rewiring metabolic pathways[6,19]. We show that TBC1D23 promotes Golgi-AMPK and facilitates the phosphorylation of GBF1 and Arf1, which are essential for Golgi assembly and cellular trafficking. Third, different AMPK activity dynamics in response to metabolic perturbation. Previous studies using a FRET-based probe reveal that the activation of Golgi-AMPK is much faster than AMPK activation on lysosomes and several other subcellular compartments[29]. Our study may explain the different dynamics, as TBC1D23 is a Golgi-resident protein and tethers LKB1 to Golgi. In contrast, AXIN is not a lysosome-resident protein and requires additional lysosome-proteins to recruit LKB1 on lysosomes.

TBC1D23 is a ubiquitously expressed protein that mutated in the neurological disorder PCH, characterized by abnormal development of the pons and cerebellum[33,34]. We and others have previously elucidated the roles of TBC1D23 in regulating cellular trafficking, and suggested that defective endosome-to-TGN trafficking could be involved in the pathogenesis of TBC1D23-associated PCH[31,32,35]. Here, we provide multiple lines of evidence suggesting that TBC1D23 and LKB1 cooperate to regulate neuronal development, and their defective binding could contribute to the pathogenesis of PCH. First, depletion of TBC1D23 and LKB1 in zebrafish leads to similar phenotypes, including impaired neuronal growth and brain size. Second, a Golgi-targeting LKB1, but not its kinase-dead mutant or LKB1, could rescue TBC1D23 deficiency in zebrafish. Third, LKB1 regulates neuronal development through its interaction with TBC1D23, as the LKB1 WT, but not the F354L or F354A mutants that are defective in binding to TBC1D23, rescues defects caused by LKB1 deficiency in zebrafish. These data suggest that the interaction between TBC1D23 and LKB1, more specifically, TBC1D23-mediated LKB1 activity at Golgi, is essential for neuronal development. Since LKB1 also regulates the development of the nervous system through downstream kinases other than AMPKα1/α2[9], we propose that TBC1D23 regulates the functions of LKB1 in both AMPK-dependent and -independent manners.

Our study also uncovers a mechanism by which cellular trafficking is coupled with nutrient availability. LKB1 and FAM21, a critical regulator of endosomal trafficking[60], bind to the same region and sites of TBC1D23 with similar affinity. Thus, LKB1 and FAM21 bind competitively to TBC1D23. We show that energy stress promotes the interaction between LKB1 and TBC1D23, and inhibits the interaction between FAM21 and TBC1D23, thus suppressing TBC1D23-mediated cellular trafficking. Since trafficking involves in multiple energy-consuming events, including GDP/GTP exchange and actin polymerization, this competitive mechanism enables energy homeostasis by slowing down energy-consuming trafficking events, and by promoting ATP-generating processes via activated AMPK. The dual functions of TBC1D23 could allow TBC1D23-mediated cellular trafficking to be coupled with nutrient availability. It is also plausible that TBC1D23 regulates nutrient-dependent cellular trafficking through additional mechanisms; for instance, Golgi-AMPK activation could serve as the initiation signal for cellular trafficking. Recently, AMPK-mediated GBF1 activation was reported to promote ER-Golgi trafficking[50]. We observe that endosome-to-TGN trafficking was downregulated under stress conditions. Whether and how AMPK activation regulates other trafficking pathways remains to be investigated.

The importance of the Golgi apparatus in neurodevelopment is increasingly clear. Golgi functions as central trafficking and sorting station, center for biosynthesis of glycoproteins and lipids, and also as a signaling hub. Golgi is also involved in the formation and maintenance of neurons' dendritic and axonal morphology. Consistent with these fundamental roles of Golgi, mutations of genes encoding Golgi proteins cause variable and heterogeneous genetic disorders, and many of them affect the nervous system[61]. For instance, 14 ARF1 variants have been identified in affected individuals that displayed symptoms including microcephaly, intellectual disability, seizures and periventricular nodular heterotopia[62–65]. Recently, mutations of Golgi-localized ARF3, a GTPase regulating Golgi dynamics, were identified in patients with developmental disorder impairing nervous system and skeletal formation[66]. Here, we provide additional evidence for the essential roles of Golgi in neuronal development by demonstrating that a disrupted Golgi-LKB1 signaling contributes to the pathogenesis of PCH.

Through utilizing morpholino to control gene expression and rescue experiments, we demonstrated that TBC1D23 and LKB1 cooperate to control neuronal development in zebrafish models. Current morpholino technique is known to induce off-target effects and to have limited duration. CRISPR/Cas9-based technologies, such as the

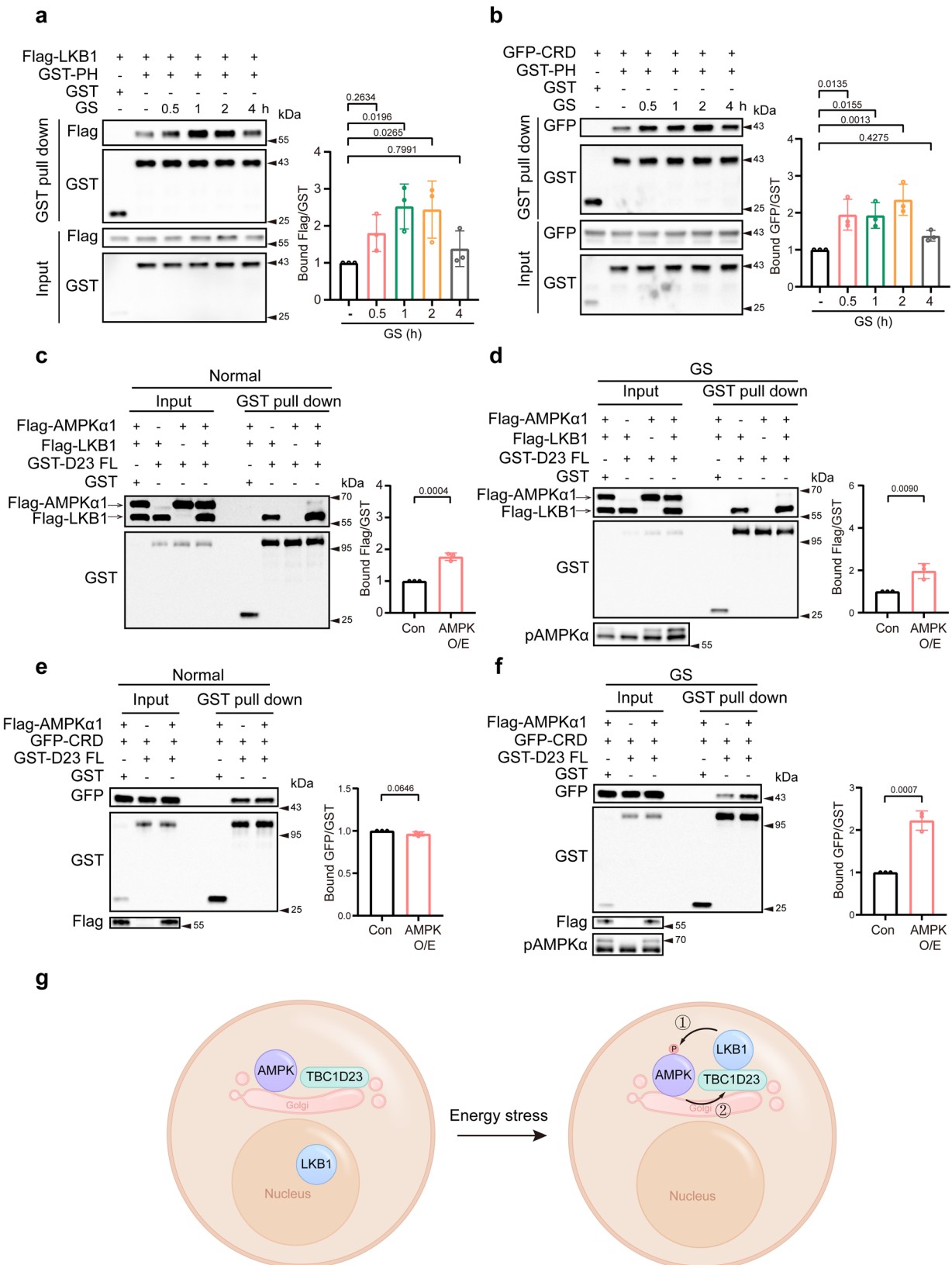

previously reported F0 knockout method[67], have emerged as important alternative approaches to manipulate gene expression in zebrafish. Although we confirmed the specificity of observed phenotypes and excluded the possibility of off-target effects by recue experiments, consistent results from morphants and crispants could further consolidate our conclusions. Furthermore, utilization of additional

models, such as human iPSC-derived neurons, could enhance the strength of our conclusions.

Collectively, our current study reveals the unique function and regulation of LKB1 signaling at Golgi, and unifies the roles of TBC1D23 and LKB1 in regulating neuronal development, which may shed light on understanding the pathogenesis of PCH and other diseases.

**Fig. 5 | The interaction between TBC1D23 and LKB1 is dynamically regulated by AMPK activation. a**, **b** HEK293T cells were transfected with Flag-LKB1 (**a**) or GFP-LKB1 CRD (**b**) and GST-TBC1D23 PH, co-transfection of GST-vector and Flag-LKB1 (**a**) or GFP-LKB1 CRD (**b**) as the negative control. 24 h post-transfection, cell lysates were subjected to glucose starvation for indicated time and precipitation with GST beads, followed by immunoblotting with indicated antibodies. The graph showed ratios of bound Flag to GST (**a**) or GFP to GST (**b**) quantified by densitometry using Image J software. Values for each time point were shown relative to the ratio of control cells without glucose starvation. **c**, **d** HEK293T cells were transfected with indicated constructs. Twenty-four hours later, cells were treated with (**d**) or without (**c**) culture medium deprived of glucose for 2 h before harvest and precipitation with GST beads. Precipitates were resolved by SDS-PAGE and analyzed by western blotting. The graph showed ratios of bound Flag to GST quantified by densitometry using Image J software. **e**, **f** HEK293T cells were co-transfected with indicated constructs. Twenty-four hours later, cells were collected and subjected to pulldown

with GST beads (**e**) or glucose-starvation for 2 h before harvest and pulldown with GST beads. Precipitates were resolved by SDS-PAGE and analyzed by western blotting with indicated antibodies. The graph showed ratios of bound GFP to GST quantified by densitometry using Image J software. $n = 3$ independent experiments (**a**–**f**). Results are presented as mean ± SD. $P$ values were determined one-way ANOVA, followed by Dunnett's test (**a**, **b**), or by unpaired two-tailed $t$ test (**c**–**f**). Con indicates co-transfection of Flag-LKB1 (**c**) or GFP-CRD (**e**) and GST-TBC1D23 FL. AMPK O/E indicates co-transfection of Flag-AMPKα1, Flag-LKB1 (**d**) or GFP-CRD (**f**), and GST-TBC1D23 FL. **g** A simplified model depicts the mechanism underlying the dynamic interaction between TBC1D23 and LKB1. Energy stress stimulates the TBC1D23/LKB1 interaction, which promotes the recruitment of LKB1 to the Golgi, which in turn promotes phosphorylation and activation of AMPK (①), and activated AMPK further facilitates the interaction between TBC1D23 and LKB1 (②). Source data are provided as a Source data file.

## Methods

### Plasmids

PCR-amplified full-length TBC1D23 (1–684) was cloned into both pEBG and pmCherry-N1 vectors. PCR-amplified Flag-LKB1 and LKB1 (1–433) were cloned into pcDNA3.1+ vector and pEGFP-N1, respectively. For the expression of TBC1D23 PH in vitro, TBC1D23 PH (545–684) was cloned into pGEX4T1 vector. For the expression of LKB1 CRD mutants in vitro, LKB1 CRD (347–433) was cloned into pMAL vector. All TBC1D23 and LKB1 mutants were generated through site-directed mutagenesis (MCLAB). plvx-LKB1-HA (MIAOLING BIOLOGY, P12233). A GGSGGSGGS linker followed by GIANTIN (3131–3259) was attached to the C-terminus of LKB1 to generate Flag-LKB1-GIANTIN in pcDNA3.1+. Golgi-ABKAR (organelle-targeting sequence: MGNLKSVAQEPGPP CGLGLGLG LGLCGKQGPA, N-terminus) and Mito-ABKAR (organelle-targeting sequence: MAIQLRSLFP LALPGMLALLGWWWFFSRKKA, N-terminus) were created by Takanari Inoue lab and Jin Zhang lab (Addgene: # 61507, #61509). Lyso-ExRai-AMPKAR (organelle-targeting sequence: full length of lysosome-associated membrane protein 1 (LAMP1), N-terminus) was created by Jin Zhang lab (Addgene: # 192449).

### Reagents and antibodies

Reagents used for this study were as follows: DMEM (glucose free) (Gibco, 11966-025), EBSS (Sigma, E2888), Metformin (Selleck), AICAR (MCE, HY-13417), A23187 (MCE, HY-N6687), Glutathione Beads (SmartLifesciences, SA008100), ANTI-FLAG Affinity Gel (Bimake, B23102), GFP beads (LABLEAD, PGA025), CCCP (Selleck, S6494), DAPI (Servicebio, G1012), protease inhibitor cocktail (Selleck, B14001), and phosphatase inhibitor cocktail (Selleck, B15001).

Antibodies used in this study were as follows: rabbit polyclonal anti-TBC1D23 (Proteintech, 17002-1-AP, Western blot (WB) 1:1000), rabbit anti-LKB1(Cell Signaling Technology, 3047, WB 1:1000), rabbit anti-FAM21 (donated by Dr. Daniel D. Billadeauh, WB 1:1000), rabbit anti-AMPKα (Cell Signaling Technology, 2532, WB 1:1000), rabbit anti-phospho-AMPKα-Thr172 (Cell Signaling Technology, 2535, WB 1:1000), mouse anti-phospho-CaMKII-Thr286 (abcam, ab171095, WB 1:1000), rabbit anti-Flag (Proteintech, 20543-1-AP, WB 1:2000), mouse anti-Flag (Sigma-Aldrich, F1804, WB 1:2000, IF 1:300), rabbit anti-golgin-97 (Proteintech, 12640-1-AP, IF 1:200, WB 1:1000), mouse anti-LAMP1 (abcam, ab289548, WB 1:1000), mouse anti-TOM20 (santa cruz, sc-17764, WB 1:500), rat anti-HA (Roche, 11867423001,WB 1:2000), rabbit anti-GST (Proteintech, 10000-0-AP, WB 1:2000), mouse anti-GFP (Proteintech, 66002-1-Ig, WB 1:2000), rabbit anti-GBF1 (Proteintech, 25183-1-AP, WB 1:1000), rabbit anti-phospho-GBF1-Thr1337 (Immuno-Biological lab, 28065, WB 5 μg/mL), rabbit anti-GAPDH (Proteintech, 10494-1-AP, WB 1:2000), rabbit anti-beta actin (ABclonal, AC026, WB 1:2000), rabbit anti-mCherry (Proteintech, 26765-1-AP, WB 1:2000), rabbit anti-Phospho-(Ser/Thr) (Cell Signaling Technology, 9631, WB

1:1000), rabbit anti-GM130 (Abcam, 52649, IF 1:300), mouse anti-GM130 (BD, 610822, IF 1:300), rabbit anti-ZFPL1 (Invitrogen, PA5-53254, IF 1:300), mouse anti-TGN46 (Abcam, 50595, IF 1:300) and mouse anti-CIMPR (Bio-Rad, MCA2048, IF 1:300). Goat anti-Rabbit IgG Secondary Antibody HRP conjugated (SAB, L3012-2), Goat anti-Mouse IgG Secondary Antibody HRP conjugated (SAB, L3032-2). HRP-conjugated Affinipure Goat anti-Rat IgG(H + L) (Proteintech, SA00001-15). Alexa-labeled secondary antibodies were from Invitrogen (1:2000).

### Cell culture, transfection, and immunofluorescence

Cell lines HEK293T (ATCC), HeLa (ATCC), A549 (ATCC), NIH 3T3 (ATCC) and HepG2 (Jennio Biotech) were cultured in Dulbecco's modified Eagle's medium (DMEM, Gibco) supplemented with 10% fetal bovine serum (NEWZERUM) and 1% penicillin/streptomycin (Hyclone) at 37 °C with 5% $CO_2$. Transient transfection of plasmids in HEK293T cells was performed using polyethylenimine (PEI, Sigma) according to the manufacturer's protocol. HeLa and HepG2 cells were transiently transfected using Lipofectamine 3000 (Thermo Fisher, L3000015) according to the manufacturer's instructions. For RNA interference, siNC or siTBC1D23 duplexes were transfected into HUVEC cells using Lipofectamine RNAiMAX (Invitrogen) according to the manufacturer's instructions. 48 h post transfection, cells were subjected to the indicated treatments and collected for immunoblotting analysis. For immunofluorescence, HepG2 cells were fixed using 4% paraformaldehyde after a 24 h transfection period and permeabilized in 0.1% (v/v) Triton X-100 in PBS, then blocked with 5% BSA in PBS for 1 h. Cells were then incubated with indicated primary antibodies overnight at 4 °C. After washing three times with PBS, cells were incubated with the appropriate Alexa-labeled secondary antibodies at room temperature for 1 h. Cells were mounted, and confocal images were acquired using an Olympus FV-3000 confocal microscope (100× oil objective, NA = 1.45). X, Y scans were acquired at 8 μs/pixel with 1024 × 1024 resolution (sequential mode: line). Image acquisition and analysis within a certain set of experiment were performed with the same parameters. For Golgi assembly, Golgi elements (stained with GM130) analysis is performed by manual counting. And we use blind image analysis for counting Golgi fragments. We also analyzed Golgi assembly by determing the ratio between the total area occupied by GM130 and the nucleus (DAPI) using ImageJ. For co-localization analyses, Pearson's coefficients and Manders' coefficients were calculated using JACoP plug-in of ImageJ. For each condition, the number of cells analyzed were indicated in the figure legends or Supplementary Table 1.

### Immunoprecipitation

HEK293T cells were washed with PBS and lysed using NETN buffer (100 mM NaCl, 1 mM EDTA, 20 mM Tris−HCl, pH 8.0, 0.5% NP-40)

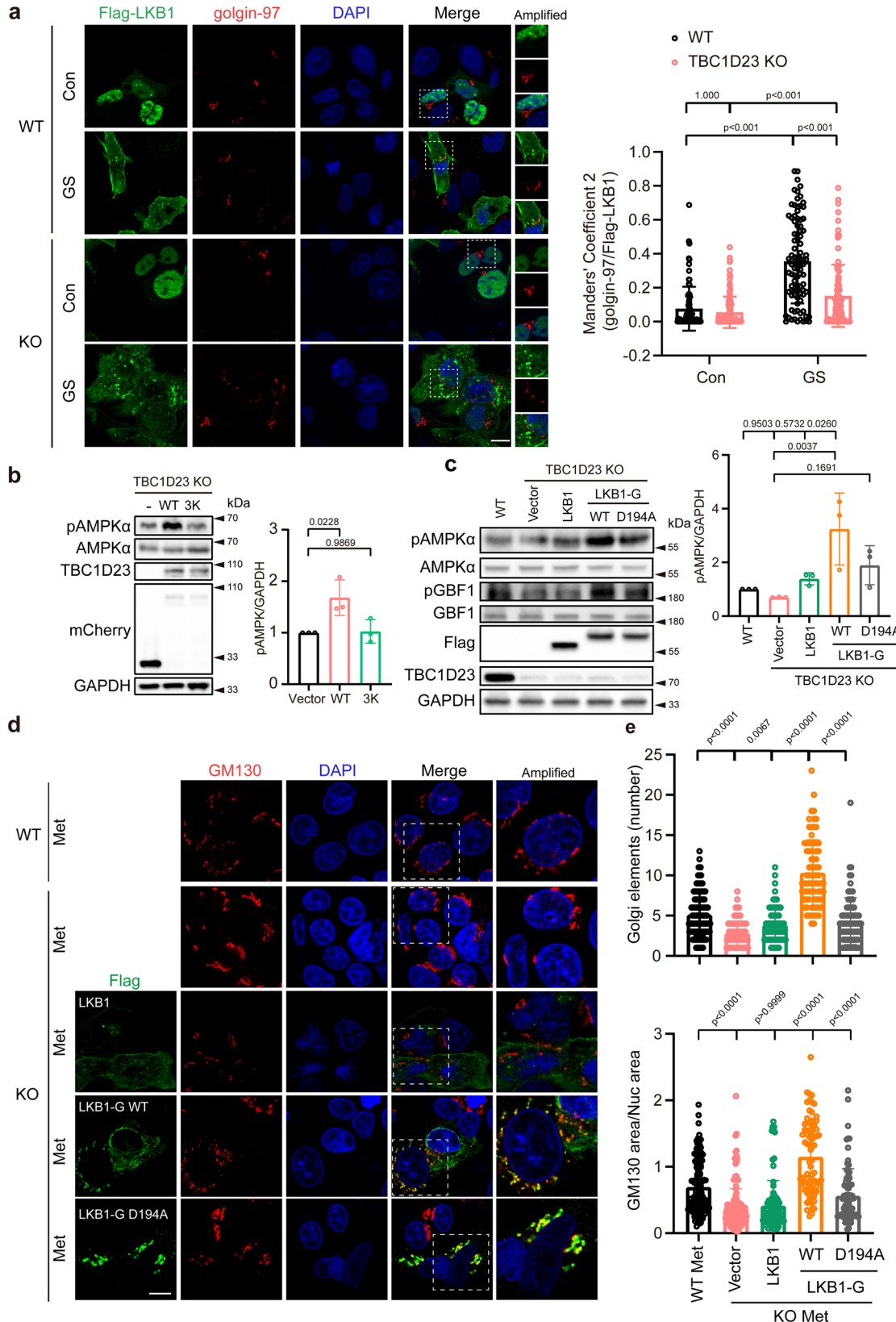

containing protease inhibitor and phosphatase inhibitors for 30 min at 4 °C. Lysates were centrifuged at 12,000 × g for 20 min at 4 °C, the supernatants were subjected to immunoprecipitation using anti-Flag Affinity Gel (overnight at 4 °C) or GFP beads (4 °C, 2 h). Precipitated immunocomplexes were washed three times with NETN buffer and boiled in 2× sample buffer. Samples were separated using SDS-PAGE and detected by immunoblot with indicated antibodies.

**Immunoblotting**

Cells were harvested in 1×SDS sample buffer (62.5 mM Tris-HCl, pH 6.8, 2% SDS, 20 mM DTT and 10% glycerol) and boiled at 95 °C for 10 min.

**Fig. 6 | TBC1D23 and LKB1 cooperate to promote Golgi-AMPK activation. a** WT and TBC1D23 KO HepG2 cells were transiently transfected with Flag-LKB1 for 24 h and glucose starved for 2 h. Cells were stained with antibodies against Flag and golgin-97. Scale bar, 10 µm. Mander's coefficient 2 (golgin-97 overlapping with Flag-LKB1) were graphed as mean ± SD. Each dot represents Mander's coefficient from one cell. WT Con: $n = 81$; WT GS: $n = 92$; KO Con: $n = 116$; KO GS: $n = 98$. **b** TBC1D23 KO HEK293T cells were transiently transfected with mCherry-vector, mCherry-TBC1D23 WT or mCherry-TBC1D23 3 K mutant for 24 h. Cells were treated with a medium deprived of glucose for 2 h before harvest. Whole-cell lysates were analyzed by immunoblotting. The graph shows the levels of pAMPK quantified by densitometry using Image J software and normalized to GAPDH. **c** TBC1D23 KO HEK293T cells were transiently transfected with indicated constructs for 24 h, WT HEK293T cells as the control. Cells were treated with medium deprived of glucose for 2 h before harvest and immunoblotted with indicated antibodies. The graph shows the levels of pAMPK quantified by densitometry using Image J software and normalized to GAPDH. **d** Confocal micrographs of WT or TBC1D23 KO HepG2 cells expressing the indicated constructs were treated with metformin (Met, 10 mM, 4 h) and co-stained with antibodies against Flag and GM130. Scale bar, 10 µm. **e** Bar graph represents the quantitation of the Golgi disassembly. Each dot in the figure represents the number of Golgi elements from one cell (upper pannel); the ratio between the total area occupied by GM130 and the nucleus (DAPI) was also used to indicate Golgi disassembly (lower pannel). WT: $n = 142$; KO: $n = 165$; KO + LKB1: $n = 88$; KO + LKB1-G WT: $n = 79$; KO + LKB1-G D194A: $n = 85$. n indicates pooling cells from one replicate (**a**, **d**). Similar results were obtained in three independent experiments (**a**–**d**). Results are presented as mean ± SD (**a**–**d**), and $p$ values were determined by Scheirer-Ray-Hare Test (**a**) or by one-way ANOVA, followed by Dunnett's test (**b**, **c**), or by Kruskal–Wallis test (**e**). Source data are provided as a Source data file.

Proteins were separated by SDS-PAGE and blotted onto a polyvinylidene fluoride (PVDF) membrane (Millipore). After blocking with 5% (w/v) milk in TBST (0.1% (v/v) Tween-20 in TBS) for 1 h, membranes were probed using specific primary antibodies overnight at 4 °C, followed by exposure to HRP-conjugated secondary antibodies. The bands were visualized by Chemiluminescence (Millipore). Source data are provided as a Source data file.

### Generation of CRISPR knockout cell lines
The single guide RNA targeting TBC1D23 (sgTBC1D23) (5′-CTGCCAACGTCGAGCGGCGA-3′ targeting human TBC1D23, TBC1D23 KO in HepG2 and HEK293T; 5′-GGCTCTGAACGTTGCAGGGAA-3′ targeting mouse TBC1D23, TBC1D23 KO in 3T3) was inserted into the construct lentiCRISPRv2. Constructs encoding Cas9 and gRNA were co-transfected with viral packaging plasmids (pMD2.G: psPAX2:v2/sgTBC1D23 = 1:2:4) into HEK293T cells using PEI. After 48 h, the viral supernatants were collected and filtered. The supernatant was used to infect HepG2 cells with 4 µg/mL polybrene (YEASEN). Cells were cultured in selection medium containing 5 µg/mL puromycin (BBI Life Sciences, A610593) for two generations, then trypsinized and diluted into 96-well plates. Single colonies were expanded and analyzed by immunoblotting. For TBC1D23 KO in HEK293T, 3T3, HeLa and A549, cells were infected with virus encoding hSpCas9 or hSpCas9&sgTBC1D23, and WT and TBC1D23 KO cell pool (5 µg/mL puromycin selection for two generations) were used for subsequent experiments.

### siRNA knockdown of TBC1D23
HUVEC cells at approximately 30% confluence were transfected with the siRNA pool and non-targeting control siRNA using Lipofectamine RNAiMax Reagent (Invitrogen), according to the manufacturer's instruction. Cells were subjected to the indicated treatments and harvested 72 h after transfection. Lysates were analyzed by western blotting.

siControl: 5′-UUCUCCGAACGUGUCACGUTT-3′
siTBC1D23-1: 5′-AGAGAUCCUUCAAGCGAAU-3′
siTBC1D23-2: 5′-GGGAGAUUGUUUCACGGAA-3′
siTBC1D23-3: 5′-GCGCUGAAUUCUGUAGUUA-3′
siTBC1D23-4: 5′-CCGUUAAUGUCAGGGAAAA-3′

### Cell proliferation assay
For cell proliferation assay, WT and TBC1D23 KO HepG2 cells were seeded into 96-well culture plates (1000 cells per well) in triplicate. After incubation for the indicated time, cells were evaluated using the CCK8 assay kit (Servicebio). The absorbance was measured at 570 nm. For the colony formation assay, equal numbers of WT or TBC1D23 KO HepG2 cells were seeded into 6 cm dishes (500 cells per dish) in triplicate. After 14 days, the colonies were fixed with 4% PFA for 15 min, followed by staining with 0.5% crystal violet for 15 min, then washed with ddH$_2$O.

### FRET
Cells were plated on confocal dishes (35 mm) and transfected with Mito-ABKAR or Golgi-ABKAR. AMPK activity at mitochondria or Golgi apparatus was measured 24 h later. To examine AMPK activity, the cells were maintained in DMEM (Gibco) containing 25 mM glucose for basal conditions or without glucose for glucose starvation prior to image acquisition. The images were acquired using a Leica DMI6000B total internal reflection fluorescence microscope (Leica), equipped with mercury lamp as laser power. The images were acquired using BP420/10 excitation filter, a 440/520 dichroic mirror and two emission filters (BP472/30 for cyan fluorescent protein (CFP) and BP542/27 for YFP). Throughout the experiment, images were collected every 5 s and lasted for 30 s. The pseudocolor images of FRET/CFP ratio showing the FRET response were obtained using FRET Wizard of LAS X (Leica).

### Measurement of lysosomal AMPK activation using Lyso-ExRai-AMPKAR
Cells were grown on 24 well glass-bottom plate (Cellvis) and transfected with Lyso-ExRai-AMPKAR. After 20–24 h, cells were incubated with or without glucose-free medium for another 2 h. For glucose starvation, we used glucose-free DMEM (Gibco, 11966-025) supplemented with 10% FBS. Images were then acquired by Olympus FV-3000 confocal microscope (100× oil objective, NA = 1.45). X, Y scans were acquired at 8 µs/pixel with 1024 × 1024 resolution (sequential mode: line). Dual GFP excitation-ratio imaging was performed (405 nm excitation: PMT voltage = 618 V, 7% laser transmissivity, detection wavelength is 465–565 nm; 488 nm excitation: PMT voltage = 503 V, 3.8% laser transmissivity, detection wavelength is 465–565 nm), and images of each group were captured within 15 min. The fluorescence emission with 488 nm excitation and 405 nm excitation was measured using ImageJ software, and the ratio of these two fluorescence intensities (excitation 488/excitation 405) was calculated for further analysis.

### Isolation of Golgi fractions
Minute Golgi Apparatus Enrichment Kit (GO-037, Invent Biotechnologies) was used to enrich Golgi fractions, as previously described[68]. Briefly, 10 cm dish HeLa cells stably transfected plvx-neo or plvx-LKB1-HA grown to 90–95% confluence were collected, washed once with cold PBS and suspended in 550 µL buffer A containing protease inhibitor and phosphatase inhibitors. Vortex the tube vigorously for 20–30 s. The cell suspension was then loaded into the filter cartridge, centrifuged at 16,000 × $g$ for 30 s and then for 5 min at 5000 × $g$ at 4 °C without removing the filter. The supernatants were transferred to a fresh 1.5 mL tube and centrifuged for 30 min at 16,000 × $g$ at 4 °C. After centrifugation, 400 µL supernatant was transferred to a fresh 1.5 mL tube, and equal volume of buffer B was added to the tube. The tube was incubated on ice for 15 min and then centrifuged at 8000 × $g$ for 5 min at 4 °C. The pellet was resuspended in precooled 200 µL buffer A by pipetting up

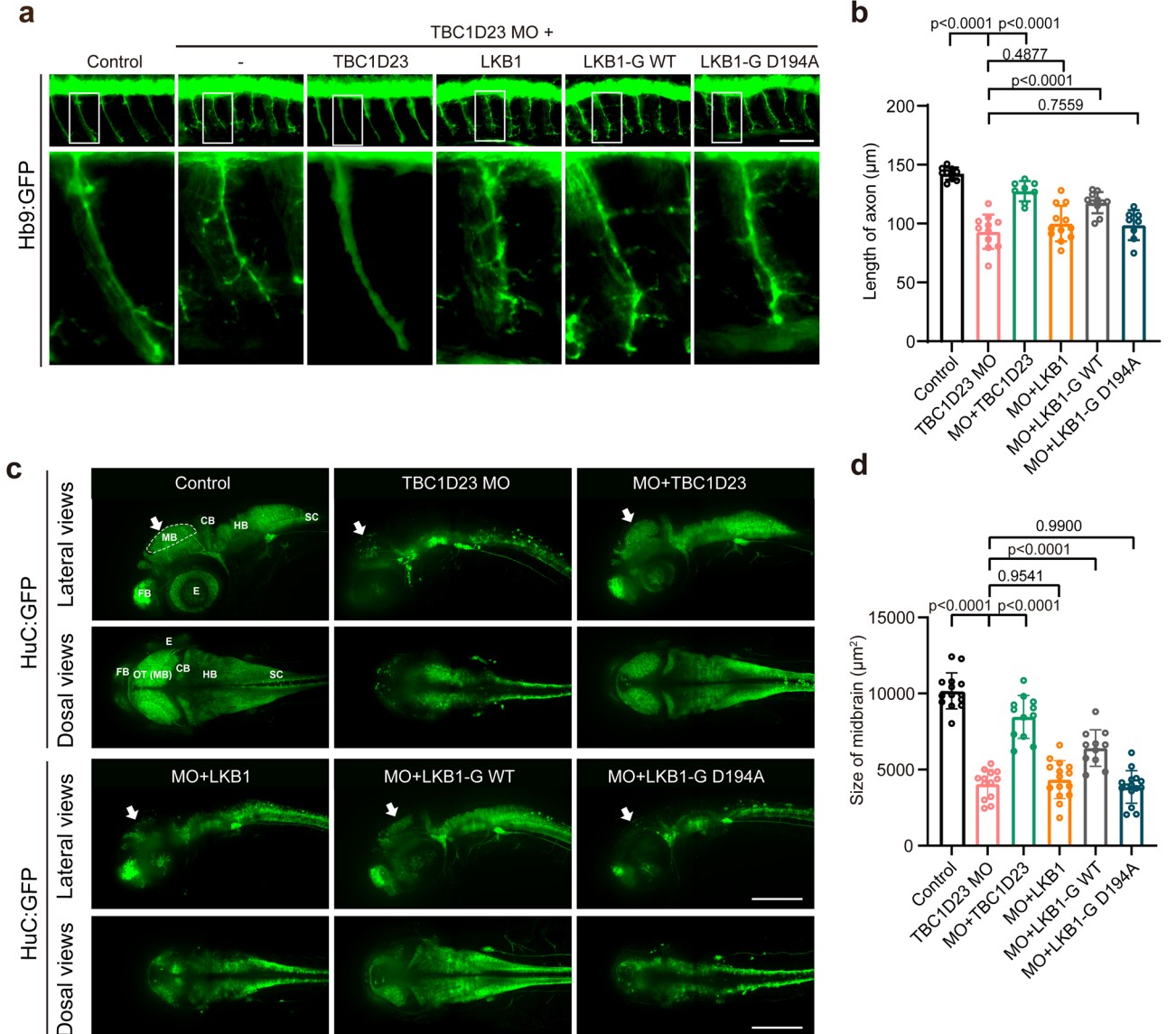

**Fig. 7 | Golgi-targeted expression of LKB1 partially rescues TBC1D23 deficiency in zebrafish. a** Morphology of CaP axons in Tg[Hb9:GFP]ml2 transgenic zebrafish at 48 hpf with z-axis scanning, scale bar, 50 μm. Lateral views (upper) and enlarged views of rectangles (lower) are shown. **b** Statistical analysis of the length of CaP axons in zebrafish at 48 hpf. Approximately 8 to 12 Tg[Hb9:GFP] transgenic zebrafish embryos from each group were measured for 3 CaP axons at approximately the third to fifth CaP axon located on the yolk extension, with each point representing the average axon length of one embryo. Control, control morpholino (MO) injection (*n* = 10 embryos); TBC1D23-MO, TBC1D23-MO injection (*n* = 11 embryos); TBC1D23-MO + TBC1D23, TBC1D23-MO, and human TBC1D23 WT mRNA co-injection (*n* = 8 embryos); TBC1D23-MO + LKB1, TBC1D23-MO and human LKB1 WT mRNA co-injection (*n* = 12 embryos); TBC1D23-MO + LKB1-G WT, TBC1D23-MO, and LKB1-G WT mRNA co-injection (*n* = 11 embryos); TBC1D23-MO + LKB1-G D194A, TBC1D23-MO, and LKB1-G D194A mRNA co-injection (*n* = 8 embryos). All injections

are performed at the one-cell stage of development. **c** HuC (green) expression in Tg[HuC:GFP] transgenic zebrafish at 48 hpf. The overview fluorescent images are presented as maximum intensity z-projections. White arrowheads point to the midbrain. FB, forebrain; MB, midbrain; OT, optic tectum; CB, cerebellum; HB, hindbrain; SC, spinal cord; E, eye. Top, lateral view; bottom, dorsal view, scale bar, 200 μm. **d** The size of zebrafish midbrain at 48 hpf. The size of the midbrain was measured from the lateral view, and 11 to 15 embryos from each group were used for comparison. Con: *n* = 14 embryos; TBC1D23 MO: *n* = 13 embryos; MO + TBC1D23: n = 12 embryos; MO + LKB1: *n* = 15 embryos; MO + LKB1-G WT: *n* = 11 embryos; MO + LKB1-G D194A: *n* = 15 embryos. Results are presented as mean ± SD. *P* values were calculated using one-way ANOVA, Dunnett's multiple comparisons test (**b**, **d**). Experiments were repeated 3 times. Source data are provided as a Source data file.

and down for 50 times, and then centrifuged at 8000 × *g* for 5 min. The supernatant was transferred to a fresh 1.5 mL tube and mixed with 100 μL cold buffer C by vortexing. The tube was incubated on ice for 20 min and centrifuged at 8000 × *g* for 10 min at 4 °C. The pellets containing Golgi fractions were resuspended in 200 μL 1×SDS sample buffer, boiled for 10 min at 98 °C. The samples were separated by SDS-PAGE and detected by immunoblot with indicated antibodies.

## Isothermal titration calorimetry (ITC)

ITC experiments were performed as previously described[69,70]. Prior to experimentation, proteins and peptides were dialyzed into the ITC buffer (100 mM Tris pH 8.0, 500 mM NaCl, 5% v/v Glycerol). LKB1-LFa motif peptides (1.0 mM) were titrated into TBC1D23 PH domain (50 μM) at 25 °C. Each experiment was performed at least three times. Data were analyzed using Origin 7.0 software.

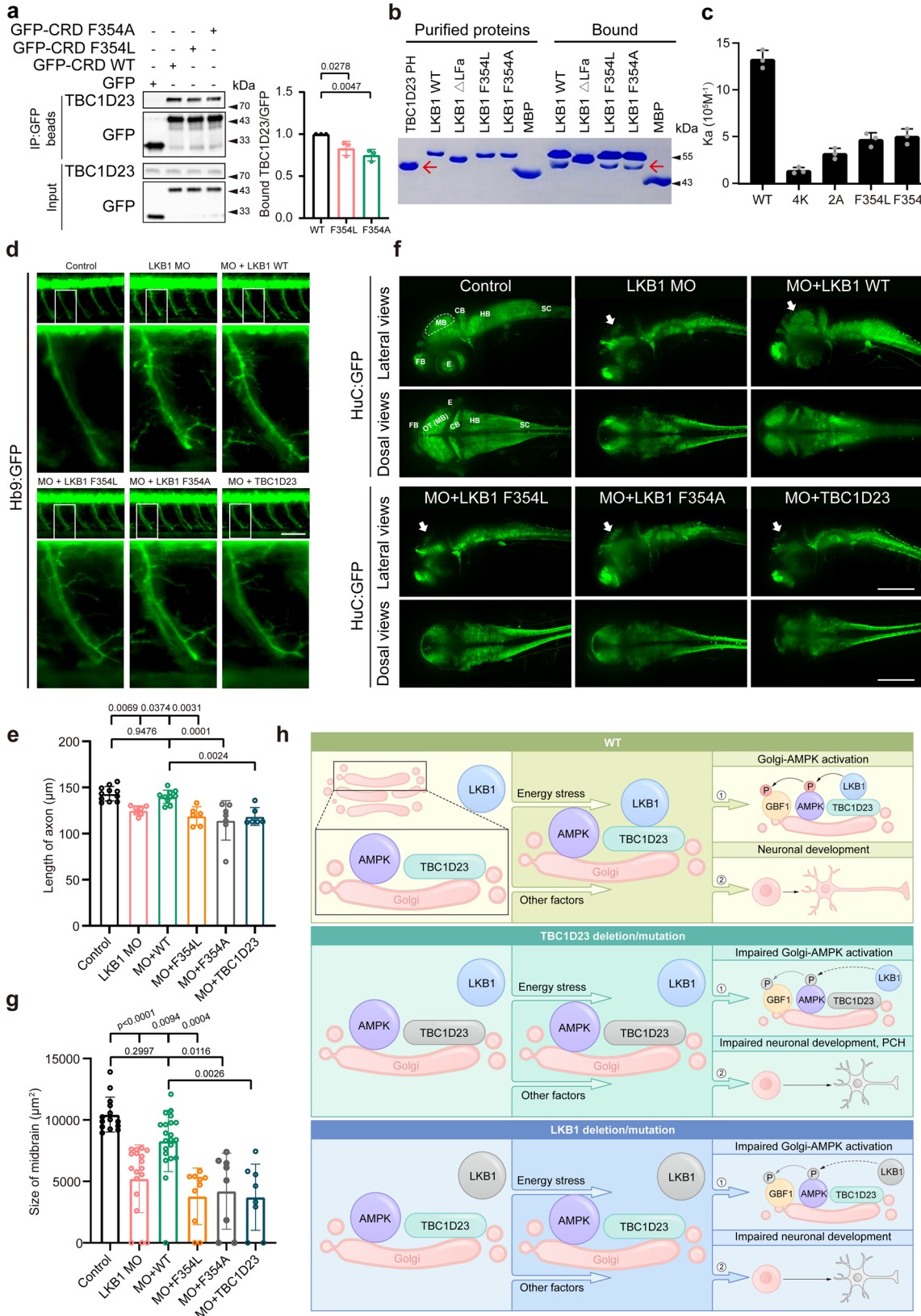

The sequences of LKB1 LFa peptide used in this study are shown as below:

WT: GADEDEDLFDIEDDIIYTQD
4K: GADKKKKLFDIEDDIIYTQD
2A: GADEDEDAADIEDDIIYTQD
F354L: GADEDEDLLDIEDDIIYTQD

F354A: GADEDEDLADIEDDIIYTQD

## Pull-down experiments

For in vitro binding assays using recombinant proteins, the mixture contained 20 μg of GST or GST-tagged protein, and 500 μg of bait proteins. The proteins were mixed with glutathione-Sepharose beads

**Fig. 8 | Interaction with TBC1D23 is required for LKB1 to promote neuronal development and brain growth. a** HEK293T cells were transfected with indicated constructs, and then immunoprecipitated with GFP beads and probed with indicated antibodies. The graph showed ratios of bound TBC1D23 to GFP quantified by densitometry. **b** MBP-LKB1 CRD-LFa WT, mutants, or MBP pull-down of purified GST-TBC1D23 PH domain. Red arrow indicates GST-TBC1D23 PH. **c** Affinity between TBC1D23 PH domain and LKB1-LFa WT or mutants, determined by ITC. Mean association constants (Ka) are shown, with error bars indicating SD from 3 independent titrations. **d** Morphology of CaP axons in Tg[Hb9:GFP]$^{ml2}$ transgenic zebrafish at 48 hpf with z-axis scanning, scale bar, 50 μm. Lateral views (upper) and enlarged views of rectangles (lower) are shown. **e** Statistical analysis of the length of CaP axons in zebrafish at 48 hpf. Approximately 6 to 11 Tg[Hb9:GFP] transgenic zebrafish embryos from each group were measured for 3 CaP axons at approximately the third to fifth CaP axon located on the yolk extension, with each point representing the average axon length of one embryo. **f** HuC (green) expression in Tg[HuC:GFP] transgenic zebrafish at 48 hpf. Top, lateral view; bottom, dorsal view. scale bar, 200 μm. **g** The size of zebrafish midbrain at 48 hpf. The overview fluorescent images are presented as maximum intensity z-projections. White arrowheads point to the midbrain. FB, forebrain; MB, midbrain; OT, optic tectum; CB, cerebellum; HB, hindbrain; SC, spinal cord; E, eye. The size of the midbrain was measured from the lateral view, and 8 to 21 embryos from each group were used for comparison. Results are presented as mean ± SD (**a–g**), and $p$ values were calculated using one-way ANOVA, Dunnett's multiple comparisons test (**a** and **e**) or by Kruskal–Wallis test (**g**). Experiments were repeated 3 times (**a–g**). **h** Simplified model depicting how TBC1D23 and LKB1 cooperate to promote AMPK activation at Golgi and to regulate neuronal development. TBC1D23 mutation/deletion or LKB1 mutation/deletion impairs the recruitment of LKB1 to Golgi, leading to compromised Golgi-AMPK activation, aberrant neuronal development, and eventually pontocerebellar hypoplasia (PCH). Source data are provided as a Source data file.

---

(20 μL) in 0.5 mL of pull-down buffer (50 mM Tris pH 8.0, 500 mM NaCl, 0.05% Triton X-100). After binding at 4 °C for 1 h, beads were washed three times with 1 mL of pull-down buffer. Samples were separated on 12% SDS-PAGE, and visualized by Coomassie staining. The pull-down experiment for MBP-tagged proteins follows the same procedure as for GST-tagged proteins.

For pull-down assays using cell lysates, HEK293T cells transfected with indicated plasmids were washed with PBS and lysed in NETN buffer (100 mM NaCl, 1 mM EDTA, 20 mM Tris–HCl, pH 8.0, 0.5% NP-40) containing protease inhibitor and phosphatase inhibitors for 30 min at 4 °C. Lysates were centrifuged at 12,000 × g for 20 min at 4 °C, the supernatants were precipitated using GST beads at 4 °C for 3 h. Beads were then washed three times with NETN buffer and boiled in 2 × sample buffer. Beads-bound samples were separated by SDS-PAGE and detected by immunoblot with indicated antibodies.

### Identification of TBC1D23-binding proteins by mass spectrometry
For precipitation of GST-tagged TBC1D23, two 10-cm dishes of HEK293T cells at approximately 80% confluence were transfected with plasmids encoding GST-TBC1D23 or the control GST-vector using PEI according to the manufacturer's instructions. Cells were harvested and lysed in NETN buffer containing protease inhibitor and phosphatase inhibitors for 30 min at 4 °C. Cell lysates were centrifuged, and the supernatants were incubated with 50 μL GST beads for 2 h with rotation at 4 °C. The beads were washed with NETN buffer, and bound proteins were eluted using 80 μL GSH (10 mM GSH, 20 mM Tris–HCl, pH 8.0), followed by the addition of 20 μL 5×SDS sample buffer. The eluates were separated by SDS-PAGE, and subjected to immunoblotting and mass spectrometric analysis. For mass spectrometry analysis of TBC1D23-interacting proteins, the eluates were loaded on 10% Tris-glycine SDS PAGE and run for 1–2 cm, visualized by Coomassie G-250 staining. The entire gel lane was cut into slices (1 mm³). The slices were destained using 50% ethanol and water and dehydrated in 100% acetonitrile (ACN) until gel particles became opaque white. After drying, the gel was reduced using 5 mM dithiothreitol (56 °C, 1 h), and alkylated with 15 mM iodoacetamide (IAM) in the dark (room temperature, 30 min). The gel was then digested with trypsin (Promega) at 37 °C overnight, and peptides were extracted through a decreasing gradient of trifluoroacetic acid (TFA) and an increasing gradient of ACN[71,72]. After desalination using C18 ZipTip, peptides were analyzed using an EASY-NLC 1000 nano-flow LC instrument coupled to a Q Exactive Plus mass spectrometer.

### Phosphoproteomics
Protein extraction and digestion. WT and TBC1D23 KO HEK293T cells were lysed using 1 mL RIPA buffer (50 mM Tris–HCl, 150 mM NaCl, 1% NP-40, 0.5% sodium deoxycholate, pH 7.5) containing protease inhibitor and phosphatase inhibitors for 30 min at 4 °C. Lysates were centrifuged at 12,000 × g for 20 min at 4 °C, and the protein

concentration of the supernatants was measured using a Bradford assay. 1 mg protein was reduced with 10 mM tris (2-carboxyethyl) phosphine (TCEP) at 56 °C for 1 h and alkylated with 20 mM iodoacetamide for 30 min at room temperature in the dark. The protein was precipitated using methanol/chloroform/water. Pellets were resuspended in 50 mM Triethylammonium bicarbonate (TEAB) and digested with trypsin (1:50) at 37 °C overnight. The trypsin was inactivated by heating at 95 °C for 2 min.

Phosphopeptide enrichment[73]. Desalted peptides were dissolved in loading buffer (85% acetonitrile, 0.1% TFA), and incubated with Fe-NTA-agarose beads (Cube Biotech, 31403-Fe) at 25 °C for 1 h. The beads were washed 4 times with washing buffer (80% acetonitrile, 0.1% TFA). Bound phosphopeptides were eluted with elution buffer (40% acetonitrile, 15% NH$_4$OH). The eluates were acidized with 10% TFA and dried. Phosphopeptide samples were then desalted and subjected to LC-MS/MS analysis.

LC-MS/MS analysis. The enriched phosphopeptides were desalted by ZipTip (Merck Millipore) and subjected to liquid chromatography-tandem MS analysis using an EASY-nLC 1200 system coupled to a Q Exactive HF-X mass spectrometer (Thermo Fisher). Peptides were separated with a 90 min gradient of 12% to 100% buffer B (80% acetonitrile, 0.1% formic acid) at a flow rate of 330 nL/min. For full MS, ions within 350–1800 $m/z$ were acquired by an orbitrap with 60,000 resolution at 200 $m/z$. For MS2 scans, the top 20 precursors were selected with a 1.6 $m/z$ isolation window and fragmented with a stepped normalized collisional energy (NCE) of 25% and 27%. The dynamic exclusion time was 50 s, and the maximum injection time was 64 ms.

MS data searching. Raw MS data were analyzed by MaxQuant (version 1.6) and queried against the Swiss-Prot human protein sequence database (20431 protein sequences, updated on 2019/07/16). The proteolytic enzyme was trypsin/p and at least 2 missed cleavages sites were allowed. Oxidation of methionine, acetylation of the protein N-terminus, and phosphorylation of STY were selected as variable modifications. Cysteine carbamidomethylation was set as a fixed modification. The maximum peptide mass was set to 12,000 Da, and the minimum amino acid length was set to 6. The first search mass tolerance was 20 ppm, and the main search peptide tolerance was 4.5 ppm. "Match between runs" was enabled for label free phosphoproteomics.

### Zebrafish
All zebrafish (Danio rerio) experiments were performed according to standard procedures as previously described[74,75], and both adult fish and embryos were raised at 28.5 °C in Aquatic Ecosystems. The following lines were used in this study: AB strain (wild-type), Tg[HuC:GFP] strain, and Tg[Hb9:GFP]$^{ml2}$ strain. All experimental protocols were proved by the Animal Ethical Committee, West China Hospital of Sichuan University.

## Morpholino and mRNA injections

Two antisense morpholino oligonucleotides (MO) were purchased from Gene Tools. TBC1D23-MO (CTTCCCCTACAGCATCCGCCATTGC) binds to the ATG of the zebrafish TBC1D23 gene[35], and blocks translation. LKB1-MO (ATAAGCATCAGCTCCTACTTCTTG) is designed corresponding to the second exon-intron junction and inhibits splicing, as validated in previous studies[54]. Control MO (CCTCTTACCTCAGTTA-CAATTTATA) is a standard, mismatched control. MO and mRNAs were injected into the yolk and the cell at the one-cell stage. The injection dose for TBC1D23-MO, LKB1-MO, and control MO was 5 ng per embryo, with mRNA injection doses of 100–150 pg per embryo.

## Total RNA isolation and Semi-quantitative RT-PCR

Total RNAs of approximately 40 zebrafish embryos at 48 h post fertilization (hpf) were extracted as previously described[76,77], and were then reverse transcribed (TaKaRa, PrimeScriptTM RT-PCR kit DRR014A) to obtain cDNAs for use as Quantitative Real-time PCR (qPCR) templates. The qPCR assays were performed (Accurate Biology, SYBR Green Premix Pro Taq HS qPCR Kit) using a CFX96 real-time system (Bio-Rad), and the expression level of β-actin in zebrafish embryos was used as an internal control. The primer sequences used were as follows:

LKB1 Forward: GTGAAGGAGATGCTGGACTCGG
LKB1 Reverse: CAGCACGTCCACCAGCTGAATG
ACTIN Forward: GTACCCTGGCATTGCTGAC
ACTIN Reverse: CTGCTTGCTGATCCACATCTG

## Zebrafish fluorescent imaging

Live embryos were anesthetized before microscopic observation and photography. Fluorescent images of CaP axons were captured by ZEISS AXIO Zoom.V16 microscope with z-axis scanning, and analyzed by ZEN 3.1 and Image J 1.52a software. For confocal imaging, live embryos were encased in confocal microscopy dish with a 0.5% Agarose (Low Melting Point). Fluorescent images of the midbrain were acquired by Olympus IX83 P2ZF (30× silicone immersion objective, NA = 1.05. X, Y, Z scans were acquired with 4491 × 2302 resolution and ~1.5 μm z-step size, exposure times ranged between 150.001 to 200.998 ms) and ZEISS AXIO Zoom.V16 microscope (PlanApo Z 1.0×, NA = 0.25. X, Y, Z scans were acquired with 2752 × 2208 resolution, exposure times ranged between 300 to 800 ms). Automatic exposure was used for each zebrafish to capture high quality images. Image analysis within a certain set of experiment were performed with the same parameters. For Z-stacks of zebrafish embryos, deconvolution analysis using TRUSIGHT (cellSens Dimension, Olympus) and maximum intensity projection using MAX-Z (cellSens Dimension, Olympus) were applied to improve the contrast and resolution of raw images.

## Analysis of midbrain size and CaP axon length

Midbrain size and CaP axon length analysis was performed as previously described[35]. In lateral views of Tg[HuC:GFP] transgenic zebrafish, the midbrain size was quantified by ZEN 3.1 software. As for the measurement of CaP axons, we used ZEN 3.1 software to quantify 3 CaP axons per embryo in Tg[Hb9:GFP]^ml2 transgenic zebrafish at approximately the third to fifth CaP axon located on the yolk extension. The mean of these 3 CaP axons presented each embryo was calculated for further statistical analysis.

## Statistics and reproducibility

Statistical analyses were performed using GraphPad Prism 8 Software. No statistical method was used to predetermine sample size. Each dataset was subjected to D'Agostino & Pearson test, Shapiro–Wilk test, or Kolmogorov–Smirnov test for normal distribution when applicable. Statistical analyses of zebrafish experiments were performed using one-way ANOVA, Dunnett's multiple comparisons test. For all cellular experiments, statistical significance of the difference between two groups of normally distributed data was determined using unpaired two-tailed Student's t test. And an unpaired two-tailed Mann–Whitney test was used to determine significance between two groups of data without a normal distribution. And statistical significance of the difference between multiple groups of normally distributed data, an ordinary one-way or two-way ANOVA (if two variables exist) was used, followed by Dunnett's and Sidak's test, respectively, as indicated in the figure legends. Kruskal–Wallis test was used to determine significance between multiple groups of data (one variable) without a normal distribution. And Scheirer-Ray-Hare Test was used to determine significance between multiple groups of data (two variables) without a normal distribution, which were performed using SPSS 22 (IBM). Differences were considered significant when $P < 0.05$. All data were represented as mean ± SD. All statistical specifications for each figure panel were included in Supplementary Table 1.

## Reporting summary

Further information on research design is available in the Nature Portfolio Reporting Summary linked to this article.

## Data availability

The mass spectrometry raw data were deposited to the ProteomeXchange Consortium with dataset identifier IPX0006072000 via the iProx partner repository. All data supporting the findings of this study are available within the article and its supplementary files and from the corresponding author upon request. Source data are provided with this paper.

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

## Acknowledgements

We thank Dr Yiguo Hu (Sichuan University), Dr Fengli Wang (Huazhong University of Science and Technology) for providing plasmids, Dr. Daniel D. Billadeau (Mayo Clinic) for FAM21 antibody. Research in the authors' laboratory is supported by Natural Science Foundation of China (NSFC) grants (#92254302, #81901281, #82171528), National Key Research and Development Program of China (2022YFA1105200, 2021YFA1101800), and National Science Fund for Distinguished Young Scholars (#32125012).

## Author contributions

Y.T. and D.J. conceived the project. Y.T. performed cellular experiments with assistance from X.L., M.T., L.M., Y.H.W., and R.J. Q.Y. and Z.L. carried out zebrafish studies. L.G. and L.D. completed MS analysis. J.W. and T.T. performed FRET assays. Y.W., Y.L., and P.X. helped with discussions. Y.T. and D.J. prepared the paper with input from all authors.

## Competing interests

The authors declare no competing interests.

## Additional information

[1]Key Laboratory of Birth Defects and Related Diseases of Women and Children, Department of Paediatrics, West China Second University Hospital, State Key Laboratory of Biotherapy, Sichuan University, Chengdu 610041, China. [2]State Key Laboratory of Biotherapy, West China Hospital, Sichuan University, Chengdu, China. [3]State Key Laboratory of Reproductive Medicine, Interdisciplinary InnoCenter for Organoids, Institute for Stem Cell and Neural Regeneration, School of Pharmacy, Nanjing Medical University, Nanjing, China. [4]State Key Laboratory of Membrane Biology, Institute of Zoology, Chinese Academy of Sciences, Beijing 100101, China. [5]Beijing Institute for Stem Cell and Regenerative Medicine, Beijing 100101, China. [6]University of Chinese Academy of Sciences, Beijing 100049, China. [7]Binzhou Medical University, Yantai 264003, China. [8]The MOE Key Laboratory of Biosystems Homeostasis & Protection and Zhejiang Provincial Key Laboratory of Cancer Molecular Cell Biology, Life Sciences Institute, Zhejiang University, Hangzhou 310058, China. [9]These authors contributed equally: Yingfeng Tu, Qin Yang, Min Tang. ✉e-mail: jiada@scu.edu.cn

