## [Peer Review File · Nature Communications]

TBC1D23 mediates Golgi-specific LKB1 signalingREVIEWER COMMENTS

Reviewer #1 (Remarks to the Author):

Review of Manuscript NCOMMS-23-08128-T

The manuscript titled “TBC1D23 mediates Golgi-specific LKB1 signaling” by Dr. Jia and colleagues describes a novel interaction with the Golgi-localized protein TBC1D23 and LKB1. The authors present data that argues for TBC1D23 is required for LKB1 signaling at Golgi, where AMPK is activated upon energy stress. Authors argue that this TBC1D23-LKB1 signaling plays an important role the neuronal development. The Authors first performed a mass spec-based analysis of proteins associated with GST-tagged TBC1D23 protein in HEK293 cells, where LKB1 and AMPK alpha were identified as interacting proteins. Further in the manuscript, using a series of truncations of both TBC1D23 and LKB1 the authors identify segments of TBC1D23 and LKB1 that are key for this interaction. Lastly, they demonstrate that ablation of TBC1D23 as well as mutation of TBC1D23 and LKB1 binding sites impairs development of neurons and zebrafish. The experiments are well described and logical and the studies in zebrafish add a compelling in vivo component to the manuscript. The weaknesses in the paper lie in their biochemical analysis, which at times is not as strong and will need to be address (see below). Overall, this study is well-written, appropriately referenced and clearly argues that while much is known about localization and regulation of the LKB1-AMPK signaling pathway at organelles such as the lysosomes, little is known about localization and/or function at the site of the Golgi.

Points to be addressed:

Figure 1f,g and Figure 2d: it seems that knocking down TBC1D23 basally reduces pAMPK levels and AMPK activity – does knockdown of TBC1D23 change AMP/ATP ratio in a cell?

Figure 1k: it would be helpful to show phosphorylation of calcium-dependent protein as a control for 30 min A23187 treatment.

Figure 3g: 4hr treatment with 10mM metformin led to dispersal of GM130, a marker of Golgi. Can authors comment why in Figure 2a, glucose starvation does not lead to dispersal of GM130? Both glucose starvation and metformin treatment lead to AMPK phosphorylation.

Figure 5c,d: please add labels for FLAG-tagged bands (AMPK, LKB1).

Figure 5c-f: based on the nuclear localization of LKB1 under basal conditions, it is surprising that LKB1 and TCB1D23 would interact under basal conditions. Basal conditions would also imply non-active AMPK. Can authors comment?

Figure 5e,f: last lanes in Input and GST-pull down don't have the same components (GFP-CRD is missing from GST-pull down lane but present in input lane) – please clarify components of the pull down lanes.

Figure 8a: it would be helpful to show a lighter exposure of IPed TCB1D23, it is hard to tell if there is significantly more TCB1D23 precipitated with WT compared to mutant LKB1 CRD.

Figure 8b: please clarify if all LKB1 constructs are MBP-tagged. Why is band with the red arrow the correct one, and not the band above? Western blot with GST for all lanes would help clarify this, since band with red arrow runs higher compared to TCB1D23PH.

Figure 8f: it would be helpful to outline where midbrain is localized in each picture (for non-zebrafish experts).

Reviewer #2 (Remarks to the Author):

The paper by Yingfeng Tu et al. describes a new mechanism of providing Golgi-based LKB1 signaling. Under energy stress, this leads to Golgi-specific activation of AMP-activated protein kinase (AMPK), the key cellular energy sensor and regulator. Important progress has been made during the last decade to characterize subcellular compartmentation of AMPK signaling, which helps to understand the complex cell signaling by this protein kinase. While specific signaling of the AMPK-upstream kinase LKB and AMPK itself at lysosomes and mitochondria have been described, the present paper adds a novel layer by identifying an LKB1-AMPK axis localized at Golgi membranes and phosphorylating locally Golgi proteins. The authors identify TBC1D23 as a scaffolding protein for LKB1 which provides the Golgi-localization of this signaling axis. They further provide evidence that this localized signaling is also linked to a neuro-developmental disorder, pontocerebellar hypoplasia (PCH), in which TBC1D23 is frequently mutated. However, in this case not AMPK but likely another AMPK-related kinase is the LKB1 substrate. As the authors show, such TBC1D23-supported LKB1- signaling is involved in correct axonal growth of neurons. Thus, the study not only provides a new part of the puzzling LKB1-AMPK signaling network, but also a potentially important disease link that could be exploited for new therapeutical strategies.

The study is experimentally sound and provides a complete characterization of Golgi-localized TBC1D23/LKB1 interaction and signaling. The only limitation may be the almost exclusive use of cellular overexpression systems, but likely the Golgi-localized pool of endogenous proteins is very limited. Anyway, the authors show that LKB1 forced to localize at Golgi can rescue TBC1D23 knock-down. Also the use of a Golgi-targeted, genetically encoded AMPK-sensor should be sufficient to evidence the presence of endogenous AMPK signaling at the Golgi. The paper is very well written, and figures are

mostly well designed. The paper certainly is significant for the field, and I would have only some suggestions for revision and improvement.

Signaling via AMPK or other AMPK-related kinases -

The study shows two scenarios of Golgi-located LKB1 signaling, with energy stress triggering the AMPK pathway, while for effects in brain development no evidence for an implication of AMPK is provided. As the authors suggest in the discussion (p.20), such neuronal effects are likely mediated by one of the AMPK-related kinases that are mentioned in the introduction (p.3) and were already linked to neuronal diseases. This latter scenario should be mentioned also in the Results section, so the reader is not astonished that the study does no longer address AMPK in the second part of this section (p.16 ff). Also concerning this issue, did the authors find any hint to an AMPK-related kinase in the pulldown-MS study (p.6) or the phosphopeptide-MS study in presence/absence of TBC1D23 (p.9)? If this is the case, it should be mentioned.

Use of different cell lines -

While most experiments on energy stress (glucose starvation) or metformin activation use HEK293T cells, initial observations were done with several cell lines, e.g. in Figure 1: HEK193T for (a-e), HUVEC for (f), HepG2 for (g-j), HeLa for (k)? While for the use of HeLa cells an explanation is given (naturally devoid of LKB1), the rationale for the other lines is unclear.

Use of overexpressing cell lines -

Localization and interaction experiments are based on cell lines overexpressing the proteins of interest. Did the authors try to detect endogenous TBC1D23 and LKB1 in respect to interaction between them or localization at Golgi?

TBC1D23 knock-out -

The different KO approaches should be better described in the methods section, including the difference between "KO" clones and "sgTBC1D23" vs "v2".

High affinity vs. dynamic interaction

The authors mention an affinity of 800 nM for the interaction between LKB1 (LFA motif) and TBC1D23 (PH domain), even surviving 500 mM NaCl (p.12). On the other hand, under glucose starvation, they describe a very dynamic (and thus reversible) interaction between both partners. This seems to be contradictory.

ABKAR and lysosome AMPK -

Is there a specific reason why, after using Golgi-ABKAR and mito-ABKAR sensors for localized AMPK activity, the authors did not use lyso-ABKAR for the lysosome localization? Also, even if published elsewhere, it would be useful to mention the ABKAR targeting tags in the methods section.

FAM21/LKB1 competition -

The authors nicely describe a competitive mechanism for TBC1D23 binding of either FAM21 (favoring cellular trafficking) or LKB1 (favoring compensations upon energy stress via AMPK). This would correspond to a switch from an ATP-consuming pathway (trafficking) to ATP-saving (most AMPK targets). Authors may elaborate further on this point (e.g. is it known how much ATP may be used in this kind of trafficking?).

Improvement of figures -

Some figures could be better legible if some abbreviations would be harmonized and explained in the legend (even if this introduces some redundancy).

Fig.1: (g,h) KO-5 and KO-6 could be in addition labeled with "TBC1D23"; (k) sgTBC1D23 is not explained in the main text (see above).

Fig.2: While TBC1D23 KO reduces AMPK signaling in Fig.1f,g,h, this seems not to be the case in Fig. 2f. Is this another KO clone, or is there any other explanation?

Fig.3: The color code in parts (a), (b) and (c) could be uniform. Also, the red and pink colors are too close to be distinguished easily. Also, part (g) could be enlarged.

Fig.4: Part (g) could mention that it shows LKB1 mutants.

Fig.6: Abbreviation "LKB1-G" should be explained in the legend.

Fig.7: Abbreviation "MO" should be explained in the legend.

Fig.8: in part (c), affinities are given as K_a (in 1/M), while elsewhere in the ms K_d (in nM) is used (p.12). For better comparison, uniform use of either K_a or K_d is suggested.

Spelling mistake -

Line 359: "restore" instead of "restores"

Reviewer #3 (Remarks to the Author):

In this study, Tu et al. investigate mechanisms regulating the LKB1-AMPK axis at the Golgi body and their functional importance in the context of neuronal development and disease. TBC1D23 is identified as the critical, Golgi-localized component that scaffolds LKB1's interaction with AMPK to enhance AMPK's activation at the Golgi upon energetic stress. The authors use a combination of biochemical assays, immunofluorescence, and fluorescent biosensor imaging to test their hypotheses. This model is then functionally tested in human iPSC-derived neurons and in zebrafish in the context of pontocerebellar hyperplasia, a neurodevelopmental disease. These studies indicate that LKB1's interaction with TBC1D23 is critical for LKB1's capacity to promote neuronal growth and brain development.

This work describes novel findings that enhance mechanistic and functional understanding of AMPK signaling at the Golgi body regulated by LKB1 and TBC1D23. However, there are several major and minor comments that must be addressed.

Major points:

1. Conceptual story of Golgi-specific AMPK versus whole-cell AMPK: The authors should clarify if their data indicates this site has a global "whole cell" effect on AMPK activity or if AMPK is locally activated at the Golgi. This seems to be a conflicting concept at several points in the paper. For example, in the first section, the main assertion is that "TBC1D23 interacts with LKB1 to promote AMPK activation in response to energy stress" but then this is reduced in the next section to "TBC1D23 specifically regulates Golgi-AMPK activation".
2. Experiment Reproducibility: The authors do not identify the number of biologically independent replications of experiments. Three independent replications per condition is the minimum expectation and this should be identified in figure captions. Please identify the number of independent repeats for Figures 1i, 1j, 2d, 2e etc.
3. Blot reproduction and quantification: In conjunction with major comment 2 where all blots should be repeated on at least three separate occasions, blots that are used to make quantitative comparisons need to be quantified using densitometry of repeated blots (Fig. 1f, 1g, 1h, 1k, 2f, 3d, 3e, 3f, 4b, 4c, 4d, 5, 6b, 6c, 8 etc). Indeed, several blots show questionable effect upon inspection. In Figure 4h, "disrupting" seems to be a stronger conclusion than shown by the data – it appears that interaction was only weakened. In Supplementary Figure 6a, the shRNA does not appear highly effective for shTBC1D23, which is a critical component of the human iPSC neuron investigations. Showing blot repeats and quantifications will enhance this study.
4. AMPK biosensor experiments: The authors unrealistically depict how the biosensor ABKAR works, making it appear as a kind of glucose starvation sensor, when its true design involves subunits between the CFP and YFP that detect phosphorylation specifically from AMPK – this should be corrected in Figure 2c. The y-axis of ABKAR plots in Figures 2d and 2e of "AMPK Activity" should denote what image channels are being used ie. AMPK Activity (ECFP/FRET ratio) and if any normalization is occurring. Furthermore, it is unclear why the authors used Golgi-ABKAR and Mito-ABKAR, but then switched to lysosomal acidification in Figures 2g and 2h as a quantification of subcellular AMPK activity when there is a readily available Lyso-ABKAR sensor. Therefore, the use of lysosomal acidification by Lysosensor to

quantify AMPK activity is not well rationalized. The use of Lyso-ABKAR would enhance this submission as it is a more specific readout of AMPK activity at the lysosome.

5. Golgi assembly analysis: The authors quantify Golgi fragmentation on several occasions in this study (Fig. 3h, 6e), but do not describe the quantification method. If this analysis is performed by manual counting, the authors should either (a) use blinding to enhance the rigor of quantification or (b) use unbiased image analysis for counting Golgi fragments.

6. TBC1D23-LKB1 interaction mutant localization: In Figure 4, the authors use several mutant forms of TBC1D23 and LKB1 to investigate where they interact, but do not verify that changes in interaction are not related to mutations altering the localization of TBC1D23 or LKB1. This is an important consideration within a series of hypotheses that focus highly on Golgi localization. At minimum, verifying the TBC1D23 mutant TBC+Rho and the LKB1 mutant Δ CRD do not have significantly different localization from respective WT versions would be needed.

Minor points:

1. Colocalization analysis: The authors use colocalization analysis on several occasions in this study, but do not identify which specific colocalization method and settings are being used in ImageJ (Fig. 2a, 2b, 6a). Moreover, using Pearson's Correlation Coefficient for a colocalization metric is not sufficient in this context comparing broadly distributed proteins with several downstream targets throughout the cell with Golgi-specific proteins. Instead, Mander's Overlap Coefficient would provide readers with a better understanding of the fractional localization of proteins of interest with the Golgi body. Pearson's Correlation Coefficient appears more reasonable in supplementary figures where the predominant localization of both proteins of interest are Golgi-localized.

2. Localization language: For topological correctness, the authors should ensure they are using terms such as "at the Golgi" or "on the Golgi" for outer surface Golgi localizations versus "in the Golgi" for Golgi lumen localization.

Reviewer #4 (Remarks to the Author):

The study identifies TBC1D23 protein as a novel regulator of LKB1 signaling in Golgi, acting via Golgi-AMPK activation and also discusses certain functions and regulatory mechanisms of LKB1 signaling in various cellular compartments. My questions and comments related to proteomics analysis are listed below:

1. Line 114: Figure 1(a) and 1(b) do not show any mass spectrometry data, why are these listed as done using mass spectrometric analysis?
2. According to Figure 1(c), LKB1 has only one unique peptide. That is not enough to claim its presence, could total number of peptides or PSMs be listed in the table as well?

3. Mass spectrometry methods: cite appropriate references for in-gel digestion and phosphopeptide enrichment.

4. When referring to phosphorylation analysis in the Results section, please clarify that it was done using mass spectrometry. It is mentioned in the Methods sections but not in the rest of the article, especially since it is such an important aspect of TBC1D23 pathway within Golgi.

Reviewer #5 (Remarks to the Author):

Tu and colleagues report on a new regulatory mechanism mediated by interaction of TBC1D23 and LKB1 at the Golgi to control AMPK activation under energy stress and potentially controlling trafficking. Multiple complementary experimental approaches and model systems were used.

The work is potentially of a broad interest, in principle, as it is dealing with a new circuit controlling energy stress response via Golgi. Besides the mechanistic insights, interest also results from its link to human disease (pontocerebellar hypoplasia, PCH). There are however some issues that should be taken into consideration and that require extensive revision.

This reviewer identified major concerns in data presentation, quantification, and assessment that weaken the validity of the interpretations and conclusions, and the suitability of the manuscript to the journal. Key Springer Nature standard requirements are not met in the submitted work and should be carefully resolved by the authors.

Generally, the paper would benefit from a more precise introduction to the importance of AMPK activation and its modality in health and disease, imaging parameters are not provided, and images shown are often not sufficient for the claims, control experiments are performed on different cell types without clear explanation, methods are poorly/barely described overall, hindering reproducibility. Statistical analysis is somewhat arbitrary. N, presence of replicated experiments as well as quantification and statistical assessment are not always provided for cell experiments. In addition, the role of organelles and Golgi in neurodevelopmental diseases (for instance ARF1 is only mentioned in parts of the results....etc..) should be addressed in the Discussion, to permit a broader evaluation of the results in the field.

I discuss some of points that should be addressed below.

Major points:

-Different KD and KO approaches, different cell types: authors seem to switch between cells silenced with siTBC1D23 and cells in which a KO was obtained by CRISPR-Cas9. Given the difference in KD ability, the use of one or the other approach should be addressed and discussed for each experiment. Same for the type of cells used.

-In fig.2, co-localization between LBK1 and Golgi marker: the development of super-resolution techniques has made it clear that co-localization analysis performed from standard microscopy are quite prone to errors and should be approached with caution. In these experiments, LBK1 fluorescence signal appear quite saturated (pixels saturation), which is one of the main issues leading to exaggerated and false results especially with Pearson 'correlation analysis. Authors should provide additional examples of images to see whether signal saturation is indeed an issue of the whole population of cells analyzed, if confirmed then this should be addressed for all the co-localization experiments. In addition, given a number of tools/algorithms available and the varying parameters to threshold the images beforehand authors should explain how the analysis was performed. I also suggest performing mild deconvolution on non-saturated images and it would be beneficial to examine other parameters such as Meanders'coefficients. Similar comment for Fig.6a,b and others. Here also again more cells or a larger field of view should also be provided;

-Line 333, Suppl.Fig.5: authors claim to analyze TGN46 trafficking; however, they did not perform any dynamic analysis, nor do they image cargo movement. Authors should just confine their claims, stating that they assess TGN46 localization at Golgi, where the cargo is known to be transport to. The impaired trafficking, claimed uniquely by the problematic Pearson's correlation (see above) and a static picture is only a speculation at this point. Same for CI-MPR. Authors should discuss other possible explanations for the observed reduced localization of TGN46 at Golgi;

-The choice of the statistical analysis seems arbitrary. Authors appear to choose between t-test and analysis of variance without consistency. All the zebrafish experiments with more than two groups within one experiment are correctly assessed by ANOVA (F-test + post-hoc), while some of the cell-culture experiments with the same settings are analyzed via t-test. This approach even without any post-hoc correction leads to amplification of the type I error. The probability that the observed phenomenon is obtained by chance instead of underlying a biological meaning is therefore increasing. Such arbitrary choice in statistical assessment is poor data analysis quality and makes it hard to judge the conclusions overall. It should also be considered whether a 2-way ANOVA should be used in some occasions, for instance for the experimental set up in Fig. 1i where two variables exist: time and genetic condition. Therefore, authors should re-assess all the statistical analyses in the manuscript and clearly explain the choice made for the tests used. A List with all the experimental setups and tests, including analysis of normality would be ideal. This would make the claims of the study solid;

-Very often the number of replicates (for examples for WB and IP) is not given, as well as the quantification. Please authors should always provide the number of replicates for each experiment as requested and also the quantification including mean and SEM and statistical assessment;

-In general more cells/large fields of view should be shown for each experiment, methods for image acquisition and analysis should be better described. For FRET imaging the method used to do imaging (sensitized emission?) wavelengths used and laser power as well as ratiometric analysis should be described;

-In the FRET experiment again, When providing LBK1 in Hela cells (LBK1-/-) why authors do not show the FRET signal increase correlating to AMPK activation?

-Quantification of the images in the WB should also been shown (Fig.S2). Why do authors choose AICAR instead of GS stimulus as opposed to the experiment with TBC1D23 KO cells they want to directly compare to? This should be explained. Also, the number of replicates for all these experiments is not clear (not given). This is a general comment;

-In Fig.3g/h and Fig6e it is not clear how “Golgi elements” were counted. Please specify, as the term “elements” is vague. Did the authors count large and small objects observed in the image? The occurrence of GM130 patches different in size might underly different patterns of fragmentation. More examples of cells should be provided, only 2 or 3 cells are shown now;

-From the images shown, Met seems to have an intermediate effect on Golgi disassembly in KO cells, as also the statistics (should be an ANOVA, see below) would suggest and not what the authors claim, or what do they mean that the disassembly was “impaired”? In Fig. 6d authors should show more cells and higher magnification, so far the rescue in Golgi fragmentation obtained by LBK1-G is hard to judge as only 1 cell is shown. Also, for the condition co-transfected with LBK1 GM130 seems fainter (why?);

-The microscopy images showing zebrafish brain in the transgenic fish are of poor quality, with intense signal bleed from planes other of the one in focus. Authors should specify whether these are single plane images of z-stacks. Probably an epifluorescence microscope was used. Also, they should state the parameters used for each condition/fish. Did the same laser intensity was used? The authors should at least indicate different brain sub-domains for general audience not familiar with zebrafish brain. Clearly, a major involvement of the whole brain is seen in the morphants. Why do authors choose to report only midbrain measurements? Is this correlated to the PCH? This should be explained. What is the evidence for cerebellum itself, or hindbrain beyond r2 and other regions? In case regions others than the midbrain did not show any statistically significant difference this should be reported, and the specific involvement in midbrain should be discussed;

-Fig.6, recruitment of LBK1 to the Golgi is impaired in TBC1D23 KO cells: authors claim that this and the activation of AMPK is rescued by TBC1D23, however microscopy evidence for this is not presented and a different cell type is used (HEK instead of HepG2), likely with a different KO/KD approach (sg instead of KO?). This is likely true also in other points of the manuscript. I find the nomenclature here very confusing. Authors should address the choice of using different cells and different approaches when demonstrating a rescue, this is no immediate neither probably correct;

-Neuronal defects: Fig.7 and S6, authors claim to use iPSC-derived neurons and to test the efficacy of different ShRNA to silence TBC1D23. However, in Fig.S6a they show western blot performed on HEK cells (?). How does this compare to the effect in iPSC cells? This is completely hidden within the text and only appears to an attentive reader. Please revise. Similarly to the “si”, “KO” and “sg” nomenclature story, authors should state clearly why they use different systems in different contexts and cannot rely on different cells to validate efficacy of the approach then used in a different cell type;

-Moreover, quantification of the western blot to test sh efficiency is missing. Without quantification and replicates (how many times was the experiment repeated?) it is not possible to judge the % of silencing (i.e. GAPDH seems higher in the control). Please show mean and SEM in the quantification;

-Primary branching in Fig.7 and S6: authors claim that a decreased primary branching is observed in cells depleted of TBC1D23. First, control cells (without Sh) are not shown and should be shown directly in main figure. One should be able to judge the “rescue” of the branching comparing directly the conditions + LBK1-G etc to control scenario, statistically;

-Authors claim that neuronal defects observed here confirm previous reports. However, in the Huang et al, PNAS 2019 the authors showed the opposite in terms of branching. Reduced TBC1D23 in zebrafish induced increased branching (not decreased) in the previous report. Why does a similar approach reduce branching now? How is this “confirming” previous reports? Indeed, in the current work authors clearly show representative images of CaP motoneurons with hyperbranching (Fig. 7e) confirming their previous report. Why do iPSC cells show a opposite phenotype? Are those differentiated in motoneurons or what? Authors should discuss this. Instead authors say that the data are “nicely consistent with our results in iPSC..” Branching seems again increased in LBK1 MO model that should be addressed;

-Also, was reduced length of CaP motoneurons reported for TBC1D23 MO injection in zebrafish before? In Huang et al, PNAS 2019 this does not seem the case;

-What are the dots in graphs quantifying CaP length? How many CaP were counted per fish? A nested analysis should be considered if more cells per fish are counted.

Minor points:

-Text: authors should explain better (broader context) why it is interesting to understand the role of LKB1 – AMPK regulation at Golgi? – Please revise Abstract and Intro;

-Methods are not detailed and in their current form do not fit the standards of the journal. For instance, imaging conditions with resolution, step size, speed etc... are not provided. Similarly, the algorithms used and image processing are not described. Articles should provide experimental details to improve reproducibility in the community, authors should amend accordingly all the methods;

-English should be improved in the methods section... e.g. line 906 “images...were taken”, proper terminology and details should be used, such as “x,y scans were acquired atconfocal withresolution etc..”;

-Letters in the figures should follow a logical order (not the case for some figures, see Fig.6);

-To better appreciate the differences in AMPKa regulation in siTBC1D23 and in KO cells authors should add densitometric quantifications for Figure 1f, g;

-In Fig.2 expression of LKB1 pointed (vesicles?) after GS. Could authors comment this before line 266? A higher magnification, not saturated image should be provided (see also major point below concerning Pearson);

-line 243 : “we found that”....authors should explain what they did to obtain the finding to a general audience and describe the figure in details;

-line 251: remove “intriguingly” ;

-“SgTBC1D23” equals “KO” or “si”(siRNA)? Sometimes authors refer to one and sometimes to the other, nomenclature should be consistent, what does“sg” stand for?

-Line 339: "statuses" not sure this is correct. Maybe "status".

-Line 1266: the term "confocal immunofluorescence" is not correct. It is an immunofluorescence and samples 'images were acquired via confocal microscopy. Authors should check the main text for proper terminology;

-Fig6a : "LBK1-Flag" should be written instead of just "Flag";

-Line 350 : "attenuate" = typo;

-Line 374 : remove "nicely";

-In the reporting summary it is claimed that N, degree of freedom etc are provided, but this reviewer failed in finding such info. Exact p- values are not reported.

Point-by-point responses:

We appreciate that the paper was deemed to be conceptual interest by reviewers, and we are also grateful to the reviewers' constructive suggestions that certainly help us to improve greatly the quality of our manuscript. To address the reviewers' concerns, we have performed a number of critical experiments accordingly and made extensive revision in our revised manuscript. Our point-by-point responses are listed below:

Reviewers' Comments:

Reviewer #1:

Remarks to the Author

The manuscript titled "TBC1D23 mediates Golgi-specific LKB1 signaling" by Dr. Jia and colleagues describes a novel interaction with the Golgi-localized protein TBC1D23 and LKB1. The authors present data that argues for TBC1D23 is required for LKB1 signaling at Golgi, where AMPK is activated upon energy stress. Authors argue that this TBC1D23-LKB1 signaling plays an important role the neuronal development. The Authors first performed a mass spec-based analysis of proteins associated with GST-tagged TBC1D23 protein in HEK293 cells, where LKB1 and AMPK alpha were identified as interacting proteins. Further in the manuscript, using a series of truncations of both TBC1D23 and LKB1 the authors identify segments of TBC1D23 and LKB1 that are key for this interaction. Lastly, they demonstrate that ablation of TBC1D23 as well as mutation of TBC1D23 and LKB1 binding sites impairs development of neurons and zebrafish. The experiments are well described and logical and the studies in zebrafish add a compelling in vivo component to the manuscript. The weaknesses in the paper lie in their biochemical analysis, which at times is not as strong and will need to be address (see below). Overall, this study is well-written, appropriately referenced and clearly argues that while much is known about localization and regulation of the LKB1-AMPK signaling pathway at organelles such as the lysosomes, little is known about localization and/or function at the site of the Golgi.

1. Figure 1f,g and Figure 2d: it seems that knocking down TBC1D23 basally reduces pAMPK levels and AMPK activity – does knockdown of TBC1D23 change AMP/ATP ratio in a cell?

Response: We appreciate very much the positive and thoughtful comments from reviewer #1. We measured AMP and ATP levels in WT and TBC1D23 KO cells, and observed that TBC1D23 KO cells showed slightly decreased the AMP/ATP ratio, which was not statistically significant (Graphic 1).

Graphic 1: AMP/ATP ratio in WT and TBC1D23 KO cells (n = 3 for each group). Results are presented as mean ± SD, and p value was determined using an unpaired t test.

2. *Figure 1k: it would be helpful to show phosphorylation of calcium-dependent protein as a control for 30 min A23187 treatment.*

Response: We appreciate the reviewer's thoughtful suggestions. Following the suggestion, we measured phosphorylation of CaMKII at Thr286 (PMID: 8185953) and found that 30 min A23187 treatment effectively increased phosphorylation of CaMKII (new Fig.1k). Consistent with our prediction, TBC1D23 KO did not affect CaMKII phosphorylation.

3. *Figure 3g: 4hr treatment with 10mM metformin led to dispersal of GM130, a marker of Golgi. Can authors comment why in Figure 2a, glucose starvation does not lead to dispersal of GM130? Both glucose starvation and metformin treatment lead to AMPK phosphorylation.*

Response: We observed that both 10 mM metformin and glucose starvation led to dispersal of GM130; however, a lower percentage of GM130 dispersal was observed due to glucose starvation. We suspect that the difference is due to different degree of AMPK phosphorylation as 10 mM metformin resulted in a much stronger AMPK phosphorylation.

3. *Figure 5c,d: please add labels for FLAG-tagged bands (AMPK, LKB1).*

Response: We appreciate the reviewer's suggestions. We have added labels for FLAG-tagged bands.

4. *Figure 5c-f: based on the nuclear localization of LKB1 under basal conditions, it is surprising that LKB1 and TBC1D23 would interact under basal conditions. Basal conditions would also imply non-active AMPK. Can authors comment?*

Response: In the IP experiments showed in Figure 5c-f, we used harsh lysis buffer which led to the disruption of intracellular membrane system. TBC1D23 and LKB1 were then released from its native subcellular localization, permitting their interaction.

6. *Figure 5e,f: last lanes in Input and GST-pull down don't have the same components (GFP-CRD*

is missing from GST-pull down lane but present in input lane) – please clarify components of the pull down lanes.

Response: We apologized for the mistake as we mislabeled last lanes in our original submission. We've corrected the problem in the revised manuscript.

7. Figure 8a: it would be helpful to show a lighter exposure of IPed TCB1D23, it is hard to tell if there is significantly more TCB1D23 precipitated with WT compared to mutant LKB1 CRD.

Response: Thank you for pointing this issue out. We have repeated this experiment and quantified the immunoblotting band intensity in the revised manuscript (new Fig.8a).

8. Figure 8b: please clarify if all LKB1 constructs are MBP-tagged. Why is band with the red arrow the correct one, and not the band above? Western blot with GST for all lanes would help clarify this, since band with red arrow runs higher compared to TCB1D23PH.

Response: All LKB1 constructs are MBP-tagged, and we've also updated Fig.8b by using red arrow to indicate the TCB1D23 PH protein.

9. Figure 8f: it would be helpful to outline where midbrain is localized in each picture (for non-zebrafish experts).

Response: We appreciate the reviewer's suggestions. We have labeled the midbrains in the revised manuscript.

Reviewer #2:

Remarks to the Author

The paper by Yingfeng Tu et al. describes a new mechanism of providing Golgi-based LKB1 signaling. Under energy stress, this leads to Golgi-specific activation of AMP-activated protein kinase (AMPK), the key cellular energy sensor and regulator. Important progress has been made during the last decade to characterize subcellular compartmentation of AMPK signaling, which helps to understand the complex cell signaling by this protein kinase. While specific signaling of the AMPK-upstream kinase LKB and AMPK itself at lysosomes and mitochondria have been described, the present paper adds a novel layer by identifying an LKB1-AMPK axis localized at Golgi membranes and phosphorylating locally Golgi proteins. The authors identify TBC1D23 as a scaffolding protein for LKB1 which provides the Golgi-localization of this signaling axis. They further provide evidence that this localized signaling is also linked to a neuro-developmental disorder; pontocerebellar hypoplasia (PCH), in which TBC1D23 is frequently mutated. However, in this case not AMPK but likely another AMPK-related kinase is the LKB1 substrate. As the authors show, such TBC1D23-supported LKB1- signaling is involved in correct axonal growth of neurons. Thus, the study not only provides a new part of the puzzling LKB1-AMPK signaling network, but also a potentially important disease link that could be exploited for new therapeutical strategies.

The study is experimentally sound and provides a complete characterization of Golgi-localized TBC1D23/LKB1 interaction and signaling. The only limitation may be the almost exclusive use of cellular overexpression systems, but likely the Golgi-localized pool of endogenous proteins is very limited. Anyway, the authors show that LKB1 forced to localize at Golgi can rescue TBC1D23 knock-down. Also the use of a Golgi-targeted, genetically encoded AMPK-sensor should be

sufficient to evidence the presence of endogenous AMPK signaling at the Golgi. The paper is very well written, and figures are mostly well designed. The paper certainly is significant for the field, and I would have only some suggestions for revision and improvement.

1. Signaling via AMPK or other AMPK-related kinases -

The study shows two scenarios of Golgi-located LKB1 signaling, with energy stress triggering the AMPK pathway, while for effects in brain development no evidence for an implication of AMPK is provided. As the authors suggest in the discussion (p.20), such neuronal effects are likely mediated by one of the AMPK-related kinases that are mentioned in the introduction (p.3) and were already linked to neuronal diseases. This latter scenario should be mentioned also in the Results section, so the reader is not astonished that the study does no longer address AMPK in the second part of this section (p.16 ff). Also concerning this issue, did the authors find any hint to an AMPK-related kinase in the pulldown-MS study (p.6) or the phosphopeptide-MS study in presence/absence of TBC1D23 (p.9)? If this is the case, it should be mentioned.

Response: Thanks for your positive and thoughtful comments. Following your suggestion, we have added a sentence ‘the possible roles of AMPK-related kinases in neuronal defects and brain development’ in the results section (Line 346-347). We did not find AMPK-related kinases in our pulldown-MS and phosphopeptide-MS study, which is likely due to the cell lines and stimulation conditions that we used. We performed our MS analysis with HEK293T cells in basal conditions or glucose starvation.

2. Use of different cell lines -

While most experiments on energy stress (glucose starvation) or metformin activation use HEK293T cells, initial observations were done with several cell lines, e.g. in Figure 1: HEK193T for (a-e), HUVEC for (f), HepG2 for (g-j), HeLa for (k)? While for the use of HeLa cells an explanation is given (naturally devoid of LKB1), the rationale for the other lines is unclear.

Response: We performed most of our experiments in HEK293T and HepG2 cells, and used HeLa and HUVEC cells to further prove our points. HEK293T cells are easily transfected; thus, we used HEK293T for MS analysis and pulldown/IP assays. In contrast with HeLa cells, HepG2 cells are LKB1 positive. Thus, we used HepG2 cells for IF and FRET assays. As it was reported that LKB1 is required for activation of AMPK in HUVEC cells (PMID: 18250273), so we also examined effects of TBC1D23 knockdown on AMPK activation in HUVEC cells.

3. Use of overexpressing cell lines -

Localization and interaction experiments are based on cell lines overexpressing the proteins of interest. Did the authors try to detect endogenous TBC1D23 and LKB1 in respect to interaction between them or localization at Golgi?

Response: Thanks a lot for the great suggestions. We tried to multiple antibodies to detect the interaction between endogenous TBC1D23 and LKB1, or their localization at Golgi. Unfortunately, none of them worked out, including: (1) Our antibodies against TBC1D23 and LKB1, both produced in rabbit, were not suitable for endogenous IP, particularly being difficult to distinguish LKB1 (~54 KD) from heavy chains of IgG (~50 KD). (2) A monoclonal LKB1 antibody raised in mouse detected non-specific bands. (3) Commercial LKB1 antibodies did not work well for IF.

We have tried multiple ways to circumvent these problems, and demonstrated that: (1) Endogenous LKB1 and TBC1D23 could be enriched by overexpressed TBC1D23 and LKB1 CRD, respectively (Fig.1d and Fig.8a). (2) TBC1D23 directly interacted with LKB1 (Fig.4g and Fig.8b). (3) Flag-tagged LKB1 was recruited to Golgi and co-localized with golgin-97 in response to energy stress (new Fig.2a and 6a). golgin-97 is a TGN marker and is known to directly interact with TBC1D23 (PMID: 29084197, 32453802).

4. TBC1D23 knock-out -

The different KO approaches should be better described in the methods section, including the difference between “KO” clones and “sgTBC1D23” vs “v2”.

Response: We are sorry for not having explained this more clearly in our original submission. We used crispr-cas9 to obtain TBC1D23 KO cells. “KO” indicated single clone; sgTBC1D23 is an abbreviation for “single guide RNA targeting TBC1D23”, and “sgTBC1D23” in the original submission indicated a pool of cells infected with viruses encoding hSpCas9&sgTBC1D23; “v2” indicated a pool of cells infected with viruses encoding hSpCas9 (negative control for sgTBC1D23). We added a statement for situations when we used TBC1D23 knockout pool cells. We have corrected the issues in the revised manuscript.

5. High affinity vs. dynamic interaction

The authors mention an affinity of 800 nM for the interaction between LKB1 (LFa motif) and TBC1D23 (PH domain), even surviving 500 mM NaCl (p.12). On the other hand, under glucose starvation, they describe a very dynamic (and thus reversible) interaction between both partners. This seems to be contradictory.

Response: Thanks a lot for pointing this issue out. As we demonstrated in Fig.5 a,b and supplementary Fig.3a, the interaction between TBC1D23 and LKB1 is highly dynamic and responds to energy stress. The issue is that we used the LKB1 LFa peptide (16 aa) for affinity measurement, rather than LKB1 FL (433 aa). We suspect that additional regions of LKB1 or other factors could regulate the dynamics in the context of full-length LKB1, for instance, regulating the accessibility of the LFa peptide by TBC1D23. Investigating these factors could be an exciting area of future research.

6. ABKAR and lysosome AMPK -

Is there a specific reason why, after using Golgi-ABKAR and mito-ABKAR sensors for localized AMPK activity, the authors did not use lyso-ABKAR for the lysosome localization? Also, even if published elsewhere, it would be useful to mention the ABKAR targeting tags in the methods section.

Response: Actually we did try to measure the activity of lysosomal AMPK using lyso-ABKAR. However, for unknown reason, this plasmid did not work well in our hands. Following your suggestion, we mentioned ABKAR targeting tags in the methods section (plasmids subsection), and clarified that ABKAR constructs were created by Takanari Inoue lab and Jin Zhang lab (Addgene: # 61507, #61509).

7. FAM21/LKB1 competition -

The authors nicely describe a competitive mechanism for TBC1D23 binding of either FAM21

(favoring cellular trafficking) or LKB1 (favoring compensations upon energy stress via AMPK). This would correspond to a switch from an ATP-consuming pathway (trafficking) to ATP-saving (most AMPK targets). Authors may elaborate further on this point (e.g. is it known how much ATP may be used in this kind of trafficking?).

Response: The reviewer has made good points here. We agree with your opinion, and we have modified discussion in the revised manuscript. Trafficking indeed involves in multiple energy-consuming events, including GDP/GTP exchange and actin polymerization, but we cannot find out the exact number of ATP consumed from published literatures.

8. Improvement of figures -

Some figures could be better legible if some abbreviations would be harmonized and explained in the legend (even if this introduces some redundancy).

Fig.1: (g,h) KO-5 and KO-6 could be in addition labeled with “TBC1D23”; (k) sgTBC1D23 is not explained in the main text (see above).

Response: KO-5 and KO-6 has been corrected as “TBC1D23 KO-5” and “TBC1D23 KO-6”, respectively; For consistent nomenclature, sgTBC1D23 has been corrected as “TBC1D23 KO” (please see our response above).

Fig.2: While TBC1D23 KO reduces AMPK signaling in Fig.1f,g,h, this seems not to be the case in Fig. 2f. Is this another KO clone, or is there any other explanation?

Response: Experiments in Fig.1 f-h and Fig. 2f were performed under different treatments. Fig.1 f-h: glucose starvation; Fig. 2f: low-dose metformin. Whereas glucose starvation is known to induce global AMPK activation, low-dose metformin specifically activates Lyso-AMPK. Together with other data, we conclude that TBC1D23 specifically regulates the activation of Golgi-AMPK, but not Lyso-AMPK.

Fig.3: The color code in parts (a), (b) and (c) could be uniform. Also, the red and pink colors are too close to be distinguished easily. Also, part (g) could be enlarged.

Response: Thanks a lot for the suggestion. We have improved Fig.3 as you suggested (new Fig.3a-c, and g).

Fig.4: Part (g) could mention that is shows LKB1 mutants.

Response: Thanks for your suggestion. We have added “LKB1” in Fig.4g.

Fig.6: Abbreviation “LKB1-G” should be explained in the legend.

Response: “LKB1-G” has been explained in the figure legends in the revised manuscript.

Fig.7: Abbreviation “MO” should be explained in the legend.

Response: “MO” has been explained in the figure legend as you suggested.

Fig.8: in part (c), affinities are given as K_a (in $1/M$), while elsewhere in the ms K_d (in nM) is used (p.12). For better comparison, uniform use of either K_a or K_d is suggested.

Response: Thanks for the suggestion. Affinities are given as K_a in the revised manuscript.

9. Spelling mistake -

Line 359: "restore" instead of "restores"

Response: We apologize for the mistake. We have corrected it.

Reviewer #3:

Remarks to the Author

In this study, Tu et al. investigate mechanisms regulating the LKB1-AMPK axis at the Golgi body and their functional importance in the context of neuronal development and disease. TBC1D23 is identified as the critical, Golgi-localized component that scaffolds LKB1's interaction with AMPK to enhance AMPK's activation at the Golgi upon energetic stress. The authors use a combination of biochemical assays, immunofluorescence, and fluorescent biosensor imaging to test their hypotheses. This model is then functionally tested in human iPSC-derived neurons and in zebrafish in the context of pontocerebellar hyperplasia, a neurodevelopmental disease. These studies indicate that LKB1's interaction with TBC1D23 is critical for LKB1's capacity to promote neuronal growth and brain development.

This work describes novel findings that enhance mechanistic and functional understanding of AMPK signaling at the Golgi body regulated by LKB1 and TBC1D23. However, there are several major and minor comments that must be addressed.

Major points:

1. Conceptual story of Golgi-specific AMPK versus whole-cell AMPK: The authors should clarify if their data indicates this site has a global "whole cell" effect on AMPK activity or if AMPK is locally activated at the Golgi. This seems to be a conflicting concept at several points in the paper. For example, in the first section, the main assertion is that "TBC1D23 interacts with LKB1 to promote AMPK activation in response to energy stress" but then this is reduced in the next section to "TBC1D23 specifically regulates Golgi-AMPK activation".

Response: Our MS analysis indicated LKB1 complex as potential interactors of TBC1D23, and we then confirmed the interaction between TBC1D23 and LKB1. Based on the interaction and the essential role of LKB1 in AMPK activation upon energy stress, we then examined the functional significance of TBC1D23/LKB1 interaction in AMPK activation. It turned out that TBC1D23 deficiency results in defective AMPK activation. Recent studies indicate that AMPK activation occurs at multiple subcellular compartments, such as lysosomes, mitochondria, ER, Golgi and nucleus. This prompted us to investigate which compartmentalized AMPK was regulated by TBC1D23. Consistent with the subcellular localization of TBC1D23 at Golgi, loss of TBC1D23 mainly compromises activation of Golgi-AMPK, while lyso-AMPK and Mito-AMPK activation remains intact in TBC1D23 deficient cells. Hence, we concluded that TBC1D23 specifically regulates Golgi-AMPK activation.

2. Experiment Reproducibility: The authors do not identify the number of biologically independent replications of experiments. Three independent replications per condition is the minimum expectation and this should be identified in figure captions. Please identify the number of independent repeats for Figures 1i, 1j, 2d, 2e etc.

Response: We apologize for not having explained this more clearly in our original submission. All experiments are performed at least in triplicate, and we have added this statement in section entitled

“statistics and reproducibility” in Methods.

3. *Blot reproduction and quantification: In conjunction with major comment 2 where all blots should be repeated on at least three separate occasions, blots that are used to make quantitative comparisons need to be quantified using densitometry of repeated blots (Fig. 1f, 1g, 1h, 1k, 2f, 3d, 3e, 3f, 4b, 4c, 4d, 5, 6b, 6c, 8 etc). Indeed, several blots show questionable effect upon inspection. In Figure 4h, “disrupting” seems to be a stronger conclusion than shown by the data – it appears that interaction was only weakened. In Supplementary Figure 6a, the shRNA does not appear highly effective for shTBC1D23, which is a critical component of the human iPSC neuron investigations. Showing blot repeats and quantifications will enhance this study.*

Response: We are sorry for not having interpreted our experiments (Please see our response above) and data more clearly. The quantitative data for the western is included in our revised manuscript. After quantification, we found that binding of Δ LFa or 4K mutants is reduced by 69% and 48%, respectively (new Fig. 4h). We replaced “The interaction was disrupted....” with “The interaction was impaired...”. It is noteworthy that our pulldown assays indicated that LFa motif is indispensable for LKB1 to interact with TBC1D23, particularly 4K mutation almost abolished the interaction (Fig.4g). In new Supplementary Fig.6a, the quantitative data for the western is included. When 30-40% HEK293T is successfully infected, shRNA-1 exhibited ~25% knockdown efficiency. We confirmed the knockdown efficacy by upregulating amount of virus. As shown in new Supplementary Fig.6b, shRNA-1 effectively depletes the expression of TBC1D23 when ~70-80% HEK293T is successfully infected.

4. *AMPK biosensor experiments: The authors unrealistically depict how the biosensor ABKAR works, making it appear as a kind of glucose starvation sensor, when its true design involves subunits between the CFP and YFP that detect phosphorylation specifically from AMPK – this should be corrected in Figure 2c. The y-axis of ABKAR plots in Figures 2d and 2e of “AMPK Activity” should denote what image channels are being used ie. AMPK Activity (ECFP/FRET ratio) and if any normalization is occurring. Furthermore, it is unclear why the authors used Golgi-ABKAR and Mito-ABKAR, but then switched to lysosomal acidification in Figures 2g and 2h as a quantification of subcellular AMPK activity when there is a readily available Lyso-ABKAR sensor. Therefore, the use of lysosomal acidification by Lysosensor to quantify AMPK activity is not well rationalized. The use of Lyso-ABKAR would enhance this submission as it is a more specific readout of AMPK activity at the lysosome.*

Response: Thank you for your constructive suggestion. We have corrected Fig.2c in the revised manuscript, and the y-axis of new Fig.2d and 2e has been corrected as “AMPK activity (Normalized FRET/CFP)”.

We tried to measure the activity of lysosomal AMPK using lyso-ABKAR. However, this plasmid did not work out for unknown reason. Thus, we had to choose a different approach to measure lyso-AMPK.

5. *Golgi assembly analysis: The authors quantify Golgi fragmentation on several occasions in this study (Fig. 3h, 6e), but do not describe the quantification method. If this analysis is performed by manual counting, the authors should either (a) use blinding to enhance the rigor of quantification*

or (b) use unbiased image analysis for counting Golgi fragments.

Response: We apologize for not making it clear in the original submission. The analysis is performed by manual counting, and we use unbiased image analysis for counting Golgi fragments. We have added a description about Golgi assembly analysis in section entitled “Cell culture, transfection and immunofluorescence” in Methods for this point.

6. *TBC1D23-LKB1 interaction mutant localization:* In Figure 4, the authors use several mutant forms of *TBC1D23* and *LKB1* to investigate where they interact, but do not verify that changes in interaction are not related to mutations altering the localization of *TBC1D23* or *LKB1*. This is an important consideration within a series of hypotheses that focus highly on Golgi localization. At minimum, verifying the *TBC1D23* mutant *TBC+Rho* and the *LKB1* mutant Δ CRD do not have significantly different localization from respective *WT* versions would be needed.

Response: Thanks for your suggestions. Following the suggestion, we generated two constructs: GFP-*TBC1D23* FL and GFP-*TBC1D23* *TBC+Rho*. However, *TBC1D23* *TBC+Rho* mutant displayed altered subcellular localization (**Graphic 2a**), although it was reported that TBC domain of *TBC1D23* interacts with golgin-97 and is required for its Golgi localization. In the original submission, we demonstrated that *TBC1D23* is required for recruitment of *LKB1* to Golgi in response to energy stress. Then, we determined how CRD deletion affect its Golgi localization. We found that Golgi localization of Δ CRD is down-regulated relative to *LKB1* WT after glucose starvation (**Graphic 2b**). Thus, *TBC1D23/LKB1* (CRD) interaction is essential role for Golgi localization of *LKB1*.

Graphic 2: a, Subcellular localization of *TBC1D23* and *TBC1D23* *TBC+Rho*. HepG2 cells were transiently transfected with GFP-*TBC1D23* full length (FL) or GFP-*TBC1D23* *TBC+Rho* for 24 h. Cells were stained with antibody against golgin-97, a Golgi marker. **b,** CRD domain deletion results

in significantly decreased localization of LKB1 at Golgi. HepG2 cells were transiently transfected with Flag-LKB1 or Flag-LKB1 Δ CRD for 24 h and glucose starved for 2 h. Cells were stained with antibodies against Flag and golgin-97. Scale bar, 10 μ m. Colocalization analysis was carried out using Manders' coefficients. Each dot represents Manders' coefficients from one cell. Results are presented as mean \pm SD, and p value was determined using an unpaired t test.

Minor points:

1. *Colocalization analysis: The authors use colocalization analysis on several occasions in this study, but do not identify which specific colocalization method and settings are being used in ImageJ (Fig. 2a, 2b, 6a). Moreover, using Pearson's Correlation Coefficient for a colocalization metric is not sufficient in this context comparing broadly distributed proteins with several downstream targets throughout the cell with Golgi-specific proteins. Instead, Mander's Overlap Coefficient would provide readers with a better understanding of the fractional localization of proteins of interest with the Golgi body. Pearson's Correlation Coefficient appears more reasonable in supplementary figures where the predominant localization of both proteins of interest are Golgi-localized.*

Response: We appreciate the excellent suggestion! We used Manders' coefficients for Colocalization analysis (new Fig. 2a, 2b and 6a) in the revised manuscript.

2. *Localization language: For topological correctness, the authors should ensure they are using terms such as "at the Golgi" or "on the Golgi" for outer surface Golgi localizations versus "in the Golgi" for Golgi lumen localization.*

Response: Thanks a lot for the suggestion. We have updated the text by using "at the Golgi".

Reviewer #4:

Remarks to the Author

The study identifies TBC1D23 protein as a novel regulator of LKB1 signaling in Golgi, acting via Golgi-AMPK activation and also discusses certain functions and regulatory mechanisms of LKB1 signaling in various cellular compartments. My questions and comments related to proteomics analysis are listed below:

1. *Line 114: Figure 1(a) and 1(b) do not show any mass spectrometry data, why are these listed as done using mass spectrometric analysis?*

Response: Thanks for your positive and thoughtful comments. Fig.1a and Fig.1b is the silver staining of the proteins enriched by GST-TBC1D23 or GST-vector (the negative control), and western blot analysis with GST antibody, respectively. Fig. 1a indicated that much more proteins are enriched by GST-TBC1D23 relative to GST-vector, and Fig. 1b demonstrated the correct expression of indicated constructs (GST-vector and GST-TBC1D23).

2. *According to Figure 1(c), LKB1 has only one unique peptide. That is not enough to claim its presence, could total number of peptides or PSMs be listed in the table as well?*

Response: Thank you for your constructive suggestion. PSMs are included in new Fig.1c.

3. *Mass spectrometry methods: cite appropriate references for in-gel digestion and phosphopeptide enrichment.*

Response: Thank you for pointing this out. The references are added in the revised manuscript.

4. When referring to phosphorylation analysis in the Results section, please clarify that it was done using mass spectrometry. It is mentioned in the Methods sections but not in the rest of the article, especially since it is such an important aspect of TBC1D23 pathway within Golgi.

Response: We appreciate your suggestions. We have revised our manuscript as you suggested.

Reviewer #5:

Remarks to the Author

Tu and colleagues report on a new regulatory mechanism mediated by interaction of TBC1D23 and LKB1 at the Golgi to control AMPK activation under energy stress and potentially controlling trafficking. Multiple complementary experimental approaches and model systems were used.

The work is potentially of a broad interest, in principle, as it is dealing with a new circuit controlling energy stress response via Golgi. Besides the mechanistic insights, interest also results from its link to human disease (pontocerebellar hypoplasia, PCH). There are however some issues that should be taken into consideration and that require extensive revision.

This reviewer identified major concerns in data presentation, quantification, and assessment that weaken the validity of the interpretations and conclusions, and the suitability of the manuscript to the journal. Key Springer Nature standard requirements are not met in the submitted work and should be carefully resolved by the authors.

Generally, the paper would benefit from a more precise introduction to the importance of AMPK activation and its modality in health and disease, imaging parameters are not provided, and images shown are often not sufficient for the claims, control experiments are performed on different cell types without clear explanation, methods are poorly/barely described overall, hindering reproducibility. Statistical analysis is somewhat arbitrary. N, presence of replicated experiments as well as quantification and statistical assessment are not always provided for cell experiments. In addition, the role of organelles and Golgi in neurodevelopmental diseases (for instance ARF1 is only mentioned in parts of the results....etc..) should be addressed in the Discussion, to permit a broader evaluation of the results in the field.

Response: We appreciate the reviewer' comments and constructive suggestions. To address the reviewer' concerns, we have performed a number of critical experiments accordingly and made extensive revision in our revised manuscript.

I discuss some of points that should be addressed below.

Major points:

1. *Different KD and KO approaches, different cell types: authors seem to switch between cells silenced with siTBC1D23 and cells in which a KO was obtained by CRISPR-Cas9. Given the difference in KD ability, the use of one or the other approach should be addressed and discussed for each experiment. Same for the type of cells used.*

Response: To determine how TBC1D23 deficiency affect AMPK activation, we initially used siRNA to interfere expression of TBC1D23. Subsequent cellular experiments were performed in TBC1D23 KO cells (HEK293T and HepG2), which were generated by CRISPR-Cas9. Following

your suggestion, we examined effects of TBC1D23 knockdown on AMPK activation in response to energy stress. our results indicated that shRNA-mediated TBC1D23 ablation also resulted in decreased AMPK activation (**Graphic 3**), similar to TBC1D23 KO by CRISPR-Cas9.

Graphic 3 TBC1D23 deficiency results in compromised AMPK activation. HEK293T cells infected with virus encoding control shRNA (shNC) or shTBC1D23 were subjected to glucose starvation (GS) for the indicated time. Cells were collected and immunoblotted with antibodies indicated.

For assays in human iPSC-derived neurons, shRNAs targeting human TBC1D23 were used to attenuate its expression. We tried CRISPR-Cas9 in iPSC-derived neurons. Due to low efficiency of virus infection (single vector for co-delivery of Cas9 and gRNAs), we failed to interfere TBC1D23 expression. Instead, TBC1D23 was knockdown by shRNA.

We performed most of our experiments in HEK293T and HepG2 cells, and used HeLa and HUVEC cells to further prove our points. HEK293T cells are easily transfected; thus, we used HEK293T for MS analysis and pulldown/IP assays. In contrast with HeLa cells, HepG2 cells are LKB1 positive. Thus, we used HepG2 cells for IF and FRET assays. As it was reported that LKB1 is required for activation of AMPK in HUVEC cells, so we also examined effects of TBC1D23 knockdown on AMPK activation in HUVEC cells[1].

2. In fig.2, co-localization between LBK1 and Golgi marker: the development of super-resolution techniques has made it clear that co-localization analysis performed from standard microscopy are quite prone to errors and should be approached with caution. In these experiments, LBK1 fluorescence signal appear quite saturated (pixels saturation), which is one of the main issues leading to exaggerated and false results especially with Pearson 'correlation analysis. Authors should provide additional examples of images to see whether signal saturation is indeed an issue of the whole population of cells analyzed, if confirmed then this should be addressed for all the co-localization experiments. In addition, given a number of tools/algorithms available and the varying parameters to threshold the images beforehand authors should explain how the analysis was performed. I also suggest performing mild deconvolution on non-saturated images and it would be beneficial to examine other parameters such as Meanders'coefficients. Similar comment for Fig.6a,b and others. Here also again more cells or a lager field of view should also be provided;

Response: Thanks for your suggestions. We have repeated co-localization experiments carefully and use Manders' coefficients for Colocalization analysis in the revised manuscript (new Fig. 2a, 2b and 6a). A lager field of view is provided in the revised manuscript (new Fig. 2a, 2b,3g, 6a, 6d).

3. Line 333, Suppl.Fig.5: authors claim to analyze TGN46 trafficking; however, they did not perform any dynamic analysis, nor do they image cargo movement. Authors should just confine their claims,

stating that they assess TGN46 localization at Golgi, where the cargo is known to be transport to. The impaired trafficking, claimed uniquely by the problematic Pearson's correlation (see above) and a static picture is only a speculation at this point. Same for CI-MPR. Authors should discuss other possible explanations for the observed reduced localization of TGN46 at Golgi;

Response: Thanks a lot for the suggestions! We have modified our text as suggested (Line 329-337). And we used Manders' coefficients for Colocalization analysis (new supplementary Fig. 5f, h) in the revised manuscript.

4. The choice of the statistical analysis seems arbitrary. Authors appear to choose between t-test and analysis of variance without consistency. All the zebrafish experiments with more than two groups within one experiment are correctly assessed by ANOVA (F-test + post-hoc), while some of the cell-culture experiments with the same settings are analyzed via t-test. This approach even without any post-hoc correction leads to amplification of the type I error. The probability that the observed phenomenon is obtained by chance instead of underlying a biological meaning is therefore increasing. Such arbitrary choice in statistical assessment is poor data analysis quality and makes it hard to judge the conclusions overall. It should also be considered whether a 2-way ANOVA should be used in some occasions, for instance for the experimental set up in Fig. 1i where two variables exist: time and genetic condition. Therefore, authors should re-assess all the statistical analyses in the manuscript and clearly explain the choice made for the tests used. A List with all the experimental setups and tests, including analysis of normality would be ideal. This would make the claims of the study solid;

Response: We are grateful to the reviewers' criticisms. Following your suggestion, we have checked our statistical analysis carefully. Statistical significance of the difference between two group was determined using Student's t test. And statistical significance of the difference between multiple groups, an ordinary one-way or two-way ANOVA (if two variables exist) was used, followed by Tukey's, Sidak's, or Dunnett's as indicated in the figure legends.

5. Very often the number of replicates (for examples for WB and IP) is not given, as well as the quantification. Please authors should always provide the number of replicates for each experiment as requested and also the quantification including mean and SEM and statistical assessment;

Response: We apologize for not having explained this more clearly in our original submission. All experiments are performed at least in triplicate, and we have added this statement and quantification in section entitled "statistics and reproducibility" in Methods.

6. In general more cells/large fields of view should be shown for each experiment, methods for image acquisition and analysis should be better described. For FRET imaging the method used to do imaging (sensitized emission?) wavelengths used and laser power as well as ratiometric analysis should be described;

Response: Following your suggestions, we have revised our figures with a larger field of view and updated methods with better description. For FRET imaging, "The images were acquired using a Leica DMI6000B total internal reflection fluorescence microscope (Leica) equipped with BP420/10 excitation filter, a 440/520 dichroic mirror and two emission filters (BP472/30 for cyan fluorescent protein (CFP) and BP542/27 for YFP)" changed as: "The imaging experiments were performed using a Leica DMI6000B total internal reflection fluorescence microscope (Leica), equipped with

mercury lamp as laser power. The images were acquired with BP420/10 excitation filter, a 440/520 dichroic mirror and two emission filters (BP472/30 for cyan fluorescent protein (CFP) and BP542/27 for YFP)". BP420/10 indicates the filter detects the wavelength from 415 nm to 425 nm. To indicate the ratiometric analysis, we have corrected the y axis of Fig.2d and e as "AMPK activity (Normalized FRET/CFP)".

7. *In the FRET experiment again, When providing LBK1 in HeLa cells (LBK1^{-/-}) why authors do not show the FRET signal increase correlating to AMPK activation?*

Response: We appreciate the reviewer's thoughtful suggestions. Our LKB1 has a GFP tag when introduced in HeLa cells. As the wavelength of GFP (excitation:488 nm, emission:507 nm) is close to that of CFP (excitation:405 nm, emission:485 nm), the FRET experiment would be very difficult. Thus, we determined AMPK activation at the whole cell level. We agree that FRET will provide more specific information at the organelle level.

8. *Quantification of the images in the WB should also been shown (Fig.S2). Why do authors choose AICAR instead of GS stimulus as opposed to the experiment with TBC1D23 KO cells they want to directly compare to? This should be explained. Also, the number of replicates for all these experiments is not clear (not given). This is a general comment;*

Response: We appreciate the reviewer's suggestions. We have quantified the western blot intensity and the quantitative data have been included in the revised manuscript (new supplementary Fig.2). In supplementary Fig.2b, glucose starvation was chosen. AICAR, a widely used AMPK activator, which was reported to cause the phosphorylation of GBF1 at T337 and subsequent Golgi disassembly[2]. We sought to determine the role of LKB1 in Golgi-AMPK activation, so we treated cells with AICAR. We have included the number of replicates for all experiments in section entitled "statistics and reproducibility" in Methods.

9. *In Fig.3g/h and Fig6e it is not clear how "Golgi elements" were counted. Please specify, as the term "elements" is vague. Did the authors count large and small objects observed in the image? The occurrence of GM130 patches different in size might underly different patterns of fragmentation. More examples of cells should be provided, only 2 or 3 cells are shown now;*

Response: We apologize for not making it clear in the original submission. The analysis is performed by manual counting, both large and small objects being counted. We have included "Golgi assembly analysis" in Methods (Cell culture, transfection and immunofluorescence). And more examples of cells are provided in the revised manuscript.

10. *From the images shown, Met seems to have an intermediate effect on Golgi disassembly in KO cells, as also the statistics (should be an ANOVA, see below) would suggest and not what the authors claim, or what do they mean that the disassembly was "impaired"? In Fig. 6d authors should show more cells and higher magnification, so far the rescue in Golgi fragmentation obtained by LBK1-G is hard to judge as only 1 cell is shown. Also, for the condition co-transfected with LBK1 GM130 seems fainter (why?);*

Response: In the revised manuscript, the statistical significance of the difference is determined by one-way ANOVA, followed by Tukey's test. Phosphorylation of GBF1 by AMPK was reported to promote Golgi disassembly[2]. Consistent with impaired AMPK activation of TBC1D23 KO cells,

Golgi disassembly of KO cells treated with metformin is also significantly decreased, in comparison with WT cells. Fig. 6d has been updated with more cells and higher magnification. As shown in new Fig. 6d, GM130 intensity in cells transfected with LKB1 is comparable to the adjacent cells.

11. The microscopy images showing zebrafish brain in the transgenic fish are of poor quality, with intense signal bleed from planes other of the one in focus. Authors should specify whether these are single plane images of z-stacks. Probably an epifluorescence microscope was used. Also, they should state the parameters used for each condition/fish. Did the same laser intensity was used? The authors should at least indicate different brain sub-domains for general audience not familiar with zebrafish brain. Clearly, a major involvement of the whole brain is seen in the morphants. Why do authors choose to report only midbrain measurements? Is this correlated to the PCH? This should be explained. What is the evidence for cerebellum itself, or hindbrain beyond r2 and other regions? In case regions others than the midbrain did not show any statistically significant difference this should be reported, and the specific involvement in midbrain should be discussed;

Response: We are grateful to the reviewer's constructive suggestions. Following your suggestions, we have improved the quality of zebrafish brain pictures by using Olympus IX83 P2ZF spinning disk confocal microscopy, and presented as maximum intensity z-stack projections (new Fig.7g and Fig.8f).

Due to individual differences in zebrafish, the fluorescence brightness of each zebrafish is not consistent. We characterized midbrain size by measuring relative area, and fluorescence intensity did not affect our measurements. Therefore, we use automatic exposure for each zebrafish to capture higher quality images.

As you mentioned, the development of the whole brain is affected in our morphants, including cerebellum, hindbrain beyond r2 and other brain regions. We chose the size of the midbrain to characterize the brain abnormalities, and the disruption of midbrain appear to be consistent with the midbrain defects observed with brain imaging of human patients. Besides, midbrain abnormalities have been reported in several articles in previous studies of PCH disease[3-6].

11. Fig.6, recruitment of LKB1 to the Golgi is impaired in TBC1D23 KO cells: authors claim that this and the activation of AMPK is rescued by TBC1D23, however microscopy evidence for this is not presented and a different cell type is used (HEK instead of HepG2), likely with a different KO/KD approach (sg instead of KO?). This is likely true also in other points of the manuscript. I find the nomenclature here very confusing. Authors should address the choice of using different cells and different approaches when demonstrating a rescue, this is no immediate neither probably correct;

Response: Rescue of impaired AMPK activation of TBC1D23 KO cells with TBC1D23 was performed in HEK293T cells, which was easily transfected. Subsequent rescue assay (mainly IF) was performed in TBC1D23 KO HepG2 cells with LKB1 or Golgi-targeted LKB1, aiming to demonstrate the significance of TBC1D23-mediated LKB1 recruitment to Golgi. We have corrected the issues about nomenclature in the revised manuscript. As for KO/KD approach and different cells, please see our response above.

12. Neuronal defects: Fig.7 and S6, authors claim to use iPSC-derived neurons and to test the

efficacy of different ShRNA to silence TBC1D23. However, in Fig.S6a they show western blot performed on HEK cells (?). How does this compare to the effect in iPSC cells? This is completely hidden within the text and only appears to an attentive reader. Please revise. Similarly to the “si”, “KO” and “sg” nomenclature story, authors should state clearly why they use different systems in different contexts and cannot rely on different cells to validate efficacy of the approach then used in a different cell type;

Response: The infection efficiency of iPSC-derived neurons is low, so it would be difficult to determine the knockdown efficiency of shRNA. It is common to determine knockdown efficiency in cell lines, including HEK293T[7-9]. Since these shRNA target human TBC1D23, then the effective shRNA in HEK293T cells will work in human iPSC cells when delivered successfully. We then chose the neurons successfully infected (GFP positive) for morphology analysis. We have corrected the issues about nomenclature in the revised manuscript.

13. Moreover, quantification of the western blot to test sh efficiency is missing. Without quantification and replicates (how many times was the experiment repeated?) it is not possible to judge the % of silencing (i.e. GAPDH seems higher in the control). Please show mean and SEM in the quantification;

Response: In new Supplementary Fig.6a, the quantitative data of three replicates is included. When 30-40% HEK293T is successfully infected, shRNA-1 exhibited ~25% knockdown efficiency. We confirmed the knockdown efficacy by upregulating amount of virus. As shown in new Supplementary Fig.6b, shRNA-1 effectively deplete the expression of TBC1D23 when ~70-80% HEK293T is successfully infected. (new supplementary Fig.6b).

14. Primary branching in Fig.7 and S6: authors claim that a decreased primary branching is observed in cells depleted of TBC1D23. First, control cells (without Sh) are not shown and should be shown directly in main figure. One should be able to judge the “rescue” of the branching comparing directly the conditions + LBK1-G etc to control scenario, statistically;

Response: Thanks a lot for the suggestions. Neuronal morphology of control neurons (shNC), shTBC1D23 neurons, and shTBC1D23 neurons with ectopic expression of indicated proteins are shown together. For better layout of the figures, the representative images are shown in new supplementary Fig.6c, and the statistical analysis data are shown in main figure (new Fig.7a-c).

15. Authors claim that neuronal defects observed here confirm previous reports. However, in the Huang et al, PNAS 2019 the authors showed the opposite in terms of branching. Reduced TBC1D23 in zebrafish induced increased branching (not decreased) in the previous report. Why does a similar approach reduce branching now? How is this “confirming” previous reports? Indeed, in the current work authors clearly show representative images of CaP motoneurons with hyperbranching (Fig. 7e) confirming their previous report. Why do iPSC cells show a opposite phenotype? Are those differentiated in motoneurons or what? Authors should discuss this. Instead authors say that the data are “nicely consistent with our results in iPSC..” Branching seems again increased in LBK1 MO model that should be addressed;

Response: We apologize for have explained this more clearly in our original submission. The effects of TBC1D23 deficiency on neurite length was consistent with the observation in Neuro2a neuroblastoma cells. iPSC-derived cells are cortical neurons. CaP motoneurons is a primary motor

neuron part of the spinal cord. The axon of the CaP motoneuron extends ventrally from the ventral root, within the space between the notochord and the medial surface of the axial muscles. The analyzed branch of iPSC-derived neurons locates at the proximity neurite, while branch of CaP motoneurons is the distal neurite region. TBC1D23 might have differential role in neuronal branch at proximal and distal neurite. We discussed the discrepancy regarding the effects of TBC1D23 deficiency on branch, and we assumed that it might be due to different types of neuron and stage of neuronal differentiation. TBC1D23 deficiency caused abnormal branch in both iPSC-derived neurons and zebrafish, in comparison with control groups, indicating the essential role TBC1D23 in maintaining normal neuronal development. LKB1 MO caused increased branch in CaP motoneurons of zebrafish, similar to TBC1D23 MO, which is consistent our model that TBC1D23 and cooperate in regulating neuronal development. However, LKB1 knockdown result in significantly less branched axons in cortical neurons after 5 day of culture in vitro (DIV). This differential result t is similar to what we observed in TBC1D23 deficient iPSC-derived neurons and zebrafish.

16. Also, was reduced length of CaP motoneurons reported for TBC1D23 MO injection in zebrafish before? In Huang et al, PNAS 2019 this does not seem the case;

Response: In previous studies[3, 4], we also observed a significant reduction in length of CaP motoneuron axons in zebrafish injected with TBC1D23-MO, compared with the control. We focused on the branch phenotype in these papers.

17. What are the dots in graphs quantifying CaP length? How many CaP were counted per fish? A nested analysis should be considered if more cells per fish are counted.

Response: We measured 3 to 4 CaP motoneuron axons at the same locations in each zebrafish, with each point representing the relative length of one axon.

Minor points:

1. Text: authors should explain better (broader context) why it is interesting to understand the role of LKB1 – AMPK regulation at Golgi? – Please revise Abstract and Intro;

Response: We have revised the abstract and introduction as suggested.

2. Methods are not detailed and in their current form do not fit the standards of the journal. For instance, imaging conditions with resolution, step size, speed etc... are not provided. Similarly, the algorithms used and image processing are not described. Articles should provide experimental details to improve reproducibility in the community, authors should amend accordingly all the methods;

Response: We apologize for not having explained methods more clearly in our previous submission. We have revised our “methods” section with more detailed description.

3. English should be improved in the methods section... e.g. line 906 “images...were taken”, proper terminology and details should be used, such as “x,y scans were acquired atconfocal withresolution etc..”;

Response: We have modified the methods section in the revised manuscript.

4. Letters in the figures should follow a logical order (not the case for some figures, see Fig.6);

Response: We have fixed this.

5. To better appreciate the differences in AMPKa regulation in siTBC1D23 and in KO cells authors should add densitometric quantifications for Figure 1f, g;

Response: Thanks for your wonderful suggestion. The quantitative data is included in our revised manuscript.

6. In Fig.2 expression of LKB1 pointed (vesicles?) after GS. Could authors comment this before line 266? A higher magnification, not saturated image should be provided (see also major point below concerning Pearson);

Response: We appreciate the reviewer's suggestions. We have repeated this experiment, and updated the manuscript with new Fig.2a. The pointed LKB1 might be endosomal and/or lysosomal LKB1.

7. line 243 : "we found that"...authors should explain what they did to obtain the finding to a general audience and describe the figure in details;

Response: Thanks for the suggestion. We have changed our text as the following, "Our previous studies indicated that three consecutive positively-charged residues (K632K633K634) in the PH domain of TBC1D23 are critical for its binding to FAM21. To determine whether the same residues are involved in LKB1 interaction, we generated a triple mutant by convert all three residues to the opposite charge (3K: K632E/K633E/K634E). TBC1D23 3K mutant almost abolished the binding to LKB1".

8. line 251: remove "intriguingly" ;

Response: We have fixed this.

10. "SgTBC1D23" equals "KO" or "si"(siRNA)? Sometimes authors refer to one and sometimes to the other, nomenclature should be consistent, what does "sg" stand for?

Response: We apologize for not having explained this more clearly in the original submission. sgTBC1D23 is an abbreviation for "single guide RNA targeting TBC1D23", and "sgTBC1D23" in the original submission indicated a pool of cells infected with viruses encoding hSpCas9&sgTBC1D23. We have corrected the issues about nomenclature in the revised manuscript.

11. Line 339: "statures" not sure this is correct. Maybe "status".

Response: Corrected as "status", thank you!

12. Line 1266: the term "confocal immunofluorescence" is not correct. It is an immunofluorescence and samples 'images were acquired via confocal microscopy. Authors should check the main text for proper terminology;

Response: Thank you very much for your suggestion. We revised "confocal immunofluorescence" as "Confocal imaging"

13. Fig6a : “LBK1-Flag” should be written instead of just “Flag”;

Response: We have fixed this.

14. Line 350 : “attenuate” = typo;

Response: Corrected as “attenuate”, thank you!

15. Line 374 : remove “nicely”;

Response: We have modified the sentence as suggested.

16. In the reporting summary it is claimed that N, degree of freedom etc are provided, but this reviewer failed in finding such info. Exact p- values are not reported.

Response: Exact p- values are included in the figures of the revised manuscript.

References:

1. Xie, Z., et al., *Phosphorylation of LKB1 at serine 428 by protein kinase C-zeta is required for metformin-enhanced activation of the AMP-activated protein kinase in endothelial cells.* Circulation, 2008. **117**(7): p. 952-62.
2. Miyamoto, T., et al., *AMP-activated protein kinase phosphorylates Golgi-specific brefeldin A resistance factor 1 at Thr1337 to induce disassembly of Golgi apparatus.* J Biol Chem, 2008. **283**(7): p. 4430-8.
3. Huang, W., et al., *Structural and functional studies of TBC1D23 C-terminal domain provide a link between endosomal trafficking and PCH.* Proc Natl Acad Sci U S A, 2019. **116**(45): p. 22598-22608.
4. Liu, D., et al., *Structure of TBC1D23 N-terminus reveals a novel role for rhodanese domain.* PLoS Biol, 2020. **18**(5): p. e3000746.
5. Boczonadi, V., et al., *EXOSC8 mutations alter mRNA metabolism and cause hypomyelination with spinal muscular atrophy and cerebellar hypoplasia.* Nat Commun, 2014. **5**: p. 4287.
6. Slavotinek, A., et al., *Biallelic variants in the RNA exosome gene EXOSC5 are associated with developmental delays, short stature, cerebellar hypoplasia and motor weakness.* Hum Mol Genet, 2020. **29**(13): p. 2218-2239.
7. Chen, J.G., et al., *Zfp312 is required for subcortical axonal projections and dendritic morphology of deep-layer pyramidal neurons of the cerebral cortex.* Proc Natl Acad Sci U S A, 2005. **102**(49): p. 17792-7.
8. Garcez, P.P., et al., *Cenpj/CPAP regulates progenitor divisions and neuronal migration in the cerebral cortex downstream of Ascl1.* Nat Commun, 2015. **6**: p. 6474.
9. Tracy, T.E., J.J. Yan, and L. Chen, *Acute knockdown of AMPA receptors reveals a trans-synaptic signal for presynaptic maturation.* EMBO J, 2011. **30**(8): p. 1577-92.

REVIEWER COMMENTS

Reviewer #1 (Remarks to the Author):

The authors have sufficiently answered all questions and concerns. No further revisions are requested.

Reviewer #2 (Remarks to the Author):

The authors have responded to this reviewers questions, and the proposed revisions are acceptable. However, on reinspection in the light of the comment of other reviewers, it appears that the sample size in individual experiments is unclear. A statement that experiments were performed at least in triplicate (p.42) is generally not sufficient. There are several bar graphs that do not show the data points, and where the true sample size is unknown. This information should be provided in the figure legends before acceptance.

Reviewer #3 (Remarks to the Author):

The authors have addressed many of the reviewers' comments. However, the following points still need to be addressed:

1. Blot reproduction and quantification: Blots that are used to make quantitative comparisons need to be quantified using densitometry of repeated blots (Fig. 1f, 1g, 1h, 1k, 2f, 3d, 3e, 3f, 4b, 4c, 4d, 5, 6b, 6c, 8 etc). As these blots have been done in triplicate, please include statistical error (eg. 0.61 ± 0.8).
2. Lyso AMPK activity measurement: The use of lysosomal acidification by Lysosensor is not the same as quantifying AMPK activity at the lysosome. If lyso-ABKAR "did not work out for unknown reason", what about lyso-ExRai AMPKAR [1], cell fractionation followed by western [2,3], or immunoblotting on lysosomes purified by LysoIP [4]?
3. Golgi assembly analysis: The description provided in lines 747-750 is still inadequate/unclear. First, the authors state that they "manually counting, both large and small" and then state they used "unbiased image analysis" to count fragments. Are fragments the same as elements? Manual counting vs. unbiased image analysis are contradictory statements – if one is manually counting this is not unbiased. If image analysis is used, please describe the filtering, segmentation, etc. settings used to identify objects. Furthermore, there is no detailing of what large or small objects are quantitatively defined as.
4. Colocalization analysis (minor): Please clarify in the methods at line 750 which ImageJ tool was used (JaCOP, Coloc2, etc.) and the settings used within (PSF, Costes/Bisection, thresholded Pearson's or Mander's coefficients, etc.). If a custom Macro was used, briefly describe the steps/settings used.

References

1. Schmitt et al. Nat Commun 2022 Jul 5;13(1):3856. doi: 10.1038/s41467-022-31190-x.
2. Bai et al. Autophagy 2022 Jul;18(7):1673-1693. doi: 10.1080/15548627.2021.1997051.
3. Zhang et al. Methods Enzymol 2017;587:465-480. doi: 10.1016/bs.mie.2016.09.071.
4. Jia et al. Mol Cell 2020 Mar 5;77(5):951-969.e9. doi: 10.1016/j.molcel.2019.12.028.

Reviewer #4 (Remarks to the Author):

Thank you for addressing my comments and adding new references. Reference #70 however, is not appropriate to the methodology executed in this manuscript. That paper does IMAC differently than what is reported in your manuscript, making it a bit confusing. It can be dropped or replaced by another reference.

Reviewer #5 (Remarks to the Author):

General remarks.

In the revised manuscript, authors attempted to address most of my concerns and answer to the remarks. The manuscript has been improved, and I believe the findings are conceptually valid. However, there are still major aspects that should be fixed along the concerns originally expressed that are still required to strengthen the solidity of the mechanistic findings before the work can be considered as suitable for publication.

Globally, description and rationale of the experimental design and the experimental/analytic approaches for testing the working hypotheses are still confusing and not fully explained in the text. The manuscript still lacks basic details on measurements and acquisition parameters. Statistics is not fully described and many p-values honestly to not seem to match the observed variability. In general, I find the zebrafish data not technically and mechanistically informative to compensate for the shortcomings of the in vitro experiments. Only the morpholino approach was used for instance, and data on IPSCs do not actually match the results obtained in vivo. Such complexity which was not addressed originally and barely has been in the revised version of the work should be discussed, and the considerations related to brain development and disease should be toned down. All of this make the current version not compatible with the rigorous style of Nature Communications.

Major points:

1.

Different KD and KO approaches, different cell types: authors seem to switch between cells silenced with siTBC1D23 and cells in which a KO was obtained by CRISPR-Cas9. Given the difference in KD ability, the use of one or the other approach should be addressed and discussed for each experiment. Same for the type of cells used.

Authors' reply: To determine how TBC1D23 deficiency affect AMPK activation, we initially used siRNA to interfere expression of TBC1D23. Subsequent cellular experiments were performed in TBC1D23 KO cells (HEK293T and HepG2), which were generated by CRISPR-Cas9. Following your suggestion, we examined effects of TBC1D23 knockdown on AMPK activation in response to energy stress. Our results indicated that shRNA-mediated TBC1D23 ablation also resulted in decreased AMPK activation (Graphic 3), similar to TBC1D23 KO by CRISPR-Cas9.

Graphic 3 TBC1D23 deficiency results in compromised AMPK activation. HEK293T cells infected with virus encoding control shRNA (shNC) or shTBC1D23 were subjected to glucose starvation (GS) for the indicated time. Cells were collected and immunoblotted with antibodies indicated.

For assays in human iPSC-derived neurons, shRNAs targeting human TBC1D23 were used to attenuate its expression. We tried CRISPR-Cas9 in iPSC-derived neurons. Due to low efficiency of virus infection (single vector for co-delivery of Cas9 and gRNAs), we failed to interfere TBC1D23 expression. Instead, TBC1D23 was knockdown by shRNA.

We performed most of our experiments in HEK293T and HepG2 cells, and used HeLa and HUVEC cells to further prove our points. HEK293T cells are easily transfected; thus, we used HEK293T for MS analysis and pulldown/IP assays. In contrast with HeLa cells, HepG2 cells are LKB1 positive. Thus, we used HepG2 cells for IF and FRET assays. As it was reported that LKB1 is required for activation of AMPK in HUVEC cells, so we also examined effects of TBC1D23 knockdown on AMPK activation in HUVEC cells[1] .

Remarks: I thank the authors for clarifying these points. However, they should explain the rationale of using different cell types for the different experiments performed (as they do at line 146/167 of the revised manuscript). I cannot see this properly done in the rest of the revised version of the manuscript.

Line 132-133 for instance: the authors do not specify in which cells they do the first silencing (si) experiments, but then refer to HepG2 for CRISPR-Cas9 soon after leaving the reader to wonder why did they use different strategies without explaining the rationale. Authors should explain each time within the text why they switched to HUVEC and HepG2. Also, authors should specify the type of cells used in "Methods" and in the figure legends for each of the panels. Also, it should be clear from the text that a subset of the results was corroborated in HUVEC cells.

2.

In fig.2, co-localization between LBK1 and Golgi marker: the development of super-resolution techniques has made it clear that co-localization analysis performed from standard microscopy are quite prone to errors and should be approached with caution. In these experiments, LBK1 fluorescence signal appear quite saturated (pixels saturation), which is one of the main issues leading to exaggerated and false results especially with Pearson 'correlation analysis. Authors should provide additional examples of images to see whether signal saturation is indeed an issue of the whole population of cells analyzed, if confirmed then this should be addressed for all the colocalization

experiments. In addition, given a number of tools/algorithms available and the varying parameters to threshold the images beforehand authors should explain how the analysis was performed. I also suggest performing mild deconvolution on non-saturated images and it would be beneficial to examine other parameters such as Manders' coefficients. Similar comment for Fig.6a,b and others. Here also again more cells or a larger field of view should also be provided.

Authors' reply: Thanks for your suggestions. We have repeated co-localization experiments carefully and use Manders' coefficients for Colocalization analysis in the revised manuscript (new Fig. 2a, 2b and 6a). A larger field of view is provided in the revised manuscript (new Fig. 2a, 2b,3g, 6a, 6d).

Remarks: I can see that co-localization analysis was majorly improved. Manders' coefficients are normally $2 (M1 \text{ and } M2)$ with respect to pixels in Image-1 overlapping with Image-2 and viceversa. Authors can provide both, or specify the use of one. In the latter case, the label cannot be "Manders' coefficients" but

rather “Manders’coefficient 1 or 2”. The manuscript also still lacks a detailed explanation of the method used. For instance, which algorithm was used in Image J? Did the author use masking of a certain region of the cell? Again, a larger field of view should be always shown and at least two cells per condition.

4.

The choice of the statistical analysis seems arbitrary. Authors appear to choose between t-test and analysis of variance without consistency. All the zebrafish experiments with more than two groups within one experiment are correctly assessed by ANOVA (F-test + post-hoc), while some of the cell-culture experiments with the same settings are analyzed via t-test. This approach even without any post-hoc correction leads to amplification of the type I error. The probability that the observed phenomenon is obtained by chance instead of underlying a biological meaning is therefore increasing. Such arbitrary choice in statistical assessment is poor data analysis quality and makes it hard to judge the conclusions overall. It should also be considered whether a 2-way ANOVA should be used in some occasions, for instance for the experimental set up in Fig. 1i where two variables exist: time and genetic condition. Therefore, authors should re-assess all the statistical analyses in the manuscript and clearly explain the choice made for the tests used. A List with all the experimental setups and tests, including analysis of normality would be ideal. This would make the claims of the study solid.

Authors’ reply: We are grateful to the reviewers’ criticisms. Following your suggestion, we have checked our statistical analysis carefully. Statistical significance of the difference between two group was determined using Student’s t test. And statistical significance of the difference between multiple groups, an ordinary one-way or two-way ANOVA (if two variables exist) was used, followed by Tukey’s, Sidak’s, or Dunnett’s as indicated in the figure legends.

Remarks: Authors have improved the statistical analysis, overall. However, they should more clearly explain why a certain hypothesis test was used beyond citing t-test or ANOVA. Where the data checked for normality? Why did the authors choose Tukey vs Sidak etc ...as post-hoc test? Also in the “Reporting summary” authors claim to provide info such as “CI”, “degrees of freedom”... though this reviewer did not find them reported in the manuscript. I suggest authors to make a supplementary table with all the statistical specs for each figure panel. Also, authors declare they provide “estimates of effect size” but I don’t see this in the manuscript. Same for “one side, two sides”.

5.

The number of replicates very often is not given (see WB and IP), as well as quantification. The authors should always provide the number of replicates for each experiment as requested and also the quantification including mean and SEM and statistical assessment.

Authors' reply: We apologize for not having explained this more clearly in our original submission. All experiments are performed at least in triplicate, and we have added this statement and quantification in section entitled "statistics and reproducibility" in Methods.

Remarks: N should be given for each experiment in the figure panel. When analyzing co-localization or Golgi elements from each cell, how many cells per experiment were counted? Do the dots in the graphs represent pooling of cells from different biological replicates? or different fields? Such aspects are not explained, however they are flagged in the Reporting Summary.

6.

In general, more cells/large fields of view should be shown for each experiment, methods for image acquisition and analysis should be better described. For FRET imaging the method used to do imaging (sensitized emission?) wavelengths used and laser power as well as ratiometric analysis should be described.

Authors' reply: Following your suggestions, we have revised our figures with a larger field of view and updated methods with better description. For FRET imaging, "The images were acquired using a Leica DMI6000B total internal reflection fluorescence microscope (Leica) equipped with BP420/10 excitation filter, a 440/520 dichroic mirror and two emission filters (BP472/30 for cyan fluorescent protein (CFP) and BP542/27 for YFP)" changed as: "The imaging experiments were performed using a Leica DMI6000B total internal reflection fluorescence microscope (Leica), equipped with mercury lamp as laser power. The images were acquired with BP420/10 excitation filter, a 440/520 dichroic mirror and two emission filters (BP472/30 for cyan fluorescent protein (CFP) and BP542/27 for YFP)". BP420/10 indicates the filter detects the wavelength from 415 nm to 425 nm.

To indicate the ratiometric analysis, we have corrected the y axis of Fig.2d and e as “AMPK activity (Normalized FRET/CFP)”.

Remarks: what does “Normalized FRET/CFP” mean? Normalized vs what? Line 1296: “The FRET/CFP ratio was measured and normalized to cells incubated with DMEM” The authors should explain how was this normalization performed? What does it mean? Authors should really make an effort to use precise terminology and detail the analysis they performed such that readers are in condition to assess accuracy as well as to support reproducibility. Imaging and analysis using cells could further benefit of improvement in the description, along the lines of the details provided for zebrafish, see also the other points. A statement whether images and analysis within a certain experiment were obtained/performed with the same parameters with respect to acquisition and post-processing should be added, or otherwise explained if that is not the case.

7.

In the FRET experiment again, When providing LBK1 in HeLa cells (LBK1^{-/-}) why authors do not show the FRET signal increase correlating to AMPK activation?

Authors' reply: We appreciate the reviewer's thoughtful suggestions. Our LKB1 has a GFP tag when introduced in HeLa cells. As the wavelength of GFP (excitation:488 nm, emission:507 nm) is close to that of CFP (excitation:405 nm, emission:485 nm), the FRET experiment would be very difficult. Thus, we determined AMPK activation at the whole cell level. We agree that FRET will provide more specific information at the organelle level.

Remarks: This limitation could be surpassed by using a LBK1 without tag in your transfection experiment. Using FRET to test this hypothesis would be more consistent, appropriate and elegant. WB per se would corroborate the finding.

Given the specificity of the FRET AMPK sensor to Golgi or mitochondria, one would also expect to see the actual images of the FRET signal within cells. Why don't the authors show the images used to make calculations? (FRET and CFP channel and the computed ratiometric image)? It is somewhat bizarre that quantification is reported without representative images of the results. Why is now the scale bar and numbers of the graph (Fig.2d etc) different from that one submitted originally?

The authors claim that there is a strong reduction of AMPK activity in Golgi in cells depleted of TBC1D23 in both basal and energy stress condition by looking at the FRET data. However, if one examines carefully the graphs presented now, the data show that there is a little decrease of activity in basal conditions (1 vs 0.7, I guess), and that there is still a small -yes negligible- increase in activity upon stress also in

TBC1D23 depleted cells. Authors should discuss these results more extensively along these lines and perhaps show a fold change quantification.

8.

Quantification of the images in the WB should also been shown (Fig.S2). Why do authors choose AICAR instead of GS stimulus as opposed to the experiment with TBC1D23 KO cells they want to directly compare to? This should be explained. Also, the number of replicates for all these experiments is not clear (not given). This is a general comment.

Authors' reply: We appreciate the reviewer's suggestions. We have quantified the western blot intensity and the quantitative data have been included in the revised manuscript (new supplementary Fig.2). In supplementary Fig.2b, glucose starvation was chosen. AICAR, a widely used AMPK activator, which was reported to cause the phosphorylation of GBF1 at T337 and subsequent Golgi disassembly[2]. We sought to determine the role of LKB1 in Golgi-AMPK activation, so we treated cells with AICAR. We have included the number of replicates for all experiments in section entitled "statistics and reproducibility" in Methods.

Remarks: I only see FRET/CFP quantification but not WB quantification. The reason to use AICAR provided here should be included in the text. N of replicates from which quantifications are performed should be specifically provided in figure legends.

9.

In Fig.3g/h and Fig6e it is not clear how "Golgi elements" were counted. Please specify, as the term "elements" is vague. Did the authors count large and small objects observed in the image? The occurrence of GM130 patches different in size might underly different patterns of fragmentation. More examples of cells should be provided, only 2 or 3 cells are shown now.

Authors' reply: We apologize for not making it clear in the original submission. The analysis is performed by manual counting, both large and small objects being counted. We have included "Golgi assembly analysis" in Methods (Cell culture, transfection and immunofluorescence). And

more examples of cells are provided in the revised manuscript.

Remarks: Basic details of image analysis are missing still. What do the authors mean now by “unbiased image analysis”? Do they mean “blind”? A better analysis should also consider the nucleus dimension, a ratio between the total area occupied by GM130 and the nucleus are should be added, which is less prone to errors as compared to manually counting fragments.

10.

From the images shown, Met seems to have an intermediate effect on Golgi disassembly in KO cells, as also the statistics (should be an ANOVA, see below) would suggest and not what the authors claim, or what do they mean that the disassembly was “impaired”? In Fig. 6d authors should show more cells and higher magnification, so far the rescue in Golgi fragmentation obtained by LBK1-G is hard to judge as only 1 cell is shown. Also, for the condition co-transfected with LBK1 GM130 seems fainter (why?).

Authors’ reply: In the revised manuscript, the statistical significance of the difference is determined by one-way ANOVA, followed by Tukey’s test. Phosphorylation of GBF1 by AMPK was reported to promote Golgi disassembly[2]. Consistent with impaired AMPK activation of TBC1D23 KO cells, Golgi disassembly of KO cells treated with metformin is also significantly decreased, in comparison with WT cells. Fig. 6d has been updated with more cells and higher magnification. As shown in new Fig. 6d, GM130 intensity in cells transfected with LKB1 is comparable to the adjacent cells.

Remarks: In Fig.6d, the field still only one cell is shown for the LKB1 and rescue experiment so it is hard to judge. Again, the GM130 channel is too low and the quality of the image seems worst compared to submitted. Also, the authors should paid attention to the phrasing and English. For instance, line 232 and 1117: “knockout of TBC1D23 strongly impairs Golgi disassembly” seems strange. Most likely authors wanted to say that TBC1D23 KO prevents Golgi disassembly upon Met treatment.

11.

The microscopy images showing zebrafish brain in the transgenic fish are of poor quality, with intense signal bleed from planes other of the one in focus. Authors should specify whether these are

single plane images of z-stacks. Probably an epifluorescence microscope was used. Also, they should state the parameters used for each condition/fish. Did the same laser intensity was used? The authors should at least indicate different brain sub-domains for general audience not familiar with zebrafish brain. Clearly, a major involvement of the whole brain is seen in the morphants. Why do authors choose to report only midbrain measurements? Is this correlated to the PCH? This should be explained. What is the evidence for cerebellum itself, or hindbrain beyond r2 and other regions? In case regions others than the midbrain did not show any statistically significant difference this should be reported, and the specific involvement in midbrain should be discussed.

Authors' reply: We are grateful to the reviewer's constructive suggestions. Following your suggestions, we have improved the quality of zebrafish brain pictures by using Olympus IX83 P2ZF spinning disk confocal microscopy, and presented as maximum intensity z-stack projections (new Fig.7g and Fig.8f).

Due to individual differences in zebrafish, the fluorescence brightness of each zebrafish is not consistent. We characterized midbrain size by measuring relative area, and fluorescence intensity did not affect our measurements. Therefore, we use automatic exposure for each zebrafish to capture higher quality images.

As you mentioned, the development of the whole brain is affected in our morphants, including cerebellum, hindbrain beyond r2 and other brain regions. We chose the size of the midbrain to characterize the brain abnormalities, and the disruption of midbrain appear to be consistent with the midbrain defects observed with brain imaging of human patients. Besides, midbrain abnormalities have been reported in several articles in previous studies of PCH disease[3-6].

Remarks: I see the brain images improved. Still, if authors used different laser energy or parameters because of the difference in fluorescence, they should state the specific setting in the Methods. In Fig.8g they should explain what exactly "Relative midbrain size" means. Is it an area relative to what and how was it calculated? (again should be added in "Methods"). With respect to different brain regions, authors should clearly describe that the effect is observed on many different regions regardless of the association with human patients in PCH... otherwise, this is the classical cherry-picking attitude in selecting the data to report. The impact of these LoF on the whole brain could be functionally important for brain development understanding and might underly differences across vertebrates

12.

Neuronal defects: Fig.7 and S6, authors claim to use iPSC-derived neurons and to test the efficacy of different ShRNA to silence TBC1D23. However, in Fig.S6a they show western blot performed on HEK cells (?). How does this compare to the effect in IPSC cells? This is completely hidden within the text and only appears to an attentive reader. Please revise. Similarly to the “si”, “KO” and “sg” nomenclature story, authors should state clearly why they use different systems in different contexts and cannot rely on different cells to validate efficacy of the approach then used in a different cell type.

Authors' reply: The infection efficiency of iPSC-derived neurons is low, so it would be difficult to determine the knockdown efficiency of shRNA. It is common to determine knockdown efficiency in cell lines, including HEK293T[7-9]. Since these shRNA target human TBC1D23, then the effective shRNA in HEK293T cells will work in human iPSC cells when delivered successfully.

We then chose the neurons successfully infected (GFP positive) for morphology analysis. We have corrected the issues about nomenclature in the revised manuscript.

Remarks: I understand the technical issue, but this kind of approximation is not acceptable, and could be solved by sorting the positive cells and showing the reduced TBC1D23 expression. If authors want to keep the IPSC cell data, they should better show the efficiency of reduction from these cells. Also, the differentiation in cortical neurons is not specified and should be described. Importantly, I am concerned about the speculative interpretation the authors make based on the IPSC poor quality data -see my point below.

14.

Primary branching in Fig.7 and S6: authors claim that a decreased primary branching is observed in cells depleted of TBC1D23. First, control cells (without Sh) are not shown and should be shown directly in main figure. One should be able to judge the “rescue” of the branching comparing directly the conditions + LBK1-G etc to control scenario, statistically.

Authors' reply: Thanks a lot for the suggestions. Neuronal morphology of control neurons (shNC), shTBC1D23 neurons, and shTBC1D23 neurons with ectopic expression of indicated proteins are shown together. For better layout of the figures, the representative images are shown in new supplementary Fig.6c, and the statistical analysis data are shown in main figure (new Fig.7a-c).

Remarks: There are major concerns here, which I summarize: images of iPSC cells should be shown together with graphs and not in separated figures; they are really poor in quality and again only single/few cortical neurons are shown. A higher number of cells should be shown. The panel seems to be cut at the level where branches continue so it is not possible to really judge about branching. The graph indicates that control cells have something like 4 primary branching on average when in fact the only cell shown exhibits one long branch. How can we judge about reduction here? All the other panels showing 1 or 2 cells demonstrate similar branching number. Overall, the dots in the graph are too thick, this should be improved in all the graphs such that we are able to see the data and statistics seems really strange (p-values too good?).

15.

Authors claim that neuronal defects observed here confirm previous reports. However, in the Huang et al, PNAS 2019 the authors showed the opposite in terms of branching. Reduced TBC1D23 in zebrafish induced increased branching (not decreased) in the previous report. Why does a similar approach reduce branching now? How is this "confirming" previous reports? Indeed, in the current work authors clearly show representative images of CaP motoneurons with hyperbranching (Fig. 7e) confirming their previous report. Why do iPSC cells show an opposite phenotype? Are those differentiated in motoneurons or what? Authors should discuss this. Instead authors say that the data are "nicely consistent with our results in iPSC.." Branching seems again increased in LBK1 MO model that should be addressed.

Authors' reply: We apologize for not having explained this more clearly in our original submission. The effects of TBC1D23 deficiency on neurite length was consistent with the observation in Neuro2a neuroblastoma cells. iPSC-derived cells are cortical neurons. CaP motoneurons is a primary motor neuron part of the spinal cord. The axon of the CaP motoneuron extends ventrally from the ventral root, within the space between the notochord and the medial surface of the axial muscles. The analyzed branch of iPSC-derived neurons locates at the proximity neurite, while branch of CaP

motoneurons is the distal neurite region. TBC1D23 might have differential role in neuronal branch at proximal and distal neurite. We discussed the discrepancy regarding the effects of TBC1D23 deficiency on branch, and we assumed that it might be due to different types of neuron and stage of neuronal differentiation. TBC1D23 deficiency caused abnormal branch in both iPSC-derived neurons and zebrafish, in comparison with control groups, indicating the essential role TBC1D23 in maintaining normal neuronal development. LKB1 MO caused increased branch in CaP motoneurons of zebrafish, similar to TBC1D23 MO, which is consistent our model that TBC1D23 and cooperate in regulating neuronal development. However, LKB1 knockdown result in significantly less branched axons in cortical neurons after 5 day of culture in vitro (DIV). This differential result t is similar to what we observed in TBC1D23 deficient iPSC-derived neurons and zebrafish.

Remarks: I am familiar with neuronal branching. The explanation offered here is a pure speculation and I do not advice to have it in the main manuscript. Simply authors are comparing different cell types in different species having opposite results. On top of this, primary branching analysis was not performed in CaP motoneurons to be able to actual compare the same thing in different cell types. Also how does then the “longest neurite” in IPSC compare with neuritogenesis in CaP from fish? I do not think the data are strong enough to prove any specific involvement of the genes of interest in the development and neuritogenesis of both cell types and are conflicting. I propose to refrain the use of iPSC cells unless a better analysis is performed coupled to an improved discussion of the effects seen. Authors’ sentence is also conflicting: “This differential result t is similar to what we observed in TBC1D23 deficient iPSC-derived neurons and zebrafish.” is not clear, as in zebrafish TBC1D23 MO increases branching.

17.

What are the dots in graphs quantifying CaP length? How many CaP were counted per fish? A nested analysis should be considered if more cells per fish are counted.

Authors’ reply: We measured 3 to 4 CaP motoneuron axons at the same locations in each zebrafish, with each point representing the relative length of one axon.

Remarks: I do not quite understand the answer, here. If each dot represents one CaP, and more CaP in one fish were analysed, then a nested ANOVA/graph should be performed or authors should show the

mean of these 3 CaP they have analyzed, otherwise the graph is conceptually wrong. As the result could be driven by few fish having a major phenotype for instance with a clear clustering effect.

Minor points

2.

Methods are not detailed and in their current form do not fit the standards of the journal. For instance, imaging conditions with resolution, step size, speed etc... are not provided. Similarly, the algorithms used and image processing are not described. Articles should provide experimental details to improve reproducibility in the community, authors should amend accordingly all the methods.

Authors' reply: We apologize for not having explained methods more clearly in our previous submission.

We have revised our "methods" section with more detailed description

Remarks: The revised version of the methods is not satisfactory. Many basic details are still missing (see above remarks), which hinders reproducibility. This needs further improvement, and authors can refer to published Nat Commun papers for this.

3.

English should be improved in the methods section... e.g. line 906 "images...were taken", proper terminology and details should be used, such as "x,y scans were acquired atconfocal withresolution etc..".

Authors' reply: We have modified the methods section in the revised manuscript.

Remarks: Here again further improvement is necessary. Typos and bizarre sentences are still present. Typos: "Supplementary" repeated twice, line 174; ...Titles – figures legends: often the titles are not summarizing completely the content of the figure. As an example: Fig. 3 reads "Fig. 3 TBC1D23 preferentially regulates the phosphorylation of Golgi-localized proteins" but indeed a big part of the figures shows Golgi disassembly analysis upon Met. This should be included in the title.

6.

In Fig.2 expression of LKB1 pointed (vesicles?) after GS. Could authors comment this before line 266? A higher magnification, not saturated image should be provided (see also major point below concerning Pearson).

Authors' reply: We appreciate the reviewer's suggestions. We have repeated this experiment, and updated the manuscript with new Fig.2a. The pointed LKB1 might be endosomal and/or lysosomal LKB1.

Remarks: dotted expression should be addressed in the text with relevant references

16.

In the reporting summary it is claimed that N, degree of freedom etc are provided, but this reviewer failed in finding such info. Exact p- values are not reported.

Authors' reply: Exact p- values are included in the figures of the revised manuscript.

Remarks: as mentioned, authors still do not disclose always the N, degree of freedom etc and the p-values seem often too strong. I suggest to have a summary table describing all these values for each figure panel. This is what is expected for a Nat Commun paper.

Point-by-point responses:

We appreciate the reviewers' constructive suggestions that certainly helped us to improve the quality of our manuscript. To address the reviewers' concerns, we have performed several critical experiments and made extensive revisions to the revised manuscript. Our point-by-point responses are listed below:

REVIEWERS' COMMENTS

Reviewer #1 (Remarks to the Author):

The authors have sufficiently answered all questions and concerns. No further revisions are requested.

Response: We appreciate the positive comments from reviewer #1

Reviewer #2 (Remarks to the Author):

The authors have responded to this reviewers questions, and the proposed revisions are acceptable. However, on reinspection in the light of the comment of other reviewers, it appears that the sample size in individual experiments is unclear. A statement that experiments were performed at least in triplicate (p.42) is generally not sufficient. There are several bar graphs that do not show the data points, and where the true sample size is unknown. This information should be provided in the figure legends before acceptance.

Response: We appreciate the reviewer's suggestions. The sample size in individual experiments has been included in the figure legends in the revised manuscript. We also provided a supplementary table with all the statistical specifications for each figure panel within the revised manuscript. We have included data points for Fig. 8c.

Reviewer #3 (Remarks to the Author):

The authors have addressed many of the reviewers' comments. However, the following points still need to be addressed:

1. Blot reproduction and quantification: Blots that are used to make quantitative comparisons need to be quantified using densitometry of repeated blots (Fig. 1f, 1g, 1h, 1k, 2f, 3d, 3e, 3f, 4b, 4c, 4d, 5, 6b, 6c, 8 etc). As these blots have been done in triplicate, please include statistical error (eg. 0.61 ± 0.8).

Response: We have included bar graphs (data were represented as mean \pm SD) for blot quantification as suggested by the reviewer.

2. Lyso AMPK activity measurement: The use of lysosomal acidification by Lysosensor is not the same as quantifying AMPK activity at the lysosome. If lyso-ABKAR "did not work out for unknown reason", what about lyso-ExRai AMPKAR [1], cell fractionation followed by western [2,3], or

immunoblotting on lysosomes purified by LysoIP [4]?

Response: We appreciate the reviewer's thoughtful suggestions. We measured Lyso AMPK activity in WT and TBC1D23 KO HepG2 cells using lyso-ExRai AMPKAR. Consistent with our Western blot results (Fig.2f), TBC1D23 KO cells did not alter Lyso-AMPK activity, indicating that TBC1D23 specifically regulates Golgi-AMPK (new Fig.2 g,h).

3. Golgi assembly analysis: The description provided in lines 747-750 is still inadequate/unclear. First, the authors state that they "manually counting, both large and small" and then state they used "unbiased image analysis" to count fragments. Are fragments the same as elements? Manual counting vs. unbiased image analysis are contradictory statements – if one is manually counting this is not unbiased. If image analysis is used, please describe the filtering, segmentation, etc. settings used to identify objects. Furthermore, there is no detailing of what large or small objects are quantitatively defined as.

Response: We appreciate the reviewer's comment. Fragments are same as elements, and analysis is performed by manual counting. "unbiased image analysis" is improper, and has been replaced by "blind image analysis". To minimize the impact of manual counting, we also determined the ratio between the total area occupied by GM130 and the nucleus (also suggested by another reviewer), and obtained similar results (Fig. 3h and Fig. 6e).

4. Colocalization analysis (minor): Please clarify in the methods at line 750 which ImageJ tool was used (JaCOP, Coloc2, etc.) and the settings used within (PSF, Costes/Bisection, thresholded Pearson's or Mander's coefficients,etc.). If a custom Macro was used, briefly describe the steps/settings used.

Response: We have improved our description of our Methods as suggested by the reviewer. Pearson's correlation coefficients and Manders' coefficients were calculated using JACoP plug-in of ImageJ.

Reviewer #4 (Remarks to the Author):

Thank you for addressing my comments and adding new references. Reference #70 however, is not appropriate to the methodology executed in this manuscript. That paper does IMAC differently than what is reported in your manuscript, making it a bit confusing. It can be dropped or replaced by another reference.

Response: We have removed reference #70 as suggested.

Reviewer #5 (Remarks to the Author):

General remarks.

In the revised manuscript, authors attempted to address most of my concerns and answer to the remarks. The manuscript has been improved, and I believe the findings are conceptually valid. However, there are still major aspects that should be fixed along the concerns originally expressed

that are still required to strengthen the solidity of the mechanistic findings before the work can be considered as suitable for publication.

Globally, description and rationale of the experimental design and the experimental/analytic approaches for testing the working hypotheses are still confusing and not fully explained in the text. The manuscript still lacks basic details on measurements and acquisition parameters. Statistics is not fully described and many p-values honestly do not seem to match the observed variability. In general, I find the zebrafish data not technically and mechanistically informative to compensate for the shortcomings of the *in vitro* experiments. Only the morpholino approach was used for instance, and data on iPSCs do not actually match the results obtained *in vivo*. Such complexity which was not addressed originally and barely has been in the revised version of the work should be discussed, and the considerations related to brain development and disease should be toned down. All of this make the current version not compatible with the rigorous style of Nature Communications.

Response: We appreciate the reviewer's comments and constructive suggestions, which certainly helped us improve the quality of our manuscript. We have made extensive revisions, as suggested.

Major points:

1. Different KD and KO approaches, different cell types: authors seem to switch between cells silenced with siTBC1D23 and cells in which a KO was obtained by CRISPR-Cas9. Given the difference in KD ability, the use of one or the other approach should be addressed and discussed for each experiment. Same for the type of cells used.

Authors' reply: To determine how TBC1D23 deficiency affect AMPK activation, we initially used siRNA to interfere expression of TBC1D23. Subsequent cellular experiments were performed in TBC1D23 KO cells (HEK293T and HepG2), which were generated by CRISPR-Cas9. Following your suggestion, we examined effects of TBC1D23 knockdown on AMPK activation in response to energy stress. Our results indicated that shRNA-mediated TBC1D23 ablation also resulted in decreased AMPK activation (Graphic 3), similar to TBC1D23 KO by CRISPR-Cas9. Graphic 3 TBC1D23 deficiency results in compromised AMPK activation. HEK293T cells infected with virus encoding control shRNA (shNC) or shTBC1D23 were subjected to glucose starvation (GS) for the indicated time. Cells were collected and immunoblotted with antibodies indicated. For assays in human iPSC-derived neurons, shRNAs targeting human TBC1D23 were used to attenuate its expression. We tried CRISPR-Cas9 in iPSC-derived neurons. Due to low efficiency of virus infection (single vector for co-delivery of Cas9 and gRNAs), we failed to interfere TBC1D23 expression. Instead, TBC1D23 was knockdown by shRNA. We performed most of our experiments in HEK293T and HepG2 cells, and used HeLa and HUVEC cells to further prove our points. HEK293T cells are easily transfected; thus, we used HEK293T for MS analysis and pulldown/IP assays. In contrast with HeLa cells, HepG2 cells are LKB1 positive. Thus, we used HepG2 cells for IF and FRET assays. As it was reported that LKB1 is required for activation of AMPK in HUVEC cells, so we also examined effects of TBC1D23 knockdown on AMPK activation in HUVEC cells [1].

Remarks: I thank the authors for clarifying these points. However, they should explain the rationale of using different cell types for the different experiments performed (as they do at line 146/167 of

the revised manuscript). I cannot see this properly done in the rest of the revised version of the manuscript. Line 132-133 for instance: the authors do not specify in which cells they do the first silencing (si) experiments, but then refer to HepG2 for CRISPR-Cas9 soon after leaving the reader to wonder why did they use different strategies without explaining the rationale. Authors should explain each time within the text why they switched to HUVEC and HepG2. Also, authors should specify the type of cells used in “Methods” and in the figure legends for each of the panels. Also, it should be clear from the text that a subset of the results was corroborated in HUVEC cells.

Response: Thank you for your constructive suggestions. We have improved the description and rationale of the experimental design (main text and methods). Since it was reported that LKB1 is required for activation of AMPK in HUVEC cells (PMID: 18250273), we first examined the effects of TBC1D23 knockdown on AMPK activation in HUVEC cells (Line 129-131). Since LKB1-AMPK pathway responds to energy stress, we also chose the hepatocyte cell line HepG2 in subsequent experiments (Line 135-136). We also added a description indicating that the results obtained in HepG2 cells were consistent with HUVEC cells (Line 137-138).

2. In fig. 2, co-localization between LKB1 and Golgi marker: the development of super-resolution techniques has made it clear that co-localization analysis performed from standard microscopy are quite prone to errors and should be approached with caution. In these experiments, LKB1 fluorescence signal appear quite saturated (pixels saturation), which is one of the main issues leading to exaggerated and false results especially with Pearson 'correlation analysis. Authors should provide additional examples of images to see whether signal saturation is indeed an issue of the whole population of cells analyzed, if confirmed then this should be addressed for all the colocalization experiments. In addition, given a number of tools/algorithms available and the varying parameters to threshold the images beforehand authors should explain how the analysis was performed. I also suggest performing mild deconvolution on non-saturated images and it would be beneficial to examine other parameters such as Manders' coefficients. Similar comment for Fig. 6a,b and others. Here also again more cells or a larger field of view should also be provided.

Authors' reply: Thanks for your suggestions. We have repeated co-localization experiments carefully and use Manders' coefficients for Colocalization analysis in the revised manuscript (new Fig. 2a, 2b and 6a). A larger field of view is provided in the revised manuscript (new Fig. 2a, 2b, 3g, 6a, 6d).

Remarks: I can see that co-localization analysis was majorly improved. Manders' coefficients are normally 2 (M1 and M2) with respect to pixels in Image - 1 overlapping with Image-2 and vice versa. Authors can provide both, or specify the use of one. In the latter case, the label cannot be “Manders' coefficients” but rather “Manders' coefficient 1 or 2”. The manuscript also still lacks a detailed explanation of the method used. For instance, which algorithm was used in Image J? Did the author use masking of a certain region of the cell? Again, a larger field of view should be always shown and at least two cells per condition.

Response: We appreciate the reviewer's suggestions. The label for co-localization analysis has been corrected as “Manders' coefficient 2”. We also included “(Golgin-97/Flag-LKB1)” or “(Golgin-97/Flag-AMPK α 1)” in y axis, which indicates the specific information and is better than “Manders' coefficient 2”. We have improved our description of Methods as the reviewer suggested. Pearson's

correlation coefficients and Manders' coefficients were calculated using JACoP plug-in of ImageJ. We select intact cells for analysis. In the revised manuscript, a larger field of view (at least two cells suggested by the reviewer) has been shown (new Fig. 2b).

4. *The choice of the statistical analysis seems arbitrary. Authors appear to choose between t-test and analysis of variance without consistency. All the zebrafish experiments with more than two groups within one experiment are correctly assessed by ANOVA (F-test + post-hoc), while some of the cell-culture experiments with the same settings are analyzed via t-test. This approach even without any post-hoc correction leads to amplification of the type I error. The probability that the observed phenomenon is obtained by chance instead of underlying a biological meaning is therefore increasing. Such arbitrary choice in statistical assessment is poor data analysis quality and makes it hard to judge the conclusions overall. It should also be considered whether a 2-way ANOVA should be used in some occasions, for instance for the experimental set up in Fig. 1i where two variables exist: time and genetic condition. Therefore, authors should re-assess all the statistical analyses in the manuscript and clearly explain the choice made for the tests used. A List with all the experimental setups and tests, including analysis of normality would be ideal. This would make the claims of the study solid.*

Authors' reply: We are grateful to the reviewers' criticisms. Following your suggestion, we have checked our statistical analysis carefully. Statistical significance of the difference between two group was determined using Student's t test. And statistical significance of the difference between multiple groups, an ordinary one-way or two-way ANOVA (if two variables exist) was used, followed by Tukey's, Sidak's, or Dunnett's as indicated in the figure legends.

Remarks: Authors have improved the statistical analysis, overall. However, they should more clearly explain why a certain hypothesis test was used beyond citing t-test or ANOVA. Where the data checked for normality? Why did the authors choose Tukey vs Sidak etc ...as post-hoc test? Also in the "Reporting summary" authors claim to provide info such as "CI", "degrees of freedom"... though this reviewer did not find them reported in the manuscript. I suggest authors to make a supplementary table with all the statistical specs for each figure panel. Also, authors declare they provide "estimates of effect size" but I don't see this in the manuscript. Same for "one side, two sides".

Response: Statistical analyses were performed using GraphPad Prism 8 Software. No statistical method was used to predetermine the sample size. The data distribution was assumed to be normal, but this was not formally tested. The post-hoc test was recommended by GraphPad Prism software, according to the choice of multiple comparisons (compare each cell mean with the other cell mean in that row/column: Sidak; compare the mean of each column with the mean of every other column: Tukey). In the revised manuscript, the statistical significance of the difference between multiple groups (one variable), an ordinary one-way ANOVA was used, followed by Dunnett's test. And the statistically significant differences between multiple comparisons (if two variables exist) were analyzed using the two-way ANOVA, followed by Sidak's test. Differences were considered significant when $P < 0.05$. The statistical significance of the difference between two group was determined using unpaired two-tailed t test. We have checked the "Reporting summary" carefully. We have improved the description of the analytic approaches and provided a supplementary table

with all the statistical specs for each figure panel.

5. The number of replicates very often is not given (see WB and IP), as well as quantification. The authors should always provide the number of replicates for each experiment as requested and also the quantification including mean and SEM and statistical assessment.

Authors' reply: We apologize for not having explained this more clearly in our original submission. All experiments are performed at least in triplicate, and we have added this statement and quantification in section entitled "statistics and reproducibility" in Methods.

Remarks: N should be given for each experiment in the figure panel. When analyzing co-localization or Golgi elements from each cell, how many cells per experiment were counted? Do the dots in the graphs represent pooling of cells from different biological replicates? or different fields? Such aspects are not explained, however they are flagged in the Reporting Summary.

Response: We apologize for not having explained this more clearly in our previous submission. We have included bar graphs (data were represented as mean \pm SD) for blot quantification in the revised manuscript. The information about the sample size, dots, and the number of replicates in individual experiments has been provided in the figure legends.

6. In general, more cells/large fields of view should be shown for each experiment, methods for image acquisition and analysis should be better described. For FRET imaging the method used to do imaging (sensitized emission?) wavelengths used and laser power as well as ratiometric analysis should be described.

Authors' reply: Following your suggestions, we have revised our figures with a larger field of view and updated methods with better description. For FRET imaging, "The images were acquired using a Leica DMI6000B total internal reflection fluorescence microscope (Leica) equipped with BP420/10 excitation filter, a 440/520 dichroic mirror and two emission filters (BP472/30 for cyan fluorescent protein (CFP) and BP542/27 for YFP)" changed as: "The imaging experiments were performed using a Leica DMI6000B total internal reflection fluorescence microscope (Leica), equipped with mercury lamp as laser power. The images were acquired with BP420/10 excitation filter, a 440/520 dichroic mirror and two emission filters (BP472/30 for cyan fluorescent protein (CFP) and BP542/27 for YFP)". BP420/10 indicates the filter detects the wavelength from 415 nm to 425 nm. To indicate the ratiometric analysis, we have corrected the y axis of Fig.2d and e as "AMPK activity (Normalized FRET/CFP)".

Remarks: what does "Normalized FRET/CFP" mean? Normalized vs what? Line 1296: "The FRET/CFP ratio was measured and normalized to cells incubated with DMEM" The authors should explain how was this normalization performed? What does it mean? Authors should really make an effort to use precise terminology and detail the analysis they performed such that readers are in condition to assess accuracy as well as to support reproducibility. Imaging and analysis using cells could further benefit of improvement in the description, along the lines of the details provided for zebrafish, see also the other points. A statement whether images and analysis within a certain

experiment were obtained/performed with the same parameters with respect to acquisition and post-processing should be added, or otherwise explained if that is not the case.

Response: We have improved the description of the experimental/analytic approaches. The FRET/CFP ratio of WT cells incubated with DMEM was set as 1, and FRET/CFP ratio of other groups was normalized to WT cells incubated with DMEM. For all cellular experiments, image acquisition and analysis within a certain set of experiment were obtained/performed with the same parameters. For zebrafish experiments, we use automatic exposure for each zebrafish to capture higher quality images. Image analysis within a certain experiment were performed with the same parameters. We have included these statements in Methods as the reviewer suggested.

7. In the FRET experiment again, When providing LKB1 in HeLa cells (LKB1^{-/-}) why authors do not show the FRET signal increase correlating to AMPK activation?

Authors' reply: We appreciate the reviewer's thoughtful suggestions. Our LKB1 has a GFP tag when introduced in HeLa cells. As the wavelength of GFP (excitation: 488 nm, emission: 507 nm) is close to that of CFP (excitation: 405 nm, emission: 485 nm), the FRET experiment would be very difficult. Thus, we determined AMPK activation at the whole cell level. We agree that FRET will provide more specific information at the organelle level.

Remarks: This limitation could be surpassed by using a LKB1 without tag in your transfection experiment. Using FRET to test this hypothesis would be more consistent, appropriate and elegant. WB per se would corroborate the finding. Given the specificity of the FRET AMPK sensor to Golgi or mitochondria, one would also expect to see the actual images of the FRET signal within cells. Why don't the authors show the images used to make calculations? (FRET and CFP channel and the computed ratiometric image)? It is somewhat bizarre that quantification is reported without representative images of the results. Why is now the scale bar and numbers of the graph (Fig. 2d etc) different from that one submitted originally?

The authors claim that there is a strong reduction of AMPK activity in Golgi in cells depleted of TBC1D23 in both basal and energy stress condition by looking at the FRET data. However, if one examines carefully the graphs presented now, the data show that there is a little decrease of activity in basal conditions (1 vs 0.7, I guess), and that there is still a small -yes negligible- increase in activity upon stress also in TBC1D23 depleted cells. Authors should discuss these results more extensively along these lines and perhaps show a fold change quantification.

Response: The instrument we used for the FRET assay (Leica DMI6000B total internal reflection fluorescence microscope) served our institute for a very long time and was phased out by our institute about one year ago. Although the raw images are available, we were unable to process the images as suggested due to lack of the original software. We failed too when we tried to open the images with third party software, such as Image J. To measure the AMPK activity at Golgi after reintroduction of LKB1 in HeLa cells, we had to take a different approach and measure the activation of AMPK at the Golgi by isolating Golgi fractionations and assessing changes by immunoblotting. As shown in new Supplementary Fig. 2d, introduction of LKB1 in HeLa cells remarkably promoted AMPK activation in Golgi fractions upon AICAR treatment.

The numbers of the graph in Fig. 2d remain unchanged, and this panel does not contain a scale bar.

Following the reviewer's suggestion (comment 14), we have removed iPSC cell-associated data (old Fig.6d and Fig.8a-c.). And following the reviewer's suggestion (comment 4), statistical significance of the difference between multiple groups was determined by ANOVA. Since we have confirmed that TBC1D23 KO does not affect Golgi assembly under basal conditions (Fig.3g-h), and we aimed to compare the rescue ability of LKB1, Golgi-targeted LKB1 (LKB1-Giantin) WT and its kinase-dead mutant after metformin treatment. Thus, indicated groups in Fig.6d were included in the revised manuscript. Since the reviewer suggested a larger field of view, we updated our images to include more cells and added a scale bar.

We have revised our discussion about "Golgi-AMPK activity reduction in cells depleted of TBC1D23 in both basal and energy stress condition" as the reviewer suggested (Line 165-168).

8. *Quantification of the images in the WB should also been shown (Fig.S2). Why do authors choose AICAR instead of GS stimulus as opposed to the experiment with TBC1D23 KO cells they want to directly compare to? This should be explained. Also, the number of replicates for all these experiments is not clear (not given). This is a general comment.*

Authors' reply: We appreciate the reviewer's suggestions. We have quantified the western blot intensity and the quantitative data have been included in the revised manuscript (new supplementary Fig.2). In supplementary Fig.2b, glucose starvation was chosen. AICAR, a widely used AMPK activator, which was reported to cause the phosphorylation of GBF1 at T337 and subsequent Golgi disassembly [2]. We sought to determine the role of LKB1 in Golgi-AMPK activation, so we treated cells with AICAR. We have included the number of replicates for all experiments in section entitled "statistics and reproducibility" in Methods.

Remarks: I only see FRET/CFP quantification but not WB quantification. The reason to use AICAR provided here should be included in the text. N of replicates from which quantifications are performed should be specifically provided in figure legends.

Response: We have included bar graphs (data were represented as mean \pm SD) for immunoblot quantification in the revised manuscript, and the information about the number of replicates has been provided in the figure legends.

9. *In Fig.3g/h and Fig6e it is not clear how "Golgi elements" were counted. Please specify, as the term "elements" is vague. Did the authors count large and small objects observed in the image? The occurrence of GM130 patches different in size might underly different patterns of fragmentation. More examples of cells should be provided, only 2 or 3 cells are shown now.*

Authors' reply: We apologize for not making it clear in the original submission. The analysis is performed by manual counting, both large and small objects being counted. We have included "Golgi assembly analysis" in Methods (Cell culture, transfection and immunofluorescence). And more examples of cells are provided in the revised manuscript.

Remarks: Basic details of image analysis are missing still. What do the authors mean now by "unbiased image analysis"? Do they mean "blind"? A better analysis should also consider the

nucleus dimension, a ratio between the total area occupied by GM130 and the nucleus are should be added, which is less prone to errors as compared to manually counting fragments.

Response: We have checked our manuscript carefully and improved the description the analytic approaches. We indeed mean “blind”, which has been corrected in the revised manuscript. And Golgi assembly analysis using the ratio between the total area occupied by GM130 and the nucleus has been added in the revised manuscript (Fig. 3h and Fig. 6e).

10. From the images shown, Met seems to have an intermediate effect on Golgi disassembly in KO cells, as also the statistics (should be an ANOVA, see below) would suggest and not what the authors claim, or what do they mean that the disassembly was “impaired”? In Fig. 6d authors should show more cells and higher magnification, so far the rescue in Golgi fragmentation obtained by LKB1-G is hard to judge as only 1 cell is shown. Also, for the condition co-transfected with LKB1 GM130 seems fainter (why?).

Authors' reply: In the revised manuscript, the statistical significance of the difference is determined by one-way ANOVA, followed by Tukey's test. Phosphorylation of GBF1 by AMPK was reported to promote Golgi disassembly[2]. Consistent with impaired AMPK activation of TBC1D23 KO cells, Golgi disassembly of KO cells treated with metformin is also significantly decreased, in comparison with WT cells. Fig. 6d has been updated with more cells and higher magnification. As shown in new Fig. 6d, GM130 intensity in cells transfected with LKB1 is comparable to the adjacent cells.

Remarks: In Fig. 6d, the field still only one cell is shown for the LKB1 and rescue experiment so it is hard to judge. Again, the GM130 channel is too low and the quality of the image seems worst compared to submitted. Also, the authors should paid attention to the phrasing and English. For instance, line 232 and 1117: “knockout of TBC1D23 strongly impairs Golgi disassembly” seems strange. Most likely authors wanted to say that TBC1D23 KO prevents Golgi disassembly upon Met treatment.

Response: Thank you for pointing this out. We have corrected “impairs” as “prevents” as the reviewer suggested. In Fig.6d, two cells were shown for LKB1, and the images were the same as the original submission. We have replaced the images with TIF files with higher resolution.

11. The microscopy images showing zebrafish brain in the transgenic fish are of poor quality, with intense signal bleed from planes other of the one in focus. Authors should specify whether these are single plane images of z-stacks. Probably an epifluorescence microscope was used. Also, they should state the parameters used for each condition/fish. Did the same laser intensity was used? The authors should at least indicate different brain sub-domains for general audience not familiar with zebrafish brain. Clearly, a major involvement of the whole brain is seen in the morphants. Why do authors choose to report only midbrain measurements? Is this correlated to the PCH? This should be explained. What is the evidence for cerebellum itself, or hindbrain beyond r2 and other regions? In case regions others than the midbrain did not show any statistically significant difference this should be reported, and the specific involvement in midbrain should be discussed.

Authors' reply: We are grateful to the reviewer's constructive suggestions. Following your suggestions, we have improved the quality of zebrafish brain pictures by using Olympus IX83 P2ZF

spinning disk confocal microscopy, and presented as maximum intensity z-stack projections (new Fig.7g and Fig.8f). Due to individual differences in zebrafish, the fluorescence brightness of each zebrafish is not consistent. We characterized midbrain size by measuring relative area, and fluorescence intensity did not affect our measurements. Therefore, we use automatic exposure for each zebrafish to capture higher quality images. As you mentioned, the development of the whole brain is affected in our morphants, including cerebellum, hindbrain beyond r2 and other brain regions. We chose the size of the midbrain to characterize the brain abnormalities, and the disruption of midbrain appear to be consistent with the midbrain defects observed with brain imaging of human patients. Besides, midbrain abnormalities have been reported in several articles in previous studies of PCH disease [3-6].

Remarks: I see the brain images improved. Still, if authors used different laser energy or parameters because of the difference in fluorescence, they should state the specific setting in the Methods. In Fig.8g they should explain what exactly “Relative midbrain size” means. Is it an area relative to what and how was it calculated? (again should be added in “Methods”). With respect to different brain regions, authors should clearly describe that the effect is observed on many different regions regardless of the association with human patients in PCH... otherwise, this is the classical cherry-picking attitude in selecting the data to report. The impact of these LoF on the whole brain could be functionally important for brain development understanding and might underly differences across vertebrates

Response: We appreciate the reviewer’s comment. We have improved our description of Methods as reviewer suggested and included information about exposure times in the Methods. We characterized midbrain size by measuring relative area, and fluorescence intensity did not affect our measurements. In lateral views of zebrafish, the midbrain size quantified by ZEN 3.1 software was defined as relative midbrain size. We indeed observed that the development of the whole brain is affected in morphants, including midbrain, cerebellum and hindbrain.. These descriptions have been included in our revised manuscript.

12. *Neuronal defects: Fig.7 and S6, authors claim to use iPSC-derived neurons and to test the efficacy of different ShRNA to silence TBC1D23. However, in Fig.S6a they show western blot performed on HEK cells (?). How does this compare to the effect in IPSC cells? This is completely hidden within the text and only appears to an attentive reader. Please revise. Similarly to the “si”, “KO” and “sg” nomenclature story, authors should state clearly why they use different systems in different contexts and cannot rely on different cells to validate efficacy of the approach then used in a different cell type.*

Authors’ reply: The infection efficiency of iPSC-derived neurons is low, so it would be difficult to determine the knockdown efficiency of shRNA. It is common to determine knockdown efficiency in cell lines, including HEK293T[7-9]. Since these shRNA target human TBC1D23, then the effective shRNA in HEK293T cells will work in human iPSC cells when delivered successfully. We then chose the neurons successfully infected (GFP positive) for morphology analysis. We have corrected the issues about nomenclature in the revised manuscript.

Remarks: I understand the technical issue, but this kind of approximation is not acceptable, and

could be solved by sorting the positive cells and showing the reduced TBC1D23 expression. If authors want to keep the iPSC cell data, they should better show the efficiency of reduction from these cells. Also, the differentiation in cortical neurons is not specified and should be described. Importantly, I am concerned about the speculative interpretation the authors make based on the iPSC poor quality data -see my point below.

Response: We have removed iPSC cell-associated data.

14. Primary branching in Fig.7 and S6: authors claim that a decreased primary branching is observed in cells depleted of TBC1D23. First, control cells (without Sh) are not shown and should be shown directly in main figure. One should be able to judge the “rescue” of the branching comparing directly the conditions + LBK1-G etc to control scenario, statistically.

Authors' reply: Thanks a lot for the suggestions. Neuronal morphology of control neurons (shNC), shTBC1D23 neurons, and shTBC1D23 neurons with ectopic expression of indicated proteins are shown together. For better layout of the figures, the representative images are shown in new supplementary Fig.6c, and the statistical analysis data are shown in main figure (new Fig.7a-c).

Remarks: The are major concerns here, which I summarize: images of iPSC cells should be shown together with graphs and not in separated figures; they are really poor in quality and again only single/few cortical neurons are shown. A higher number of cells should be shown. The panel seem to be cut at the level where branches continue so it is not possible to really judge about branching. The graph indicates that controls cells have something like 4 primary branching on average when in fact the only cell shown exhibit one long branch. How can we judge about reduction here? All the other panels showing 1 or 2 cells demonstrate similar branching number. Overall, the dots in the graph are too thick, this should be improved in all the graphs such to be able to see the data and statistics seems really strange (p-values too good?).

Response: We have removed iPSC cell-associated data.

15. Authors claim that neuronal defects observed here confirm previous reports. However, in the Huang et al, PNAS 2019 the authors showed the opposite in terms of branching. Reduced TBC1D23 in zebrafish induced increased branching (not decreased) in the previous report. Why does a similar approach reduce branching now? How is this “confirming” previous reports? Indeed, in the current work authors clearly show representative images of CaP motoneurons with hyperbranching (Fig. 7e) confirming their previous report. Why do iPSC cells show a opposite phenotype? Are those differentiated in motoneurons or what? Authors should discuss this. Instead authors say that the data are “nicely consistent with our results in iPSC..” Branching seems again increased in LBK1 MO model that should be addressed.

Authors' reply: We apologize for have explained this more clearly in our original submission. The effects of TBC1D23 deficiency on neurite length was consistent with the observation in Neuro2a neuroblastoma cells. iPSC-derived cells are cortical neurons. CaP motoneurons is a primary motor neuron part of the spinal cord. The axon of the CaP motoneuron extends ventrally from the ventral root, within the space between the notochord and the medial surface of the axial muscles. The analyzed branch of iPSC-derived neurons locates at the proximity neurite, while branch of CaP

motoneurons is the distal neurite region. TBC1D23 might have differential role in neuronal branch at proximal and distal neurite. We discussed the discrepancy regarding the effects of TBC1D23 deficiency on branch, and we assumed that it might be due to different types of neuron and stage of neuronal differentiation. TBC1D23 deficiency caused abnormal branch in both iPSC-derived neurons and zebrafish, in comparison with control groups, indicating the essential role TBC1D23 in maintaining normal neuronal development. LKB1 MO caused increased branch in CaP motoneurons of zebrafish, similar to TBC1D23 MO, which is consistent our model that TBC1D23 and cooperate in regulating neuronal development. However, LKB1 knockdown result in significantly less branched axons in cortical neurons after 5 day of culture in vitro (DIV). This differential result t is similar to what we observed in TBC1D23 deficient iPSC-derived neurons and zebrafish.

Remarks: I am familiar with neuronal branching. The explanation offered here is a pure speculation and I do not advice to have it in the main manuscript. Simply authors are comparing different cell types in different species having opposite results. On top of this, primary branching analysis was not performed in CaP motoneurons to be able to actual compare the same thing in different cell types. Also how does then the “longest neurite” in IPSC compare with neuritogenesis in CaP from fish? I do not think the data are strong enough to prove any specific involvement of the genes of interest in the development and neuritogenesis of both cell types and are conflicting. I propose to refrain the use of iPSC cells unless a better analysis is performed coupled to an improved discussion of the effects seen. Authors’ sentence is also conficting: “This differential result t is similar to what we observed in TBC1D23 deficient iPSC-derived neurons and zebrafish.” is not clear, as in zebrafish TBC1D23 MO increases branching.

Response: We have removed iPSC cell-associated data.

17. *What are the dots in graphs quantifying CaP length? How many CaP were counted per fish? A nested analysis should be considered if more cells per fish are counted.*

Authors’ reply: We measured 3 to 4 CaP motoneuron axons at the same locations in each zebrafish, with each point representing the relative length of one axon.

Remarks: I do not quite understand the answer, here. If each dot represents one CaP, and more CaP in one fish were analysed, then a nested ANOVA/graph should be performed or authors should show the mean of these 3 CaP they have analyzed, otherwise the graph is conceptually wrong. As the result could be driven by few fish having a major phenotype for instance with a clear clustering effect.

Response: We showed the mean of 3 CaP in each zebrafish that we have analyzed in the revised manuscript (new Fig.7b and 8e).

Minor points

2. *Methods are not detailed and in their current form do not fit the standards of the journal. For instance, imaging conditions with resolution, step size, speed etc... are not provided. Similarly, the algorithms used and image processing are not described. Articles should provide experimental details to improve reproducibility in the community, authors should amend accordingly all the*

methods.

Authors' reply: We apologize for not having explained methods more clearly in our previous submission. We have revised our "methods" section with more detailed description

Remarks: The revised version of the methods is not satisfactory. Many basic details are still missing (see above remarks), which hinders reproducibility. This needs further improvement, and authors can refer to published Nat Commun papers for this.

Response: We have checked the manuscript carefully and improved our description of Methods as suggested by the reviewer.

3. English should be improved in the methods section... e.g. line 906 "images...were taken", proper terminology and details should be used, such as "x,y scans were acquired atconfocal withresolution etc..".

Authors' reply: We have modified the methods section in the revised manuscript.

Remarks: Here again further improvement is necessary. Typos and bizarre sentences are still present. Typos: "Supplementary" repeated twice, line 174; ...Titles – figures legends: often the titles are not summarizing completely the content of the figure. As an example: Fig. 3 reads "Fig. 3 TBC1D23 preferentially regulates the phosphorylation of Golgi-localized proteins" but indeed a big part of the figures shows Golgi disassembly analysis upon Met. This should be included in the title.

Response: We have improved the description of the experimental approaches, and provided more detailed information. As for "Titles – Figures Legends", all panels in Fig. 3 are related with the current title of the Figure Legend. Data in panels g&h are not directly related, but are an extension of data in other panels. We have revised the title for Fig. 3 as the reviewer suggested.

6. In Fig.2 expression of LKB1 pointed (vesicles?) after GS. Could authors comment this before line 266? A higher magnification, not saturated image should be provided (see also major point below concerning Pearson).

Authors' reply: We appreciate the reviewer's suggestions. We have repeated this experiment, and updated the manuscript with new Fig.2a. The pointed LKB1 might be endosomal and/or lysosomal LKB1.

Remarks: dotted expression should be addressed in the text with relevant references

Response: We have revised the main text as the reviewer suggested.

16. In the reporting summary it is claimed that N, degree of freedom etc are provided, but this reviewer failed in finding such info. Exact p- values are not reported.

Authors' reply: Exact p- values are included in the figures of the revised manuscript.

Remarks: as mentioned, authors still do not disclose always the N, degree of freedom etc and the p-

values seem often too strong. I suggest to have a summary table describing all these values for each figure panel. This is what is expected for a Nat Commun paper.

Response: The information about the sample size and the number of replicates in individual experiments has been provided in the Figure Legends. We provided a supplementary table with all the statistical specs for each figure panel in the revised manuscript.

REVIEWER COMMENTS

Reviewer #3 (Remarks to the Author):

Reviewers' concerns have been appropriately addressed.

Reviewer #5 (Remarks to the Author):

(1)

Remarks: I thank the authors for clarifying these points. However, they should explain the rationale of using different cell types for the different experiments performed (as they do at line 146/167 of the revised manuscript). I cannot see this properly done in the rest of the revised version of the manuscript. Line 132-133 for instance: the authors do not specify in which cells they do the first silencing (si) experiments, but then refer to HepG2 for CRISPR-Cas9 soon after leaving the reader to wonder why did they use different strategies without explaining the rationale. Authors should explain each time within the text why they switched to HUVEC and HepG2. Also, authors should specify the type of cells used in "Methods" and in the figure legends for each of the panels. Also, it should be clear from the text that a subset of the results was corroborated in HUVEC cells.

Response: Thank you for your constructive suggestions. We have improved the description and rationale of the experimental design (main text and methods). Since it was reported that LKB1 is required for activation of AMPK in HUVEC cells (PMID: 18250273), we first examined the effects of TBC1D23 knockdown on AMPK activation in HUVEC cells (Line 129-131). Since LKB1-AMPK pathway responds to energy stress, we also chose the hepatocyte cell line HepG2 in subsequent experiments (Line 135-136). We also added a description indicating that the results obtained in HepG2 cells were consistent with HUVEC cells (Line 137-138).

R5 new remark: I recognize that this aspect was improved.

(2)

Remarks: I can see that co-localization analysis was majorly improved. Manders' coefficients are normally 2 (M1 and M2) with respect to pixels in Image-1 overlapping with Image-2 and vice versa. Authors can provide both, or specify the use of one. In the latter case, the label cannot be "Manders' coefficients" but rather "Manders' coefficient 1 or 2". The manuscript also still lacks a detailed explanation of the method

used. For instance, which algorithm was used in Image J? Did the author use masking of a certain region of the cell? Again, a larger field of view should be always shown and at least two cells per condition.

Response: We appreciate the reviewer's suggestions. The label for co-localization analysis has been corrected as "Manders' coefficient 2". We also included "(Golgin-97/Flag-LKB1)" or "(Golgin-97/Flag-AMPK α 1)" in y axis, which indicates the specific information and is better than "Manders' coefficient 2". We have improved our description of Methods as the reviewer suggested. Pearson's correlation coefficients and Manders' coefficients were calculated using JACoP plug-in of ImageJ. We select intact cells for analysis. In the revised manuscript, a larger field of view (at least two cells suggested by the reviewer) has been shown (new Fig.2b).

R5 new remark: I recognize that this aspect was improved.

(3)

Remarks: Authors have improved the statistical analysis, overall. However, they should more clearly explain why a certain hypothesis test was used beyond citing t-test or ANOVA. Where the data checked for normality? Why did the authors choose Tukey vs Sidak etc ...as post-hoc test? Also in the "Reporting summary" authors claim to provide info such as "CI", "degrees of freedom"... though this reviewer did not find them reported in the manuscript. I suggest authors to make a supplementary table with all the statistical specs for each figure panel. Also, authors declare they provide "estimates of effect size" but I don't see this in the manuscript. Same for "one side, two sides".

Response: Statistical analyses were performed using GraphPad Prism 8 Software. No statistical method was used to predetermine the sample size. The data distribution was assumed to be normal, but this was not formally tested. The post-hoc test was recommended by GraphPad Prism software, according to the choice of multiple comparisons (compare each cell mean with the other cell mean in that row/column: Sidak; compare the mean of each column with the mean of every other column: Tukey). In the revised manuscript, the statistical significance of the difference between multiple groups (one variable), an ordinary one-way ANOVA was used, followed by Dunnett's test. And the statistically significant differences between multiple comparisons (if two variables exist) were analyzed using the two-way ANOVA, followed by Sidak's test. Differences were considered significant when $P < 0.05$. The statistical significance of the difference between two groups was determined using unpaired two-tailed t test. We have checked the "Reporting summary" carefully. We have improved the description of the analytic approaches and provided a supplementary table with all the statistical specs for each figure panel.

R5 new remark: I see the improvement, and I find the new table useful. However, still if no sample size was estimated, then "Estimates of effect sizes" should not be selected in the Reporting summary.

Normality should be tested for each dataset in order to use the appropriate test (parametric and not parametric). I ask the authors to include these checks.

(4)

Remarks: N should be given for each experiment in the figure panel. When analyzing co-localization or Golgi elements from each cell, how many cells per experiment were counted? Do the dots in the graphs represent pooling of cells from different biological replicates? or different fields? Such aspects are not explained, however they are flagged in the Reporting Summary.

Response: We apologize for not having explained this more clearly in our previous submission. We have included bar graphs (data were represented as mean \pm SD) for blot quantification in the revised manuscript. The information about the sample size, dots, and the number of replicates in individual experiments has been provided in the figure legends.

R5 new remark: I see the improvement. When the authors say “similar results were obtained in independent experiments” or “experiments were performed in triplicate (for instance Fig.2a), was the quantification done only on one replicate? (for instance: line 1076: n.of cells n=55 is it pooling cells from one replicate? Or three? If this is pooling from three, then a super-plot should be considered or at least a fold change of the different replicates).

(5)

Remarks: what does “Normalized FRET/CFP” mean? Normalized vs what? Line 1296: “The FRET/CFP ratio was measured and normalized to cells incubated with DMEM” The authors should explain how was this normalization performed? What does it mean? Authors should really make an effort to use precise terminology and detail the analysis they performed such that readers are in condition to assess accuracy as well as to support reproducibility. Imaging and analysis using cells could further benefit of improvement in the description, along the lines of the details provided for zebrafish, see also the other points. A statement whether images and analysis within a certain experiment were obtained/performed with the same parameters with respect to acquisition and post-processing should be added, or otherwise explained if that is not the case.

Response: We have improved the description of the experimental/analytic approaches. The FRET/CFP ratio of WT cells incubated with DMEM was set as 1, and FRET/CFP ratio of other groups was normalized to WT cells incubated with DMEM. For all cellular experiments, image acquisition and analysis within a certain set of experiment were obtained/performed with the same parameters. For zebrafish experiments, we use automatic exposure for each zebrafish to capture higher quality images. Image analysis within a certain experiment were performed with the same parameters. We have included these statements in Methods as the reviewer suggested.

R5 new remark: Fig.2d: I still do not understand, I do not see any normalization. All the values plotted for WT are not 1, can authors also provides the raw measures and/or the calculation. How was intensity analyzed? Authors should show the ratiometric image in addition to this “normalized” quantification. This is important given that authors use a band-filter method not in a confocal setting, instead of a more accurate spectral unmixing methodology

(6)

Remarks: This limitation could be surpassed by using a LKB1 without tag in your transfection experiment. Using FRET to test this hypothesis would be more consistent, appropriate and elegant. WB per se would corroborate the finding. Given the specificity of the FRET AMPK sensor to Golgi or mitochondria, one would also expect to see the actual images of the FRET signal within cells. Why don't the authors show the images used to make calculations? (FRET and CFP channel and the computed ratiometric image)? It is somewhat bizarre that quantification is reported without representative images of the results. Why is now the scale bar and numbers of the graph (Fig.2d etc) different from that one submitted originally?

The authors claim that there is a strong reduction of AMPK activity in Golgi in cells depleted of TBC1D23 in both basal and energy stress condition by looking at the FRET data. However, if one examines carefully the graphs presented now, the data show that there is a little decrease of activity in basal conditions (1 vs 0.7, I guess), and that there is still a small -yes negligible-increase in activity upon stress also in TBC1D23 depleted cells. Authors should discuss these results more extensively along these lines and perhaps show a fold change quantification.

Response: The instrument we used for the FRET assay (Leica DMI6000B total internal reflection fluorescence microscope) served our institute for a very long time and was phased out by our institute about one year ago. Although the raw images are available, we were unable to process the images as suggested due to lack of the original software. We failed too when we tried to open the images with third party software, such as Image J. To measure the AMPK activity at Golgi after reintroduction of LKB1 in HeLa cells, we had to take a different approach and measure the activation of AMPK at the Golgi by isolating Golgi fractionations and assessing changes by immunoblotting. As shown in new Supplementary Fig. 2d, introduction of LKB1 in HeLa cells remarkably promoted AMPK activation in Golgi fractions upon AICAR treatment.

The numbers of the graph in Fig.2d remain unchanged, and this panel does not contain a scale bar. Following the reviewer's suggestion (comment 14), we have removed iPSC cell-associated data (old Fig.6d and Fig.8a-c.). And following the reviewer's suggestion (comment 4), statistical significance of the difference between multiple groups was determined by ANOVA. Since we have confirmed that TBC1D23 KO does not affect Golgi assembly under basal conditions (Fig.3g-h), and we aimed to compare the rescue ability of LKB1, Golgi-targeted LKB1 (LKB1-Giantin) WT and its kinase-dead mutant after metformin treatment. Thus, indicated groups in Fig.6d were included in the revised manuscript. Since the

reviewer suggested a larger field of view, we updated our images to include more cells and added a scale bar.

We have revised our discussion about “Golgi-AMPK activity reduction in cells depleted of TBC1D23 in both basal and energy stress condition” as the reviewer suggested (Line165-168).

R5 new remark: I understand that there are circumstances where the raw data can be missing, but I do not think it should be acceptable for a journal such as Nature Communications with the principle of FAIR data to have FRET ratiometric measurements performed with a simple method of filter-based imaging at epifluorescence microscope without being able ever and upon request to show the representative microscopy images. If authors can provide the same assessment in western blot I recommend to remove the FRET data from the paper. Also, I see that the authors now show Golgi-AMPK activity reduction in cells depleted of TBC1D23 in both basal and energy stress condition, but on the other hand they should still discuss the significant increase in AMPK activity shown by FRET from basal to GS even in the absence of TBC1D23 (my original question), what other mechanisms are in place then?

(7)

Remarks: I only see FRET/CFP quantification but not WB quantification. The reason to use AICAR provided here should be included in the text. N of replicates from which quantifications are performed should be specifically provided in figure legends.

Response: We have included bar graphs (data were represented as mean \pm SD) for immunoblot quantification in the revised manuscript, and the information about the number of replicates has been provided in the figure legends.

R5 new remark: The increase in AMPK activation upon AICAR treatment of cells expressing LBK1 should be better explained. Activation can be observed also without AICAR in cells expressing LBK1. Fig.2d has no quantification.

(8)

Remarks: Basic details of image analysis are missing still. What do the authors mean now by “unbiased image analysis”? Do they mean “blind”? A better analysis should also consider the nucleus dimension, a ratio between the total area occupied by GM130 and the nucleus are should be added, which is less prone to errors as compared to manually counting fragments.

Response: We have checked our manuscript carefully and improved the description the analytic approaches. We indeed mean “blind”, which has been corrected in the revised manuscript. And Golgi

assembly analysis using the ratio between the total area occupied by GM130 and the nucleus has been added in the revised manuscript(Fig. 3h and Fig.6e).

R5 new remark: I recognize that this aspect was improved.

(9)

Remarks: In Fig.6d, the field still only one cell is shown for the LKB1 and rescue experiment so it is hard to judge. Again, the GM130 channel is too low and the quality of the image seems worst compared to submitted. Also, the authors should paid attention to the phrasing and English. For instance, line 232 and 1117: “knockout of TBC1D23 strongly impairs Golgi disassembly” seems strange. Most likely authors wanted to say that TBC1D23 KO prevents Golgi disassembly upon Met treatment.

Response: Thank you for pointing this out. We have corrected “impairs” as “prevents” as the reviewer suggested. In Fig.6d, two cells were shown for LKB1, and the images were the same as the original submission. We have replaced the images with TIF files with higher resolution.

R5 new remark: I recognize that this aspect was improved.

(10)

Remarks: I see the brain images improved. Still,if authors used different laser energy or parameters because of the difference in fluorescence, they should state the specific setting in the Methods. In Fig.8g they should explain what exactly “Relative midbrain size” means. Is it an area relative to what and how was it calculated? (again should be added in “Methods”). With respect to different brain regions, authors should clearly describe that the effect is observed on many different regions regardless of the association with human patients in PCH... otherwise, this is the classical cherry-picking attitude in selecting the data to report. The impact of these LoF on the whole brain could be functionally important for brain development understanding and might underly differences across vertebrates

Response: We appreciate the reviewer’s comment. We have improved our description of Methods as reviewer suggested and included information about exposure times in the Methods. We characterized midbrain size by measuring relative area, and fluorescence intensity did not affect our measurements. In lateral views of zebrafish, the midbrain size quantified by ZEN3.1 software was defined as relative midbrain size. We indeed observed that the development of the whole brain is affected in morphants, including midbrain, cerebellum and hindbrain. These descriptions have been included in our revised manuscript.

R5 new remark: ok, I see the improvements but authors must explain what “relative” means (relative to what?) isn’t an absolute measurement? If yes please remove “relative”. Relative means it is normalized to something (body area? Head area? Whole brain area?). Also maybe authors have to specify that they analyze “dorsal midbrain/optic tectum” area and not the whole midbrain (judging from the ROI designed) Also the graph should definitely provide the measurement unit (μm^2 ?). Line 379: why did you chose “midbrain”? The sentence is too bold, other areas are interested as well. So authors should claim they use “midbrain” as a proxy of brain underdevelopment in general. Why did the author not show analysis in the cerebellum, forebrain? affected in the disease. Authors should rephrase : “PCH-like phenotypes in zebrafish” in abstract and whenever they talk about it. Loss of TBC1D23 also causes impaired cortical development, right? The phenotype observed is not PCH-like or we could not tell, it is a neurodevelopmental phenotype involving different brain areas in fish at best.

(11)

Remarks: I understand the technical issue, but this kind of approximation is not acceptable, and could be solved by sorting the positive cells and showing the reduced TBC1D23 expression. If authors want to keep the iPSC cell data, they should better show the efficiency of reduction from these cells. Also, the differentiation in cortical neurons is not specified and should be described. Importantly, I am concerned about the speculative interpretation the authors make based on the iPSC poor quality data -see my point below.

Response: We have removed iPSC cell-associated data.

R5 new remark: see below.

(12)

Remarks: There are major concerns here, which I summarize: images of iPSC cells should be shown together with graphs and not in separated figures; they are really poor in quality and again only single/few cortical neurons are shown. A higher number of cells should be shown. The panel seem to be cut at the level where branches continue so it is not possible to really judge about branching. The graph indicates that controls cells have something like 4 primary branching on average when in fact the only cell shown exhibit one long branch. How can we judge about reduction here? All the other panels showing 1 or 2 cells demonstrate similar branching number. Overall, the dots in the graph are too thick, this should be improved in all the graphs such to be able to see the data and statistics seems really strange (p-values too good?).

Response: We have removed iPSC cell-associated data.

R5 new remark: see below.

(13)

Remarks: I am familiar with neuronal branching. The explanation offered here is a pure speculation and I do not advise to have it in the main manuscript. Simply authors are comparing different cell types in different species having opposite results. On top of this, primary branching analysis was not performed in CaP motoneurons to be able to actually compare the same thing in different cell types. Also how does then the “longest neurite” in iPSC compare with neuritogenesis in CaP from fish? I do not think the data are strong enough to prove any specific involvement of the genes of interest in the development and neuritogenesis of both cell types and are conflicting. I propose to refrain the use of iPSC cells unless a better analysis is performed coupled to an improved discussion of the effects seen. Authors’ sentence is also conflicting: “This differential result is similar to what we observed in TBC1D23 deficient iPSC-derived neurons and zebrafish.” is not clear, as in zebrafish TBC1D23 MO increases branching.

Response: We have removed iPSC cell-associated data.

R5 new remark to major points regarding iPSC: It should be considered that failing to present an accurate iPSC data analysis reduces the overall impact of the work.

(14)

Remarks: I do not quite understand the answer, here. If each dot represents one CaP, and more CaP in one fish were analysed, then a nested ANOVA/graph should be performed or authors should show the mean of these 3 CaP they have analyzed, otherwise the graph is conceptually wrong. As the result could be driven by few fish having a major phenotype for instance with a clear clustering effect.

Response: We showed the mean of 3 CaP in each zebrafish that we have analyzed in the revised manuscript (new Fig. 7b and 8e).

R5 new remark: ok, authors please clarify how where the 3 CAP chosen, and provide the unit for the length (μm ?).

All the rest ok, please correct: Typo “activity” in Fig.2; Typo Line 157 “starvation” and not “starcation”.

Point-by-point responses:

We appreciate the reviewers' positive comments from reviewers. We also appreciate the reviewers' constructive suggestions that certainly helped us to improve the quality of our manuscript. To address the reviewer #5' concerns, we have performed further revisions to the revised manuscript. We carefully revised and proofread the manuscript as the editor suggested. Our point-by-point responses are listed below:

REVIEWERS' COMMENTS

Reviewer #3 (Remarks to the Author):

Reviewers' concerns have been appropriately addressed.

Response: We are glad we have addressed the reviewer's concern.

Reviewer #5 (Remarks to the Author):

(1) Remarks: I thank the authors for clarifying these points. However, they should explain the rationale of using different cell types for the different experiments performed (as they do at line 146/167 of the revised manuscript). I cannot see this properly done in the rest of the revised version of the manuscript. Line 132-133 for instance: the authors do not specify in which cells they do the first silencing (si) experiments, but then refer to HepG2 for CRISPR-Cas9 soon after leaving the reader to wonder why did they use different strategies without explaining the rationale. Authors should explain each time within the text why they switched to HUVEC and HepG2. Also, authors should specify the type of cells used in "Methods" and in the figure legends for each of the panels. Also, it should be clear from the text that a subset of the results was corroborated in HUVEC cells.

Response: Thank you for your constructive suggestions. We have improved the description and rationale of the experimental design (main text and methods). Since it was reported that LKB1 is required for activation of AMPK in HUVEC cells (PMID: 18250273), we first examined the effects of TBC1D23 knockdown on AMPK activation in HUVEC cells (Line 129-131). Since LKB1-AMPK pathway responds to energy stress, we also chose the hepatocyte cell line HepG2 in subsequent experiments (Line 135-136). We also added a description indicating that the results obtained in HepG2 cells were consistent with HUVEC cells (Line 137-138).

R5 new remark: I recognize that this aspect was improved.

Response: We are glad we have addressed the reviewer's concern.

(2) Remarks: I can see that co-localization analysis was majorly improved. Manders' coefficients are normally 2 (M1 and M2) with respect to pixels in Image-1 overlapping with Image-2 and vice versa. Authors can provide both, or specify the use of one. In the latter case, the label cannot be

“Manders’ coefficients” but rather “Manders’ coefficient 1 or 2”. The manuscript also still lacks a detailed explanation of the method used. For instance, which algorithm was used in ImageJ? Did the author use masking of a certain region of the cell? Again, a larger field of view should be always shown and at least two cells per condition.

Response: We appreciate the reviewer’s suggestions. The label for co-localization analysis has been corrected as “Manders’ coefficient 2”. We also included “(Golgin-97/Flag-LKB1)” or “(Golgin-97/Flag-AMPK α 1)” in y axis, which indicates the specific information and is better than “Manders’ coefficient 2”. We have improved our description of Methods as the reviewer suggested. Pearson’s correlation coefficients and Manders’ coefficients were calculated using JACoP plug-in of ImageJ. We select intact cells for analysis. In the revised manuscript, a larger field of view (at least two cells suggested by the reviewer) has been shown (new Fig. 2b).

R5 new remark: I recognize that this aspect was improved.

Response: We are glad we have addressed the reviewer’s concern.

(3) Remarks: Authors have improved the statistical analysis, overall. However, they should more clearly explain why a certain hypothesis test was used beyond citing t-test or ANOVA. Where the data checked for normality? Why did the authors choose Tukey vs Sidak etc ... as post-hoc test? Also in the “Reporting summary” authors claim to provide info such as “CI”, “degrees of freedom” ... though this reviewer did not find them reported in the manuscript. I suggest authors to make a supplementary table with all the statistical specs for each figure panel. Also, authors declare they provide “estimates of effect size” but I don’t see this in the manuscript. Same for “one side, two sides”.

Response: Statistical analyses were performed using GraphPad Prism 8 Software. No statistical method was used to predetermine the sample size. The data distribution was assumed to be normal, but this was not formally tested. The post-hoc test was recommended by GraphPad Prism software, according to the choice of multiple comparisons (compare each cell mean with the other cell mean in that row/column: Sidak; compare the mean of each column with the mean of every other column: Tukey). In the revised manuscript, the statistical significance of the difference between multiple groups (one variable), an ordinary one-way ANOVA was used, followed by Dunnett’s test. And the statistically significant differences between multiple comparisons (if two variables exist) were analyzed using the two-way ANOVA, followed by Sidak’s test. Differences were considered significant when $P < 0.05$. The statistical significance of the difference between two groups was determined using unpaired two-tailed t test. We have checked the “Reporting summary” carefully. We have improved the description of the analytic approaches and provided a supplementary table with all the statistical specs for each figure panel.

R5 new remark: I see the improvement, and I find the new table useful. However, still if no sample size was estimated, then “Estimates of effect sizes” should not be selected in the Reporting summary. Normality should be tested for each dataset in order to use the appropriate test (parametric and not parametric). I ask the authors to include these checks.

Response: We appreciate the reviewer's positive comments. We have corrected the "Reporting summary" as the reviewer suggested. Following your suggestion, normality was tested. An unpaired two-tailed Mann–Whitney test was used to determine significance between two groups of data without a normal distribution (Fig. 2a and 2b). Kruskal-Wallis test was used to determine significance between multiple groups of data (one variable) without a normal distribution (Fig. 6e, Fig. 8g and supplementary Fig. 7). And Scheirer-Ray-Hare Test was used to determine significance between multiple groups of data (two variables) without a normal distribution (Fig. 2h, Fig. 3h and Fig. 6a). All specific statistical details were included in the figure legends or supplementary table with all the statistical specs for each figure panel.

(4) *Remarks: N should be given for each experiment in the figure panel. When analyzing co-localization or Golgi elements from each cell, how many cells per experiment were counted? Do the dots in the graphs represent pooling of cells from different biological replicates? or different fields? Such aspects are not explained, however they are flagged in the Reporting Summary.*

Response: We apologize for not having explained this more clearly in our previous submission. We have included bar graphs (data were represented as mean \pm SD) for blot quantification in the revised manuscript. The information about the sample size, dots, and the number of replicates in individual experiments has been provided in the figure legends.

R5 new remark: I see the improvement. When the authors say "similar results were obtained in independent experiments" or "experiments were performed in triplicate (for instance Fig.2a), was the quantification done only on one replicate? (for instance: line 1076: n.of cells n=55 is it pooling cells from one replicate? Or three? If this is pooling from three, then a super-plot should be considered or at least a fold change of the different replicates).

Response: We appreciate the reviewer's positive comments. n.of cells indicates pooling cells from one replicate.

(5) *Remarks: what does "Normalized FRET/CFP" mean? Normalized vs what? Line 1296: "The FRET/CFP ratio was measured and normalized to cells incubated with DMEM" The authors should explain how was this normalization performed? What does it mean? Authors should really make an effort to use precise terminology and detail the analysis they performed such that readers are in condition to assess accuracy as well as to support reproducibility. Imaging and analysis using cells could further benefit of improvement in the description, along the lines of the details provided for zebrafish, see also the other points. A statement whether images and analysis within a certain experiment were obtained/performed with the same parameters with respect to acquisition and post-processing should be added, or otherwise explained if that is not the case.*

Response: We have improved the description of the experimental/analytic approaches. The FRET/CFP ratio of WT cells incubated with DMEM was set as 1, and FRET/CFP ratio of other groups was normalized to WT cells incubated with DMEM. For all cellular experiments, image acquisition and analysis within a certain set of experiment were obtained/performed with the same parameters. For zebrafish experiments, we use automatic exposure for each zebrafish to capture

higher quality images. Image analysis within a certain experiment were performed with the same parameters. We have included these statements in Methods as the reviewer suggested.

R5 new remark: Fig.2d: I still do not understand, I do not see any normalization. All the values plotted for WT are not 1, can authors also provides the raw measures and/or the calculation. How was intensity analyzed? Authors should show the ratiometric image in addition to this “normalized” quantification. This is important given that authors use a band-filter method not in a confocal setting, instead of a more accurate spectral unmixing methodology.

Response: The average FRET/CFP ratio of WT cells incubated with DMEM was set as 1, and FRET/CFP ratios of individual replicates from WT and other groups were normalized to this WT average (please see: Fig.2d and 2e). The raw FRET/CFP ratios were analyzed using LAS software (Leica). We have included the representative images in the revised manuscript (supplementary Fig. 2), as previously described (PMID: 25892241).

(6) Remarks: This limitation could be surpassed by using a LKB1 without tag in your transfection experiment. Using FRET to test this hypothesis would be more consistent, appropriate and elegant. WB per se would corroborate the finding. Given the specificity of the FRET AMPK sensor to Golgi or mitochondria, one would also expect to see the actual images of the FRET signal within cells. Why don't the authors show the images used to make calculations? (FRET and CFP channel and the computed ratiometric image)? It is somewhat bizarre that quantification is reported without representative images of the results. Why is now the scale bar and numbers of the graph (Fig.2d etc) different from that one submitted originally?

The authors claim that there is a strong reduction of AMPK activity in Golgi in cells depleted of TBC1D23 in both basal and energy stress condition by looking at the FRET data. However, if one examines carefully the graphs presented now, the data show that there is a little decrease of activity in basal conditions (1 vs 0.7, I guess), and that there is still a small -yes negligible-increase in activity upon stress also in TBC1D23 depleted cells. Authors should discuss these results more extensively along these lines and perhaps show a fold change quantification.

Response: The instrument we used for the FRET assay (Leica DMI6000B total internal reflection fluorescence microscope) served our institute for a very long time and was phased out by our institute about one year ago. Although the raw images are available, we were unable process the images as suggested due to lack of the original software. We failed too when we tried to open the images with third party software, such as Image J. To measure the AMPK activity at Golgi after reintroduction of LKB1 in HeLa cells, we had to take a different approach and measure the activation of AMPK at the Golgi by isolating Golgi fractionations and assessing changes by immunoblotting. As shown in new Supplementary Fig. 2d, introduction of LKB1 in HeLa cells remarkably promoted AMPK activation in Golgi fractions upon AICAR treatment.

The numbers of the graph in Fig.2d remain unchanged, and this panel does not contain a scale bar. Following the reviewer's suggestion (comment 14), we have removed iPSC cell-associated data (old Fig.6d and Fig.8a-c.). And following the reviewer's suggestion (comment 4), statistical significance of the difference between multiple groups was determined by ANOVA. Since we have confirmed that TBC1D23 KO does not affect Golgi assembly under basal conditions (Fig.3g-h), and we aimed to

compare the rescue ability of LKB1, Golgi-targeted LKB1 (LKB1-Giantin) WT and its kinase-dead mutant after metformin treatment. Thus, indicated groups in Fig.6d were included in the revised manuscript. Since the reviewer suggested a larger field of view, we updated our images to include more cells and added a scale bar.

We have revised our discussion about “Golgi-AMPK activity reduction in cells depleted of TBC1D23 in both basal and energy stress condition” as the reviewer suggested (Line 165-168).

R5 new remark: I understand that there are circumstances where the raw data can be missing, but I do not think it should be acceptable for a journal such as Nature Communications with the principle of FAIR data to have FRET ratiometric measurements performed with a simple method of filter-based imaging at epifluorescence microscope without being able ever and upon request to show the representative microscopy images. If authors can provide the same assessment in western blot I recommend to remove the FRET data from the paper. Also, I see that the authors now show Golgi-AMPK activity reduction in cells depleted of TBC1D23 in both basal and energy stress condition, but on the other hand they should still discuss the significant increase in AMPK activity shown by FRET from basal to GS even in the absence of TBC1D23 (my original question), what other mechanisms are in place then?

Response: We have included the representative images of YFP and FRET/CFP in the revised manuscript (supplementary Fig. 2). As we mentioned in the last round of response letter, we did NOT miss raw data. Our problem was that the software, which could analyze our raw images, was phased out together with our instrument. After spending huge amount of effort, we were able to obtain a copy of the software from the manufacturer and to provide these images as suggested.

We assume that other unidentified factor(s) might contribute to the activation of AMPK in TBC1D23 KO cells. We have discussed this point as suggested (Line 167-169).

(7) Remarks: I only see FRET/CFP quantification but not WB quantification. The reason to use AICAR provided here should be included in the text. N of replicates from which quantifications are performed should be specifically provided in figure legends.

Response: We have included bar graphs (data were represented as mean \pm SD) for immunoblot quantification in the revised manuscript, and the information about the number of replicates has been provided in the figure legends.

R5 new remark: The increase in AMPK activation upon AICAR treatment of cells expressing LKB1 should be better explained. Activation can be observed also without AICAR in cells expressing LKB1. Fig.2d has no quantification.

Response: Since LKB1 is responsible for AMPK activation in response to energy stress, the ectopic expression of LKB1 in HeLa (LKB1^{-/-}) cells promotes AMPK activation upon AICAR treatment. We have included the explanation in the revised manuscript as the reviewer suggested. As shown in new Fig.S3d, stable introduction of LKB1 in HeLa cells results in its partial localization in Golgi fraction, which might be responsible for moderate activation without AICAR treatment.

Quantification of new Fig.S3d has been included in the revised manuscript.

(8) *Remarks: Basic details of image analysis are missing still. What do the authors mean now by “unbiased image analysis”? Do they mean “blind”? A better analysis should also consider the nucleus dimension, a ratio between the total area occupied by GM130 and the nucleus are should be added, which is less prone to errors as compared to manually counting fragments.*

Response: We have checked our manuscript carefully and improved the description the analytic approaches. We indeed mean “blind”, which has been corrected in the revised manuscript. And Golgi assembly analysis using the ratio between the total area occupied by GM130 and the nucleus has been added in the revised manuscript(Fig. 3h and Fig.6e).

R5 new remark: I recognize that this aspect was improved.

Response: We are glad we have addressed the reviewer’s concern.

(9) *Remarks: In Fig.6d, the field still only one cell is shown for the LKB1 and rescue experiment so it is hard to judge. Again, the GM130 channel is too low and the quality of the image seems worst compared to submitted. Also, the authors should paid attention to the phrasing and English. For instance, line 232 and 1117: “knockout of TBC1D23 strongly impairs Golgi disassembly” seems strange. Most likely authors wanted to say that TBC1D23 KO prevents Golgi disassembly upon Met treatment.*

Response: Thank you for pointing this out. We have corrected “impairs” as “prevents” as the reviewer suggested. In Fig.6d, two cells were shown for LKB1, and the images were the same as the original submission. We have replaced the images with TIF files with higher resolution.

R5 new remark: I recognize that this aspect was improved.

Response: We are glad we have addressed the reviewer’s concern.

(10) *Remarks: I see the brain images improved. Still,if authors used different laser energy or parameters because of the difference in fluorescence, they should state the specific setting in the Methods. In Fig.8g they should explain what exactly “Relative midbrain size” means. Is it an area relative to what and how was it calculated? (again should be added in “Methods”). With respect to different brain regions, authors should clearly describe that the effect is observed on many different regions regardless of the association with human patients in PCH... otherwise, this is the classical cherry-picking attitude in selecting the data to report. The impact of these LoF on the whole brain could be functionally important for brain development understanding and might underly differences across vertebrates*

Response: We appreciate the reviewer’s comment. We have improved our description of Methods as reviewer suggested and included information about exposure times in the Methods. We characterized midbrain size by measuring relative area, and fluorescence intensity did not affect

our measurements. In lateral views of zebrafish, the midbrain size quantified by ZEN3.1 software was defined as relative midbrain size. We indeed observed that the development of the whole brain is affected in morphants, including midbrain, cerebellum and hindbrain. These descriptions have been included in our revised manuscript.

R5 new remark: ok, I see the improvements but authors must explain what “relative” means (relative to what?) isn't an absolute measurement? If yes please remove “relative”. Relative means it is normalized to something (body area? Head area? Whole brain area?). Also maybe authors have to specify that they analyze “dorsal midbrain/optic tectum” area and not the whole midbrain (judging from the ROI designed) Also the graph should definitely provide the measurement unit (μm^2). Line 379: why did you chose “midbrain”? The sentence is too bold, other areas are interested as well. So authors should claim they use “midbrain” as a proxy of brain underdevelopment in general. Why did the author not show analysis in the cerebellum, forebrain? affected in the disease. Authors should rephrase : “PCH-like phenotypes in zebrafish” in abstract and whenever they talk about it. Loss of *TBC1D23* also causes impaired cortical development, right? The phenotype observed is not PCH-like or we could not tell, it is a neurodevelopmental phenotype involving different brain areas in fish at best.

Response: We appreciate the reviewer’s thoughtful comments and suggestions. We measured the midbrain size in lateral views of zebrafish, which is an absolute measurement. The unit for the measurement is μm^2 . We observed that neuronal loss was particularly evident in the midbrain, cerebellum and hindbrain of morphants, which manifested altered brain morphology. Because the midbrain size could be readily measured in lateral views of zebrafish, we chose the midbrain size to characterize the defective brain development, similar to our previous studies (PMID: 31624125, 32453802). As the reviewer suggested, we have rephrased the “PCH-like phenotypes in zebrafish” to “neurodevelopmental abnormalities in zebrafish” in the revised manuscript (Line 36-37). These changes have been included in the revised manuscript.

(11) Remarks: I understand the technical issue, but this kind of approximation is not acceptable, and could be solved by sorting the positive cells and showing the reduced *TBC1D23* expression. If authors want to keep the iPSC cell data, they should better show the efficiency of reduction from these cells. Also, the differentiation in cortical neurons is not specified and should be described. Importantly, I am concerned about the speculative interpretation the authors make based on the iPSC poor quality data -see my point below.

Response: We have removed iPSC cell-associated data.

R5 new remark: *see below*.

(12) Remarks: The are major concerns here, which I summarize: images of iPSC cells should be shown together with graphs and not in separated figures; they are really poor in quality and again only single/few cortical neurons are shown. A higher number of cells should be shown. The panel seem to be cut at the level where branches continue so it is not possible to really judge about branching. The graph indicates that controls cells have something like 4 primary branching on

average when in fact the only cell shown exhibit one long branch. How can we judge about reduction here? All the other panels showing 1 or 2 cells demonstrate similar branching number. Overall, the dots in the graph are too thick, this should be improved in all the graphs such to be able to see the data and statistics seems really strange (p-values too good?).

Response: We have removed iPSC cell-associated data.

R5 new remark: *see below*.

(13) Remarks: I am familiar with neuronal branching. The explanation offered here is a pure speculation and I do not advice to have it in the main manuscript. Simply authors are comparing different cell types in different species having opposite results. On top of this, primary branching analysis was not performed in CaP motoneurons to be able to actual compare the same thing in different cell types. Also how does then the “longest neurite” in IPSC compare with neuritogenesis in CaP from fish? I do not think the data are strong enough to prove any specific involvement of the genes of interest in the development and neuritogenesis of both cell types and are conflicting. I propose to refrain the use of iPSC cells unless a better analysis is performed coupled to an improved discussion of the effects seen. Authors’ sentence is also conflicting: “This differential result t is similar to what we observed in TBC1D23 deficient iPSC-derived neurons and zebrafish.” is not clear, as in zebrafish TBC1D23 MO increases branching.

Response: We have removed iPSC cell-associated data.

R5 new remark to major points regarding iPSC: It should be considered that failing to present an accurate IPSC data analysis reduces the overall impact of the work.

Response: We agree that the iPSC data provide another layer of functional/phenotypical validation of our main conclusions, but do not regard it as essential. We removed this data since we agreed with the reviewer’s previous comments that this data needs more validation. Also, we discussed the limitations of our study concerning the lack of iPSC data (Line 509-510).

(14) Remarks: I do not quite understand the answer, here. If each dot represents one CaP, and more CaP in one fish were analysed, then a nested ANOVA/graph should be performed or authors should show the mean of these 3 CaP they have analyzed, otherwise the graph is conceptually wrong. As the result could be driven by few fish having a major phenotype for instance with a clear clustering effect.

Response: We showed the mean of 3 CaP in each zebrafish that we have analyzed in the revised manuscript(new Fig.7b and 8e).

R5 new remark: ok, authors please clarify how where the 3 CAP chosen, and provide the unit for the length (μm ?).

All the rest ok, please correct: Typo “activity” in Fig.2; Typo Line 157 “starvation” and not “starcation”.

Response: We appreciate the reviewer's positive comments. We select approximately the third to fifth CaP axon located on the yolk extension, which better represents the average length of CaP axon. The unit for the length is μm . We have included the information for CaP analyses in the figure legends as the reviewer suggested. We have fixed the typo and checked the manuscript carefully.